# *Elp2* mutations perturb the epitranscriptome and lead to a complex neurodevelopmental phenotype

Marija Kojic [1,2], Tomasz Gawda [3], Monika Gaik[3], Alexander Begg[2], Anna Salerno-Kochan[3,4], Nyoman D. Kurniawan[5], Alun Jones[2], Katarzyna Drożdżyk[3], Anna Kościelniak[3], Andrzej Chramiec-Głąbik[3], Soroor Hediyeh-Zadeh [6,7], Maria Kasherman[8], Woo Jun Shim [2,9], Enakshi Sinniah [2], Laura A. Genovesi[1,2], Rannvá K. Abrahamsen [10], Christina D. Fenger[10], Camilla G. Madsen[11], Julie S. Cohen [12,13], Ali Fatemi [12,13,14], Zornitza Stark [15,16,17], Sebastian Lunke[16,17,18], Joy Lee[15,19], Jonas K. Hansen[20], Martin F. Boxill[20], Boris Keren[21], Isabelle Marey[21], Margarita S. Saenz[22], Kathleen Brown[22], Suzanne A. Alexander[23,24], Sergey Mureev[25], Alina Batzilla [2,26], Melissa J. Davis [6,7,27], Michael Piper[8], Mikael Bodén [9], Thomas H. J. Burne [23,24], Nathan J. Palpant [2], Rikke S. Møller[10,28], Sebastian Glatt [3✉] & Brandon J. Wainwright [1,2✉]

Intellectual disability (ID) and autism spectrum disorder (ASD) are the most common neurodevelopmental disorders and are characterized by substantial impairment in intellectual and adaptive functioning, with their genetic and molecular basis remaining largely unknown. Here, we identify biallelic variants in the gene encoding one of the Elongator complex sub-units, ELP2, in patients with ID and ASD. Modelling the variants in mice recapitulates the patient features, with brain imaging and tractography analysis revealing microcephaly, loss of white matter tract integrity and an aberrant functional connectome. We show that the *Elp2* mutations negatively impact the activity of the complex and its function in translation via tRNA modification. Further, we elucidate that the mutations perturb protein homeostasis leading to impaired neurogenesis, myelin loss and neurodegeneration. Collectively, our data demonstrate an unexpected role for tRNA modification in the pathogenesis of monogenic ID and ASD and define Elp2 as a key regulator of brain development.

A full list of author affiliations appears at the end of the paper.

C orticogenesis is a highly dynamic and orchestrated multi-step process that involves the proliferation of neural progenitors, their differentiation, migration to the respective destinations, and synaptogenesis[1]. Malformations during cortical development resulting from genetic alternations lead to severe functional consequences including microcephaly, intellectual disability (ID), and neurodevelopmental delay[2,3]. ID is characterized by severe impairments in cognition and adaptive daily life skills functioning[4]. It commonly occurs in combination with global developmental delay, autism spectrum disorder (ASD), epilepsy, and ataxia[4–6]. Technological advances in high throughput sequencing and data analyses have provided insights into the complex genetic predispositions of neurodevelopmental disorders including ID[7–9]. Nonetheless, the underlying molecular mechanisms remain to be defined.

Studies of familial dysautonomia (FD) patients have implicated one of the Elongator complex subunits, ELP1, in the etiology of autonomic nervous system dysfunction[10–13]. A role for Elongator in the central nervous system (CNS) has been indicated through identified central pathology in some FD patients, including myelination[14] and retinal defects[15,16] and the population genomic studies that linked ELP3 variants with amyotrophic lateral sclerosis (ALS)[17,18], variants in the ELP2 gene with ID[19–21] and ELP4 mutations with ID, autism and epilepsy[22–25]. Furthermore, studies in mice showed that a germline mutation in the Elp6 gene led to Purkinje neuron (PN) degeneration[26], retina-specific loss of Elp1 resulted in retinal ganglion cell degeneration[27] and a conditional Elp1 deletion in the CNS perturbed the development of cortical neurons ultimately leading to their death and consequential learning and memory impairment[28]. A number of cell activities have been assigned to the complex[29], but it is widely accepted that its main function is the formation of 5-carboxymethyl-uridine (cm5U) in the wobble base ($U_{34}$) position of 12 mammalian tRNAs[30,31]. These $U_{34}$ modifications tune ribosomal speed during translational elongation, specifically affecting tRNA binding, recognition, rejection, decoding and translocation. Hence, the lack of $U_{34}$ modifications leads to changes in the co-translational folding dynamics of nascent polypeptide chains, promotes misfolding of newly synthesized proteins, and triggers detrimental cellular responses related to protein aggregation in all organisms tested[32,33]. In particular, the translational decoding of A/U-rich codons[34] requires additional stabilization by these modifications due to their weaker codon-anticodon pairing, which would otherwise by default lead to translational infidelity and proteostasis[35]. Reduced levels of the Elongator-dependent tRNA modifications have been confirmed in FD, ALS, and cerebellar ataxia as a consequence of ELP1[13], ELP3[36], and Elp6[26] mutations, respectively.

The dodecameric Elongator complex consists of two copies of each of the six subunits (Elp1-6) and forms two discrete sub-complexes, namely Elp123 and Elp456[37]. In yeast, each of the subunits is equally important for the tRNA modification activity of the complex, although it has recently been shown that the yeast Elp123 is sufficient to bind substrate tRNA and prime the active site within the Elp3 subunit in vitro[38]. The Elp2 subunit is essential for the stability and function of the catalytic Elp123 sub-complex[39].

In this work, we identify six patients presenting with a range of features including global developmental delay, ID, ASD, and drug-resistant epilepsy with ELP2 genetic variants. In combination with other case reports with a similar phenotype[19–21], we characterize eight patients in total with ELP2 variants indicating a high likelihood that these variants are associated with neurodevelopmental anomalies. We demonstrate that ELP2 variants impair ELP2 protein stability to decrease Elongator complex activity and are sufficient to recapitulate the human clinical phenotype in mice. Furthermore, we show that Elp2 is a key regulator of cortical neuron connectivity, morphology and neural progenitor cell dynamics by maintaining the translation of large AA-biased transcripts and consequently the global cortical proteome. Our comprehensive study of the clinically relevant ELP2 mutations reveals the role of epitranscriptomic processes in brain development and homeostasis which, when perturbed, lead to profound CNS defects in both humans and model systems.

## Results

**ELP2 variants cause developmental delay, ID, and autism.** As part of routine clinical studies, we identified eight individuals with missense and null mutations in the ELP2 gene (Supplementary Table 1). The most prominent clinical features consisted of severe ID, ASD, and profound developmental delay. The affected individuals were non-verbal and showed signs of motor impairment, along with hypotonia, spasticity, feeding, and walking difficulties, including cerebral palsy. Magnetic resonance imaging (MRI) revealed microcephaly, severe brain atrophy, loss of white matter, and a thin corpus callosum (Fig. 1a–c). In total, nine different ELP2 variants were identified by whole-exome sequencing (Fig. 1d). Compound heterozygosity for two missense ELP2 mutations ELP2H206R and ELP2R462W (Supplementary Table 1, Patients 1 and 2) has been previously reported[19]. Parents of all the affected individuals were found to be heterozygous carriers of the disease-causing variants.

**Elp2 mutations perturb the protein stability.** We performed in vitro analyses of purified human (hElp2) and murine (mElp2) Elp2 proteins to assess the molecular consequences of the single amino acid substitutions found in the affected patients. Based on the high amino acid sequence conservation among the eukaryotic Elp2 proteins (Supplementary Fig. 1a) and the known crystal structure of yeast Elp2[37,39], we created a structural model of hElp2. Next, we positioned hElp2 in the context of Elp1 and Elp3 utilizing the available high-resolution cryo-EM reconstruction of the yeast Elp123 complex[40]. Using this approach, we precisely localized the individual mutations in the structure of Elp2, and in the context of the assembled sub-complex. The L98F and H206R are positioned in structured loop regions on the surface of the first WD40 domain of Elp2, whereas T405I and R462W/Q are found in β-strands of the second WD40 domain (Fig. 2a). Notably, H206R and R462Q/W are located in the interface between Elp2 and the S-adenosyl methionine domain (SAM) of Elp3, which represents a crucial hotspot for the stability and catalytic activity of Elp3 and the whole Elongator complex[40,41].

To define the influence of patient-derived mutations on the individual Elp2 protein, we purified mElp2 and hElp2 to homogeneity. All purified Elp2 proteins were found to be monomeric, like yeast Elp2[37], showing an expected molecular weight of ~95 kDa (Fig. 2b, Supplementary Fig. 1b). Next, we measured the stability of wild-type mElp2 and hElp2 and compared their temperature-dependent unfolding rates with the mutated versions. The thermal shift assays revealed that both hElp2 and mElp2 showed lower thermal stability than fungal Elp2[37]. Strikingly, the T405I, R462Q, R462W mutations induced a significant destabilization of both mammalian Elp2s, which on the one hand manifested itself by a high fluorescent signal already at low temperatures early in the measurements and on the other hand by decreased calculated melting temperatures ($T_m$; Fig. 2c, Supplementary Fig. 1c–e). No significant destabilization was observed for the L98F and H206R mutations. The relative severity of the individual mutations was found to be similar for murine and human Elp2 proteins. In summary, the identified Elp2

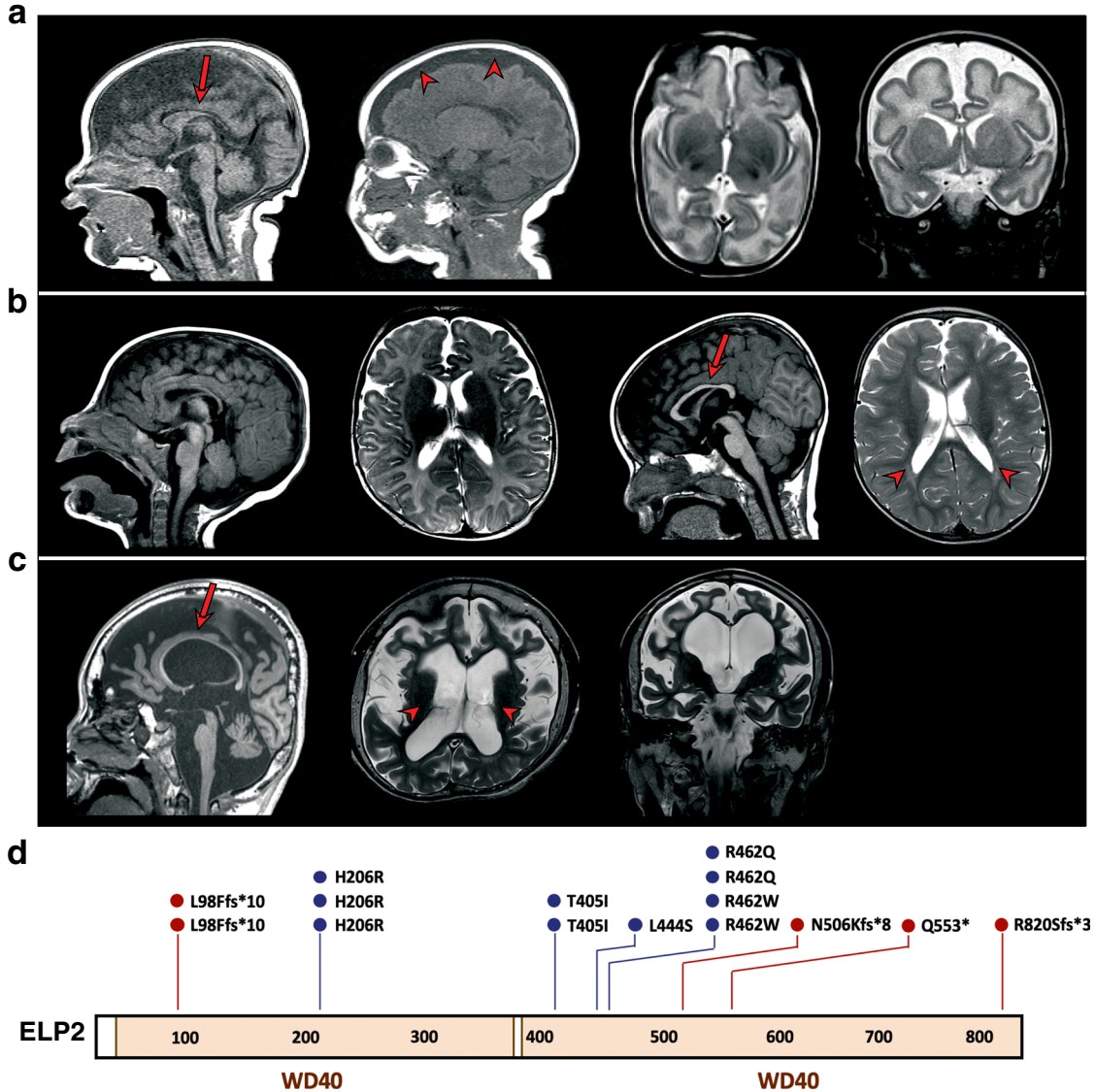

**Fig. 1 Molecular and clinical information on the *ELP2* variants found in patients with intellectual disability and autism. a** Mid- and lateral-sagittal (left), transversal (middle), and coronal (right) MRI scans of Patient 4 (refer to Supplementary Table 1.) revealing microcephaly with few and shallow sulci. **b** Sagittal and transversal MRI brain scans of Patient 5 showing normal brain at 3 months (two panels left) and decreased white matter, secondary ventricular enlargement, and thin corpus callosum at 19 months of age (two panels right). **c** Sagittal (left), transversal (middle) and coronal (right) brain MRI scans of 17-years-old Patient 6 showing microcephaly and severe brain atrophy (cortex, cerebellum, and brain stem atrophy) with white matter loss, secondary ventricular enlargement, and thin corpus callosum. Arrows indicate thin corpus callosum and arrowheads point to a broadened subarachnoid space in (**a**) and ventricular enlargement in (**b**) and (**c**). **d** Schematic representation of the *ELP2* mutations identified in patients with intellectual disability and autism. WD40 protein domains are shown in orange. Position of missense (in blue) and frameshift (fs) mutations (in red) is indicated.

variants lead to decreased stability of the otherwise well-behaved individual Elp2 proteins in isolation.

**Elp2 mutations compromise the activity of the Elongator complex**. In yeast, deletion of any of the six Elongator subunits leads to identical phenotypes and loss of Elongator-dependent tRNA modifications[31]. Elp2 is suspected to act as a scaffold protein for the other Elongator subunits and does not possess any detectable enzymatic activity or clearly defined active site. Hence, the decreased stability of Elp2 may indirectly affect the integrity and activity of the other subunits and the whole Elongator complex. We investigated the impact of the individual mutations on the human Elp123 (hElp123) sub-complex, which harbors the enzymatically active Elp3 subunit and is able to bind tRNAs via

the C-terminus of Elp1 and the active site of Elp3[40] (Fig. 3a). The expression level of Elongator in human cells is rather low and the reconstitution from recombinantly produced individual subunits is impeded by several yet unresolved technical challenges[41]. Accordingly, we employed the recently developed BigBac system[42] to simultaneously produce all three human subunits (Fig. 3b) and ultimately obtain pure, homogenous, and stoichiometric wild-type hElp123 (hElp123$_{WT}$). The complex eluted at an estimated molecular weight of ~625 kDa, showing that hElp123$_{WT}$ also forms a dimer of trimers harboring two copies of each of the three subunits (Supplementary Fig. 2a).

First, we tested the affinity of purified hElp123$_{WT}$ to in vitro transcribed fluorescently labeled human tRNA$^{Arg}_{ACG}$ and human tRNA$^{Arg}_{UCU}$ using microscale thermophoresis (MST; Fig. 3c). Importantly, hElp123$_{WT}$ did not selectively bind modifiable

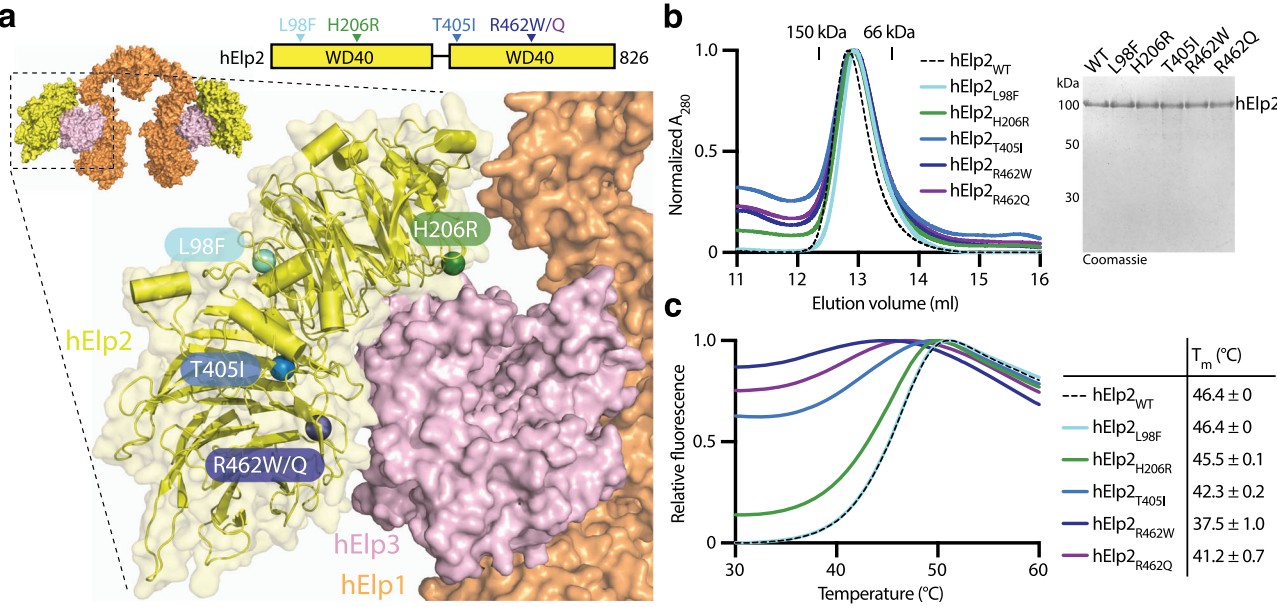

**Fig. 2 Mutations found in patients with intellectual disability and autism impair the stability of hElp2 protein. a** Localization of patient-derived mutations in the context of human ELP123 (hElp123) model prepared using Cryo-EM structure of yeast Elp123 (yElp123; PDB 6QK7). Color code for proteins: hElp1—orange, hElp2—yellow, hElp3—pink. Mutations are depicted in different colors on hElp2 domain organization scheme and a close-up of hElp123 model. **b** Purification of wild-type (WT) and mutant hElp2. Gel filtration profiles from S200 Increase 10/300 GL column (left) and Coomassie-stained SDS-PAGE gel showing purified hElp2 variants (right; $n = 3$ independent experiments). **c** Averaged melting curves from thermal shift assay for hElp2 variants with calculated melting temperatures (Tm) (mean ± SD; $n = 3$ independent measurements). Source data are provided as a Source Data file.

tRNAs, as the non-modifiable tRNA$^{Arg}_{ACG}$, which lacks a U$_{34}$ base and does not represent a canonical Elongator target, showed a similar affinity. The dissociation constants for hElp123 align with previously described values for yeast Elp123 and archaeal Elp3[40,43]. Next, we examined the enzymatic activity of hElp123$_{WT}$ by measuring tRNA-induced acetyl-CoA (ACO) hydrolysis rates. ACO provides the acetyl group that is transferred to U$_{34}$ to form cm$^5$U$_{34}$ in the initial step of the modification. Our data show that the activity of hElp123$_{WT}$ was stimulated by tRNA$^{Arg}_{UCU}$, while tRNA$^{Arg}_{ACG}$ did not stimulate its hydrolysis rate (Fig. 3c). Our results using MST and ACO assays confirm that hElp123$_{WT}$, like bacterial and archaeal stand-alone Elp3 proteins[43,44], is non-selective towards the binding of different tRNA species, but strictly requires U$_{34}$ to initiate the modification reaction.

We then asked whether the identified single amino acid substitutions in Elp2 influence the assembly and the tRNA-induced ACO hydrolysis activity of the Elp123 sub-complex. Hence, we introduced the substitutions into the hElp123$_{WT}$ construct to obtain hElp12$_{L98F}$3, hElp12$_{H206R}$3, hElp12$_{T405I}$3, hElp12$_{R462W}$3, and hElp12$_{R462Q}$3. Despite the strong effects on the stability of the individual hElp2 proteins, no significant changes were observed during protein purification of the substituted hElp123 complexes (Fig. 3b). Therefore, the respective *Elp2* mutations do not affect Elongator at the stage of complex assembly. The purity level and degradation patterns varied between individual purifications of the same complex. Nonetheless, these variations might also indicate minor effects on complex formation, stability, and integrity for hElp12$_{L98F}$3, hElp12$_{H206R}$3, and hElp12$_{R462W}$3 (Supplementary Fig. 2b). Strikingly, the tRNA-induced ACO hydrolysis activity assay showed a significant decrease for all five tested mutants in comparison to hElp123$_{WT}$ (Fig. 3d). Our results clearly demonstrate that the identified patient-derived mutations influence the stability of both tested mammalian Elp2 proteins and have a detrimental effect on the ability of hElp123 to induce the initial step of the cm$^5$ modification reaction in vitro and likely impose a similar effect in vivo.

**Elp2H206R and Elp2R464W cause global developmental delay, microcephaly, and ID/ASD-like phenotypic features in mice.** To gain insight into the neuropathological consequences of the *Elp2* mutations, we employed CRISPR-Cas9 gene editing to create a null *Elp2* mouse allele and also introduce the most commonly found *Elp2* mutations in mice, *Elp2H206R* and *Elp2R464W*. Compound heterozygosity of *Elp2* null and two missense mutations (*Elp2H206R/-* and *Elp2R464W/-*) and homozygous *Elp2R464W* resulted in early embryonic lethality (Supplementary Fig. 3a–d), while compound heterozygosity for both missense mutations (*Elp2H206R/R464W*) was found to be lethal during late gestation in C57BL/6 mice (Supplementary Fig. 3a, e). The *Elp2H206R* homozygous mice (the mutation shown to be less severe than *Elp2R464W* in vitro) exhibited physical manifestations of developmental delay (Supplementary Fig. 4a–c), microcephaly (Supplementary Fig. 4a, d), severe motor defects, and hypotonia (Supplementary Fig. 4e), but also abnormal hind-limb clasping (Supplementary Fig. 4f) and a stereotypic behavior in a form of repetitive self-grooming (Supplementary Fig. 4g), both frequently found in ID/ASD mouse models[45–48]. Ultimately, the severe phenotype of these mice was lethal at postnatal day 21 (P21) (Supplementary Fig. 4h).

In order to study the consequences of the *Elp2* mutations on a mature brain in mice beyond P21, we crossed *Elp2R464W* and *Elp2H206R* mice onto a different genetic strain, namely DBA2/J. Consistent with our findings in the C57BL/6 mice, *Elp2R464W* homozygosity was shown to be embryonically lethal and the phenotypic features of *Elp2H206R* homozygous and *Elp2H206R/R464W* mice were consistent with those found in *Elp2H206R* mutants on the C57BL/6 background. These include global developmental delay characterized by reduced body size (Fig. 4a, b) and weight (Fig. 4c), microcephaly (Fig. 4a, d), the phenotype highly similar to the patient features consisting of tremor, motor defects, and hypotonia (Fig. 4e), abnormal limb-clasping (Fig. 4f) and repetitive behavior (Fig. 4g). The lifespan of *Elp2H206R* and *Elp2H206R/R464W* mice was P120 and P60, respectively (Fig. 4h). Consistent with clinical findings and other ID/ASD mouse

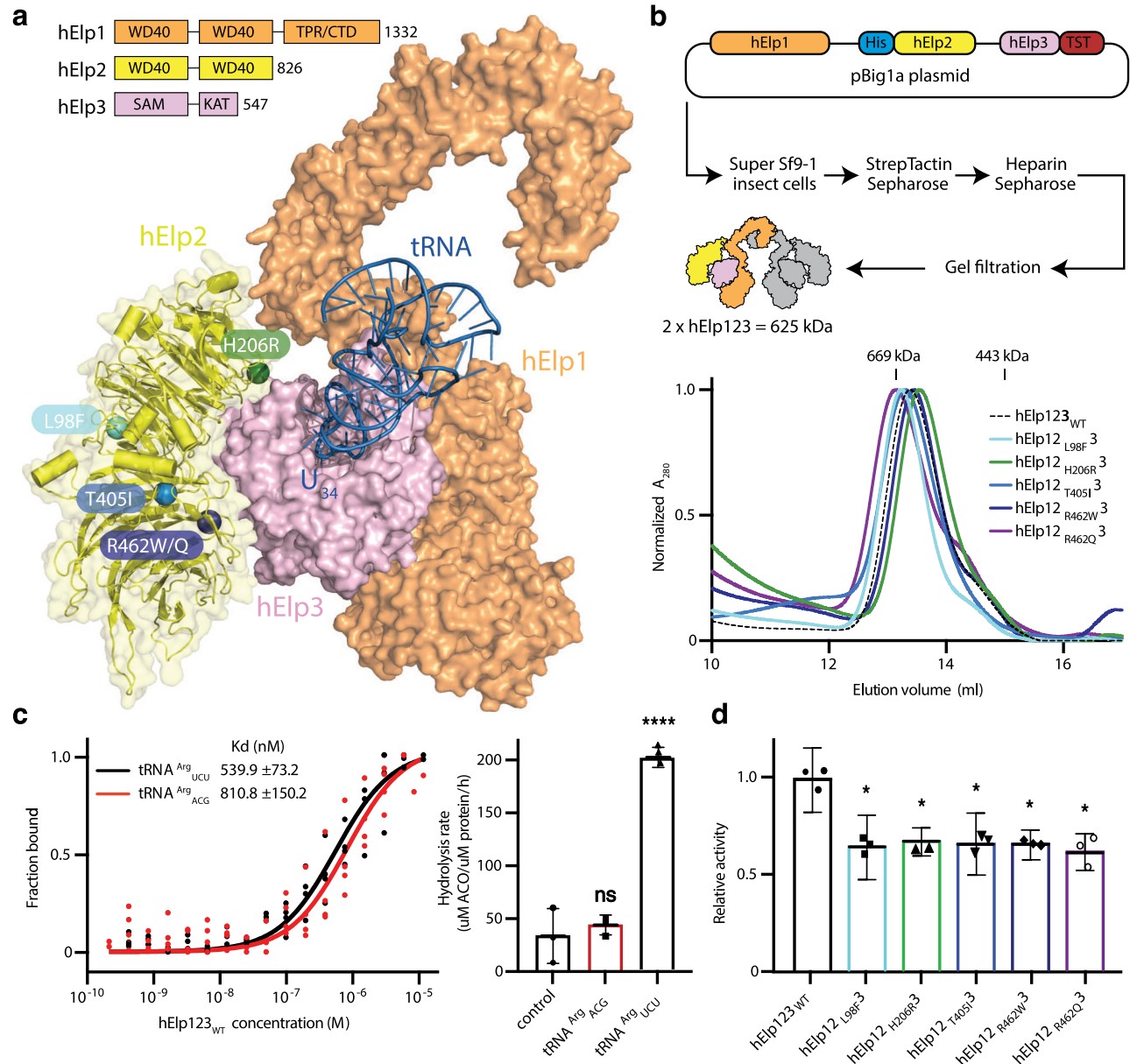

**Fig. 3 Patient *ELP2* mutations decrease Elongator activity. a** Human ELP123 (hElp123) domain organization scheme: TPR tetratricopeptide repeat domain, CTD C-terminal domain, SAM S-adenosyl methionine domain, KAT lysine acetyltransferase domain (top). Human ELP123 model with bound tRNA and depicted *ELP2* mutations (bottom). Color code for molecules: hElp1—orange, hElp2—yellow, hElp3— pink, tRNA—blue. **b** Workflow scheme for hElp123 wild-type (hElp123$_{WT}$) protein production and purification from insect cell expression system (top). Gel filtration profiles from Superose 6 Increase 10/300 GL column for all hElp123 variants (bottom; $n = 3$ independent experiments). **c** Microscale thermophoresis analysis of hElp123$_{WT}$ binding to tRNA$^{Arg}_{UCU}$ or tRNA$^{Arg}_{ACG}$ with estimated dissociation constant values ($K_d$) and $K_d$ fitting errors (left; mean ± SD; $n = 4$ independent measurements). Acetyl-CoA hydrolysis activity assay results for hElp123$_{WT}$ incubated with no tRNA (control), tRNA$^{Arg}_{ACG}$, or tRNA$^{Arg}_{UCU}$ (right; $n = 3$ independent experiments). **d** Normalized acetyl-CoA hydrolysis activity assay results for hElp123 variants incubated with tRNA$^{Arg}_{UCU}$ ($n = 3$ independent experiments). Statistical analysis: one-way ANOVA ($\alpha = 0.05$) with a Dunnett's post-hoc test. Statistically significant differences are indicated (*$p \leq 0.05$; ****$p \leq 0.0001$; ns not significant: $p = 0.6823$), Data represent mean ± SD. Source data are provided as a Source Data file.

models[49], *Elp2* mutant neonates had a severe communication deficit in the form of a markedly reduced number and duration of ultrasonic calls (Fig. 4i, j). In summary, the *Elp2* mutant mice recapitulated the clinical features of the patients, including developmental delay with microcephaly, repetitive behavior, and abnormal motor and vocal characteristics.

**Elp2 is a key regulator of the brain connectome**. In order to further define the role of Elongator-dependent regulation of CNS homeostasis, we assessed regional brain volume and connectivity

since differences in these features are commonly observed in ID/ASD[50,51]. We performed MRI imaging of the brain tissue collected from mutant animals to detect any possible neuroanatomical-neurobehavioral correlations. The volumes of different brain structures were analyzed and similar to the MRI results in the patients, *Elp2H206R* mice were found to have reduced white matter (internal capsule) and cortical volume (Supplementary Table 2).

While conventional MRI is limited to the anatomical mapping of a tissue, diffusion tensor MRI (DT-MRI) allows mapping of

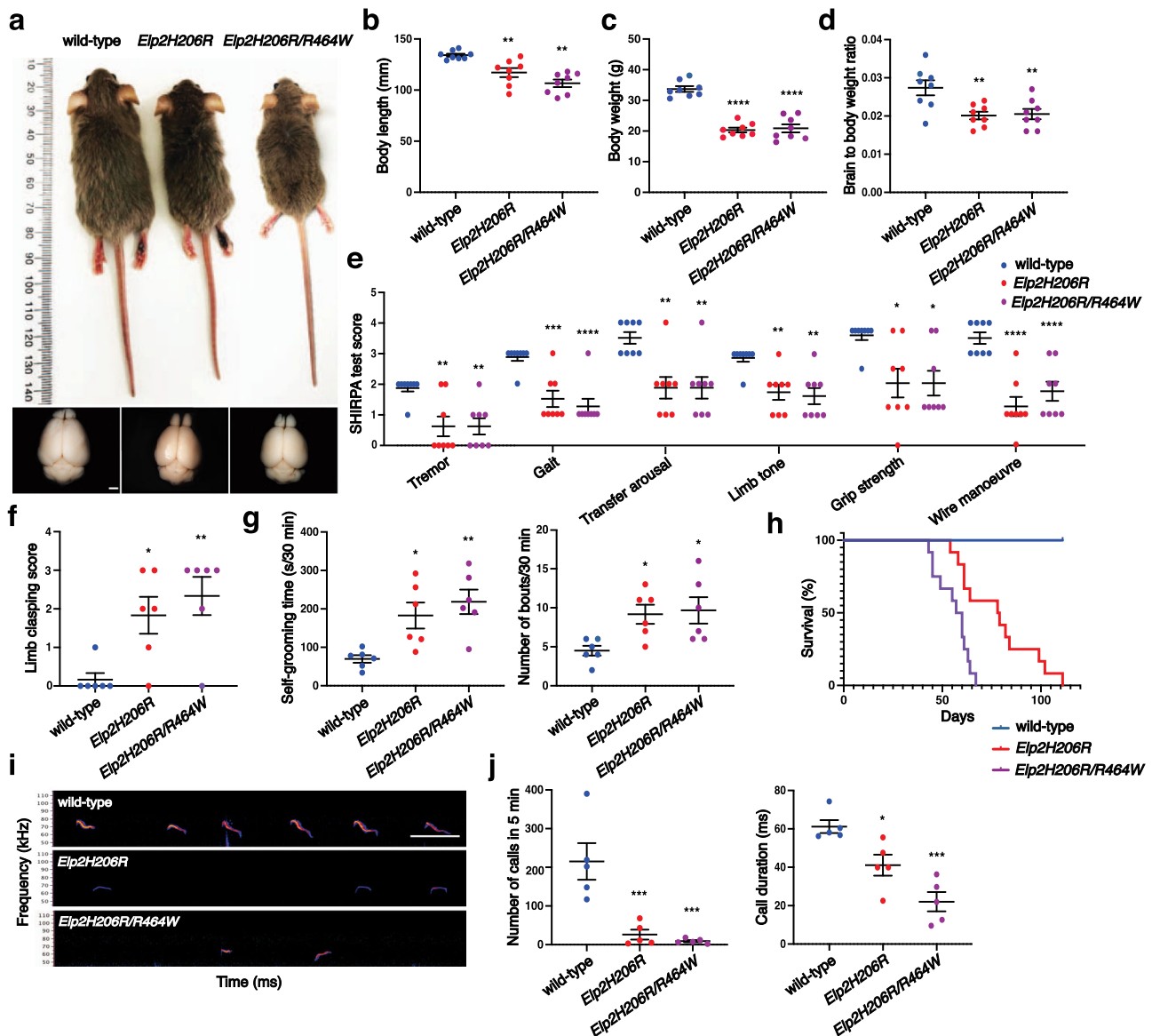

**Fig. 4 Developmental delay, microcephaly, motor defects, and autistic features in *Elp2H206R* and *Elp2H206R/R464W* mice. a** Appearance of an *Elp2H206R* homozygous and *Elp2H206R/R464W* compound heterozygous mouse on DBA2/J genetic background and the control littermate (upper panel), as well as their dissected brains (lower panel) at postnatal day 60 (P60). **b** Body length, **c** weight, and **d** relative brain weight of the indicated genotypes were quantified at P60. **e** Significant effects of *Elp2H206R* and *Elp2H206R/R464W* on scores for SHIRPA test at P60. **f** Abnormal hindlimb clasping and **g** repetitive self-grooming (recorded for 30 min in a home cage) shown as prolonged self-grooming time and increased number of self-grooming bouts in P60 *Elp2* mutant mice vs. control littermates. For (**a–e**) $n = 8$ (four males and four females), for (**f**) and (**g**) $n = 6$ (three males and three females). **h** Kaplan–Meier curve of mouse survival ($n = 12$). **i** Representative spectrograms of ultrasonic vocalizations emitted upon separation of juvenile wild-type, *Elp2H206R* and *Elp2H206R/R464W* mouse pups from their nests at P7. **j** Total number and duration of ultrasonic vocalization call emitted during a 5-min recording interval ($n = 5$ mouse pups per genotype). Scale bars, 2 mm (**a**) and 100 ms (**i**). Statistical analysis: one-way ANOVA ($\alpha = 0.05$) with a Dunnett's post-hoc test. Statistically significant differences are indicated (*$p \le 0.05$; **$p \le 0.01$; ***$p \le 0.001$; ****$p \le 0.0001$). Data represent mean ± SEM. Source data are provided as a Source Data file.

fiber tracts at axonal bundle resolution and provides deep insights into tractography and brain connectome. We analyzed five major white matter structures in the *Elp2H206R* brain, namely the hippocampal commissure, anterior commissure, corpus callosum, internal capsule, and afferent and efferent cerebellar tracts. In contrast to brains of wild-type mice (Fig. 5a, c), both commissures and internal capsules in *Elp2H206R* mice revealed significantly reduced fractional anisotropy (Fig. 5b, c). While axial diffusivity was not altered (Fig. 5d), radial diffusivity was increased across all cortical tracts in the *Elp2* mutants relative to control (Fig. 5e). Thus, *Elp2H206R* mice display aberrant

organization and composition of the main cortical tracts in the brain.

Next, we assessed the brain connectivity in the *Elp2* mutants by DT-MRI connectome profiling and compared each connection to identify nodes that were significantly altered in *Elp2H206R* mice compared to their littermate controls. Despite a reduction in the brain volume, the connectome-wise comparisons revealed a substantial hyperconnectivity across long- and short-range connections in the brains of *Elp2* mutant mice (Supplementary Fig. 5a and Supplementary Table 3). This apparently counter-intuitive observation is consistent with previous findings in ASD

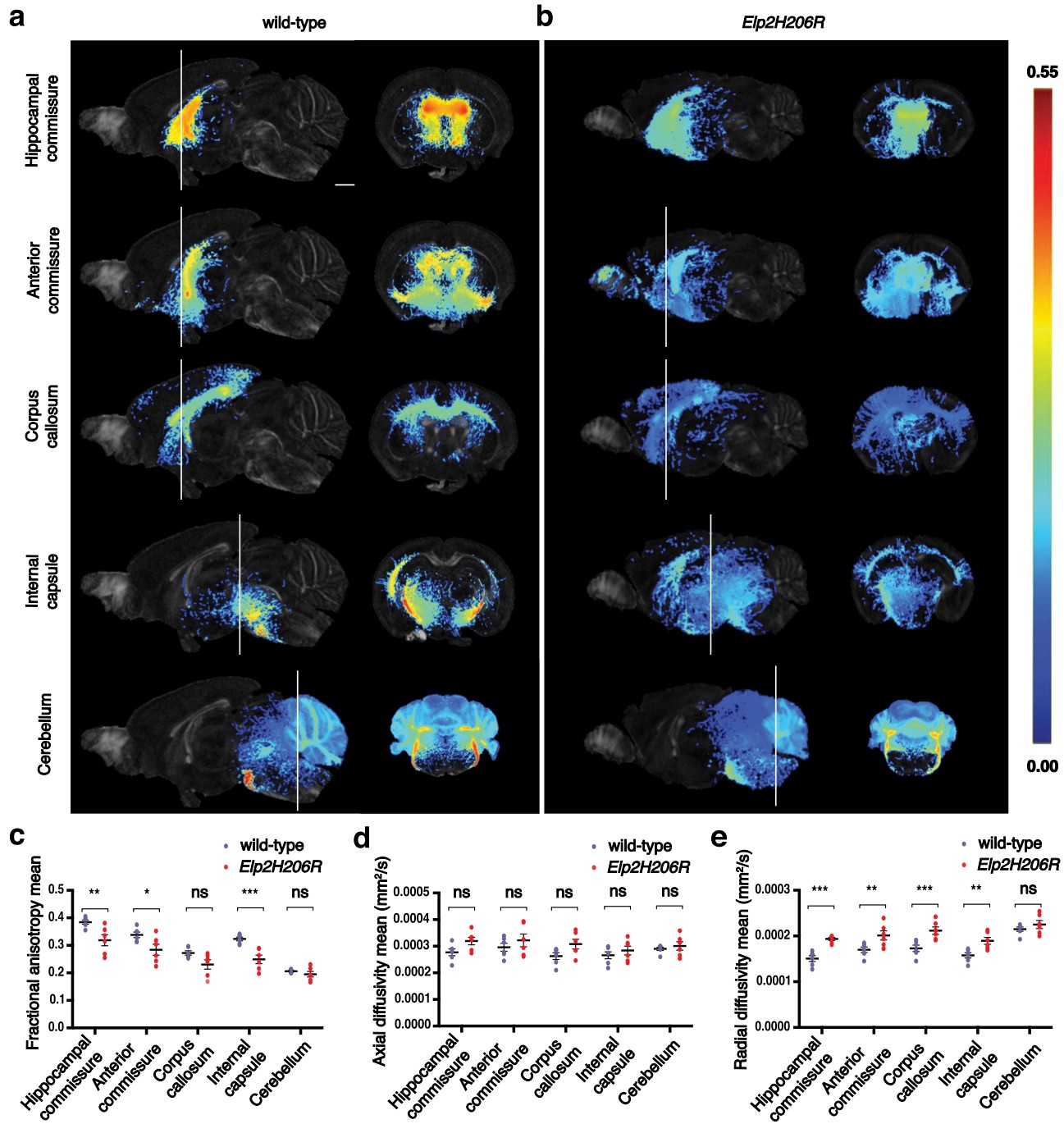

**Fig. 5 Loss of axonal tract integrity among major cortical white matter structures in the brain of *Elp2H206R* mice. a** Representative images of fractional anisotropy maps (measure of microstructural integrity of white matter axonal bundle) of the indicated brain structures in 21–23-days-old wild-type and **b** *Elp2H206R* homozygous mice shown in the mid-sagittal (left panel) and coronal plane (right panel). Scale bar, 1 mm. The fractional anisotropy maps are visualized at threshold from 0 to 0.55 (color scale shown on right). **c** Fractional anisotropy, **d** axial diffusivity (water diffusion parallel to axon bundle), and **e** radial diffusivity (water diffusion perpendicular to axon bundle) are quantified for both genotypes. $n = 6$ animals per genotype. Statistical analysis: unpaired two-tailed *t*-test ($\alpha = 0.05$) with Welch's correction. Holm-Sidak correction was applied to adjust for multiple comparisons. Statistically significant differences are indicated (*$p \leq 0.05$; **$p \leq 0.01$; ***$p \leq 0.001$; ns - not significant: for (**c**) $p = 0.0689$ for corpus callosum and $p = 0.6554$ for cerebellum; for (**d**) $p = 0.2211$ for hippocampal commissure, $p = 0.627$ for anterior commissure, $p = 0.1969$ for corpus callosum, $p = 0.6349$ for internal capsule and $p = 0.6349$ for cerebellum; for (**e**) $p = 0.3235$). Data represent mean ± SEM. Source data are provided as a Source Data file.

patients[51]. The increased connectivity was predominantly noted in a number of structures within the limbic system, the salience network (SN) and the cerebellum. Nodes with significantly increased connections in the limbic system encompassed the ectorhinal, perirhinal, and entorhinal cortices, the cingulum, piriform nucleus, dorsal fornix, and the basal ganglia (caudate putamen and nucleus accumbens). Centers with increased connections within the SN included the insular cortex, nucleus accumbens and thalamus. In contrast, the *Elp2H206R* brains exhibited significantly decreased connectivity throughout nodes of the default mode network (DMN) and corpus callosum (Supplementary Fig. 5b and Supplementary Table 4). The corpus

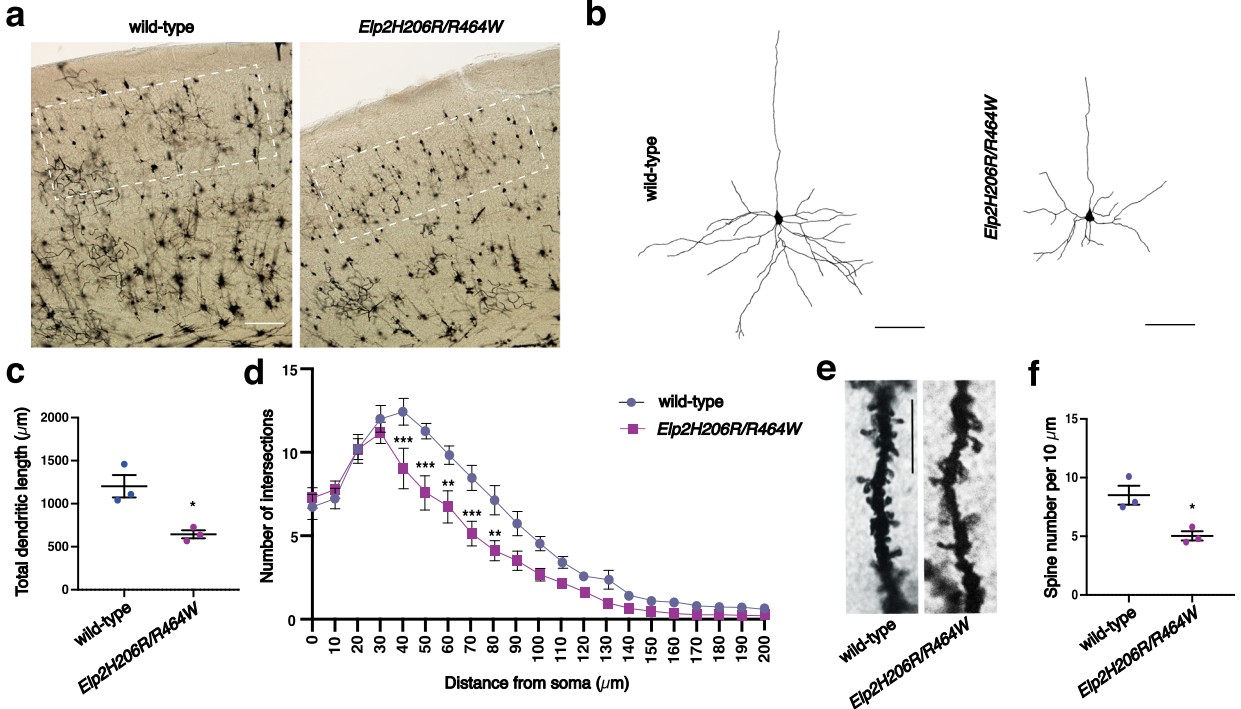

**Fig. 6 Morphological abnormalities of cortical neurons in *Elp2* mutant mice. a** Golgi-Cox-stained coronal sections in the area of the somatosensory cortex of adult (2-months-old) wild-type and *Elp2H206R/R464W* mice. White rectangles highlight pyramidal neurons from cortical layers II and III selected for neuron reconstruction and subsequent analyses. **b** Representative pyramidal neuron reconstructions are depicted (basal dendritic trees were reconstructed). **c** Quantification of total dendritic length from dendritic tree reconstructions shown in (**b**). $n = 15$, 5 neurons per animal and 3 animals per genotype. **d** Sholl analysis of the complexity of basal dendritic arbors. **e** Representative images of dendritic spines on basal dendrites of the cortical neurons shown in (**a**). **f** Spine density per 10 μm was measured on secondary dendrites. For (**d**) and (**f**) $n = 30$, 10 neurons per animal and 3 animals per genotype. Scale bars, 100 μm (**a**), 50 μm (**b**), and 10 μm (**e**). Statistical analysis: unpaired two-tailed *t*-test ($\alpha = 0.05$) with Welch's correction. Holm-Sidak correction was applied to adjust for multiple comparisons (**d**). Statistically significant differences are indicated (*$p \leq 0.05$: for (**c**) $p = 0.0379$ and for (**f**) $p = 0.0328$; **$p \leq 0.01$: for 60 μm $p = 0.002599$ and for 80 μm $p = 0.003315$; ***$p \leq 0.001$: for 40 μm $p = 0.000712$, for 50 μm $p = 0.000172$ and for 70 μm $p = 0.000930$). Data represent mean ± SEM. Source data are provided as a Source Data file.

callosum was severely affected, including its midline and peripheral sections towards the external capsule on both left and right side. Overall, our data indicate that the *Elp2* mutation leads to an abnormal connectivity between some crucial brain regions involved in cognition, social interaction, motor function, and emotional and behavioral response.

### Defective neuronal morphogenesis in *Elp2H206R/R464W* mice.
Alterations in neuronal morphology caused by impaired cytoskeletal function have been previously described in ID[52] and ASD[53]. To investigate whether neuro-morphological defects were present in the *Elp2* mutant mice, we traced Golgi-Cox-stained cortical projection neurons in *Elp2H206R/R464W* and littermate control animals (Fig. 6a). We found a severe decrease in total dendritic length in the *Elp2* mutants (Fig. 6b, c) and subjected the traced neurons to Sholl analysis for the quantification of branching defects. The analysis revealed significantly fewer intersections in concentric circles of increasing radius in the cortical neurons of the *Elp2* mutant mice (Fig. 6b, d). Quantification of spines along dendrites demonstrated a detrimental effect of the *Elp2* mutations on the spine density (Fig. 6e, f).

The observed branching defects are consistent with previous data in loss-of-function Elongator mice[54]. Accordingly, we next analyzed the acetylation status of the tubulin in the *Elp2H206R/R464W* mice by immunostaining. Tubulin acetylation was significantly reduced in the *Elp2* mutants both in the developing brain at 14.5 days post-coitum (dpc; Supplementary Fig. 6a) and the adult brain (Supplementary Fig. 6b), which

was further confirmed by immunoblot analysis (Supplementary Fig. 6c).

These findings indicate that the Elongator complex regulates neuron morphogenesis and dendrite outgrowth and one reflection of this is the key process of tubulin acetylation. Furthermore, the morphological abnormalities in the cortical neurons of the *Elp2* mutant mice are likely to impair the integration of the signals coming from individual synapses, which may be responsible for the observed brain connectome defects.

### Elp2 regulates neural progenitor self-renewal and corticogenesis.
To understand how neurobehavioral, neuroanatomical, and brain connectivity defects arose in the *Elp2* mutant mice, we used immunostaining-based techniques to analyze the developing cortical structures. The expression of Elongator subunits in the developing brain and cortical progenitors has been previously reported[54]. Nonetheless, we reassessed the expression of Elp2 in adult mouse brain structures that are most commonly affected in ID/ASD patients and are known to be crucial for memory consolidation and processing. Elp2 was found to be expressed by principal neurons across all cortical layers (Supplementary Fig. 7a), but also in hippocampal neurons (Supplementary Fig. 7b) and cerebellar granule and Purkinje neurons (PNs; Supplementary Fig. 7c). Hematoxylin and eosin (H&E) staining and analysis of brain sections revealed a reduced thickness of the cerebral cortex in *Elp2H206R* murine embryos (Supplementary Fig. 8a), which persisted in neonatal (Supplementary Fig. 8b) and adult mice (Supplementary Fig. 8c). Next, we assessed the cortical

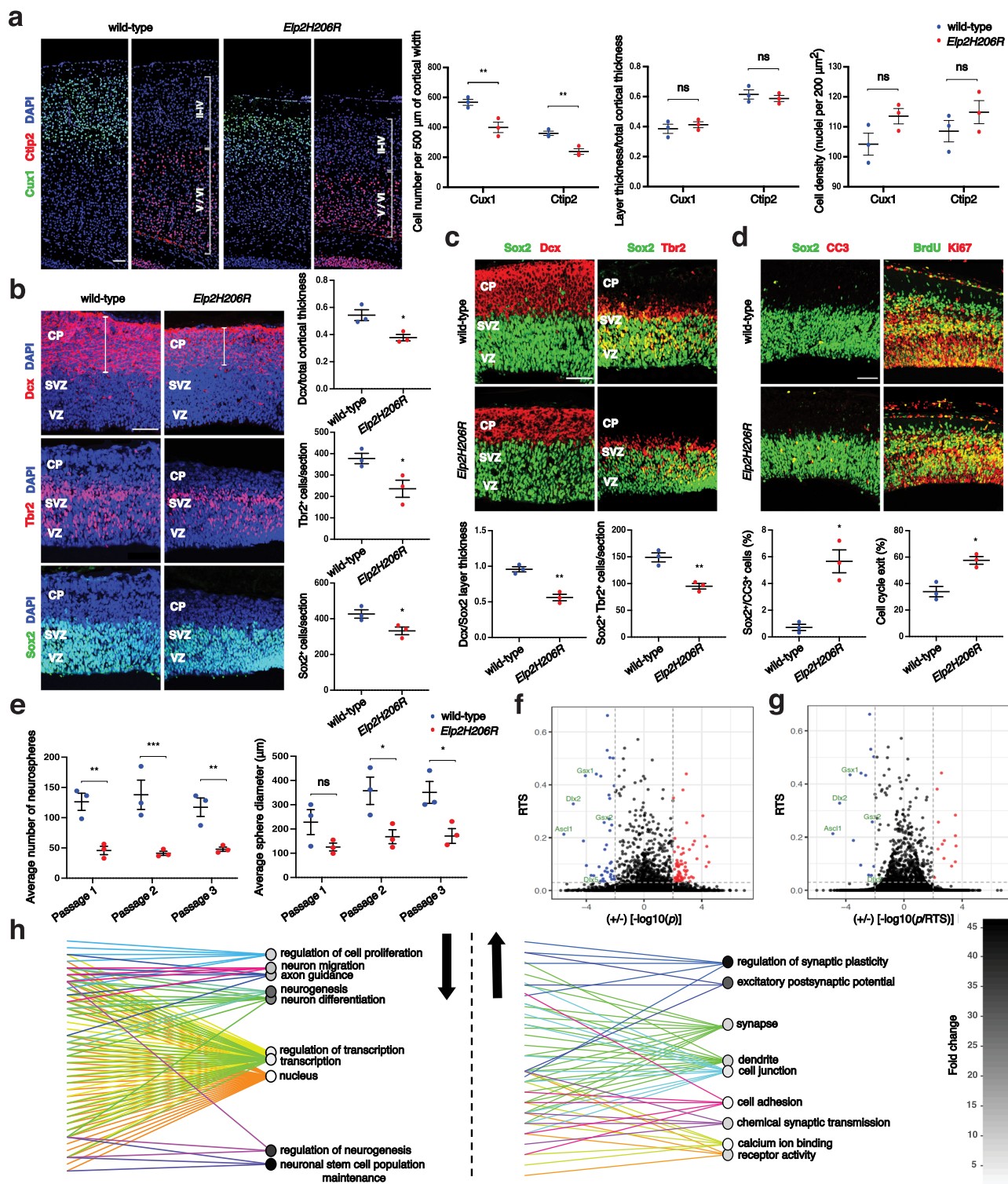

organization in the *Elp2H206R* brains by co-immunofluorescence staining of Cux1 (upper layers II-IV) and Ctip2 (deep layers V and VI). Quantification of the somatosensory cortex revealed a significant and proportional decrease in both upper and deep cortical layers in the *Elp2* mutants in comparison to the littermate controls (Fig. 7a). Thus, the reduction of the cortical thickness results from an overall decrease in neuron number across all cortical layers, while maintaining a normal cellular distribution and laminar organization.

To gain further insights into specific cellular defects underlying microcephaly, we analyzed the *Elp2* mutant brain tissue at the peak of neurogenesis (14.5 dpc)[55]. Immunofluorescence analyses of *Elp2H206R* cortical progenitors showed that apical progenitors (APs; radial glial cells; Sox2+) in the ventricular zone, intermediate progenitors (IPs; Tbr2+) in the sub-ventricular zone, and post-mitotic neurons (Dcx+) comprising the cortical plate were all reduced in number (Fig. 7b). We also noticed a reduction in post-mitotic neurons relative to APs in the mutant

**Fig. 7 Elp2H206R impairs neurogenesis in mice. a** Immunostaining of sagittal brain sections of adult wild-type and Elp2H206R mice, for layer-specific markers Cux1 (upper layers, II-IV) and Ctip2 (lower layers, V and VI). Cux1+ and Ctip2+ cell number, thickness (white lines), and cell density are shown. **b–d** Immunofluorescence staining of coronal cortical sections of 14.5 days post-coitum (dpc) wild-type and Elp2H206R embryos using Dcx (differentiating neurons), Tbr2 (intermediate progenitors) and Sox2 (apical progenitors) markers. Sections were counterstained with DAPI. **b** Dcx thickness (white lines) and Tbr2+ and Sox2+ cell number. **c** Dcx to Sox2 layer ratio and number of newborn intermediate progenitors (Sox2+Tbr2+). **d** Apoptosis (cleaved caspase-3, CC3) of Sox2+ cells and cell cycle exit (percentage of BrdU+Ki67− cells out of total BrdU+ cells). **e** Neurosphere assay of progenitor cells derived from cortices of 14.5 dpc embryos. Average number of neurospheres and sphere diameter are shown in three consecutive passages. For (**a–e**) $n = 3$ animals per genotype and 3 sections per animal (**a–d**); representative images are shown. **f** Differential expression (DE) of genes in the forebrain extracts of Elp2H206R vs. wild-type 14.5 dpc embryos ($n = 6$ per genotype). **g** Transformation of the DE data from (**f**) using TRIAGE. Top downregulated transcripts of genes involved in early neurogenesis are depicted. The vertical dotted lines demarcate $p = 0.01$ in (**f**), and p-value divided by the repressive tendency score (RTS) of 0.01 in (**g**). The horizontal dotted lines show RTS = 0.03 used as a threshold in the previous study[56]. Down- (blue) and upregulated (red) genes in Elp2H206R are shown. **h** Gene ontology (GO) analysis of DE genes showing down- and upregulated (arrows) functionally related genes in Elp2H206R. The color bar denotes fold change of up/downregulated GO terms. Scale bars, 100 μm. Statistical analysis: unpaired two-tailed t-test ($\alpha = 0.05$) with Welch's correction. Holm-Sidak correction was applied to adjust for multiple comparisons (**a** and **e**). Statistically significant differences are indicated (*$p \leq 0.05$; **$p \leq 0.01$; ***$p \leq 0.001$; ns not significant). Data represent mean ± SEM. CP cortical plate, SVZ sub-ventricular zone, VZ ventricular zone. Source data are provided as a Source Data file.

brains and a decreased number of newborn IPs (Sox2+Tbr2+) relative to control (Fig. 7c). This suggested a failure of APs to switch towards neuron generation, which prompted us to investigate potential cell death and cell cycle exit among cortical progenitors. Indeed, cleaved caspase-3 expression confirmed increased cell death in the mutant cortex. A bromodeoxyuridine (BrdU)/Ki67 co-labeling experiment was performed to mark cells that either remained in the cell cycle (BrdU+Ki67+) or exited the cell cycle (BrdU+Ki67-). Remarkably, in a 24-h time window, a greater proportion of neural progenitors exited the cell cycle in the brains of the mutant embryos when compared to control (Fig. 7d). Next, we isolated neural progenitors from the developing cortices and cultured them in vitro in a neurosphere assay. The analysis revealed the formation of significantly fewer spheres at each of the three passages in the brain tissue isolated from Elp2H206R mice (Fig. 7e). The neurospheres were also substantially smaller in cultures derived from the mutant mice, highlighting their reduced self-renewal capacity. Collectively, the data suggest that microcephaly in the Elp2H206R brains is influenced by both the reduced number of neural progenitors and their decreased ability to generate neurons due to their premature cell cycle exit and increased cell death.

To further uncover the molecular consequences of the Elp2H206R mutation in the developing brain, we analyzed the transcriptomes of Elp2H206R and wild-type forebrain structures and compared their gene expression profiles. The differentially expressed (DE) gene analysis identified 536 upregulated and 381 downregulated transcripts in Elp2H206R forebrains (Fig. 7f and Supplementary Data 1). We recognize that many of the DE genes are likely indirectly affected downstream targets and that key regulatory factors are commonly expressed at relatively low abundance. To enrich for the genes that have a strong potential to be cell identity genes, we implemented TRIAGE (transcriptional regulatory inference analysis from gene expression)[56] alongside the DE analysis. Of note, Gsx1, Gsx2, and Ascl1 were found among the most downregulated genes in the Elp2H206R developing brain (Fig. 7g and Supplementary Data 2), suggesting a very early neurogenesis defect originating from the lateral ganglionic eminence (LGE) niche. These transcription factors are expressed in early neural progenitors of the LGE and are essential for their proliferation, maturation[57], cell cycle exit, and neuronal differentiation[58]. Dlx2 and Dlx5 in addition to Ascl1 mark progenitors of cortical interneurons[59,60] and were all found to be significantly reduced in the mutant brains. This reflected in a reduced number of cortical interneurons in the brains of adult Elp2H206R animals that was further confirmed via immunostaining (Supplementary Fig. 9). A gene ontology (GO) analysis

revealed that downregulated genes were mostly associated with gene expression signatures in early neural progenitors, whereas the upregulated transcripts were enriched in differentiating neurons (Fig. 7h). In contrast to previous findings[61], we did not observe the induction of unfolded protein response (UPR) in cortical progenitors using patient-derived germline mutations (Supplementary Fig. 10).

Altogether, our findings demonstrate that Elp2H206R mutation induces cell cycle arrest in early cortical progenitors at the peak of neurogenesis, leading to an attenuation of the complete neurogenic program.

**Elp2H206R and Elp2R464W impair tRNA modification and protein homeostasis in the developing mouse brain.** The Elongator complex acts as a translational regulator[30,31] and a range of cellular activities, including tubulin acetylation, are most likely controlled indirectly via its genuine tRNA modification activity. Following from our data that hElp2H206R and hElp2R462W have a detrimental effect on the stability and activity of the Elongator complex in vitro, we further investigated the consequence of these mutations on tRNA modification levels in the Elp2H206R and Elp2H206R/R464W mice. The abundance of both cm5U-dependent modifications, ncm5U and mcm5U, as well as further modified mcm5s2U, was diminished in the developing brain of Elp2 mutants in comparison to the wild-type controls (Fig. 8a–c), whilst no significant changes were detected in the levels of Elongator-independent tRNA modifications (Fig. 8d).

In agreement with our previous study on ataxic mice carrying a mutation in the Elp6[26], we observed cerebellar atrophy and massive PN degeneration in the cerebella of Elp2H206R and Elp2H206R/R464W mice (Supplementary Fig. 11a–c). We then asked whether the observed tRNA modification defects in the Elp2 mutants induce UPR leading to neurodegeneration. Indeed, the expression of ER stress-induced transcription factor CHOP and elevated levels of ubiquitination and activated caspase-3 in the degenerating neurons (Supplementary Fig. 11d, e), suggest UPR-mediated apoptosis as the most likely route of neuronal death. The structural changes in the white matter tracts of the Elp2H206R animals identified by DT-MRI led us to further assess whether similar molecular mechanisms underlie myelin loss. First, we confirmed by immunofluorescence that Elp2 was expressed by mature oligodendrocytes (APC+ cells; Supplementary Fig. 12a) and that there was a severe reduction in myelin basic protein (MBP) in the Elp2 mutants (Supplementary Fig. 12b). Staining for the oligodendrocyte precursors (PDGFRα+ cells) and APC expression revealed that only mature cells were reduced in number, whilst the precursors were unaffected (Supplementary Fig. 12c). Ubiquitination and markers of ER stress

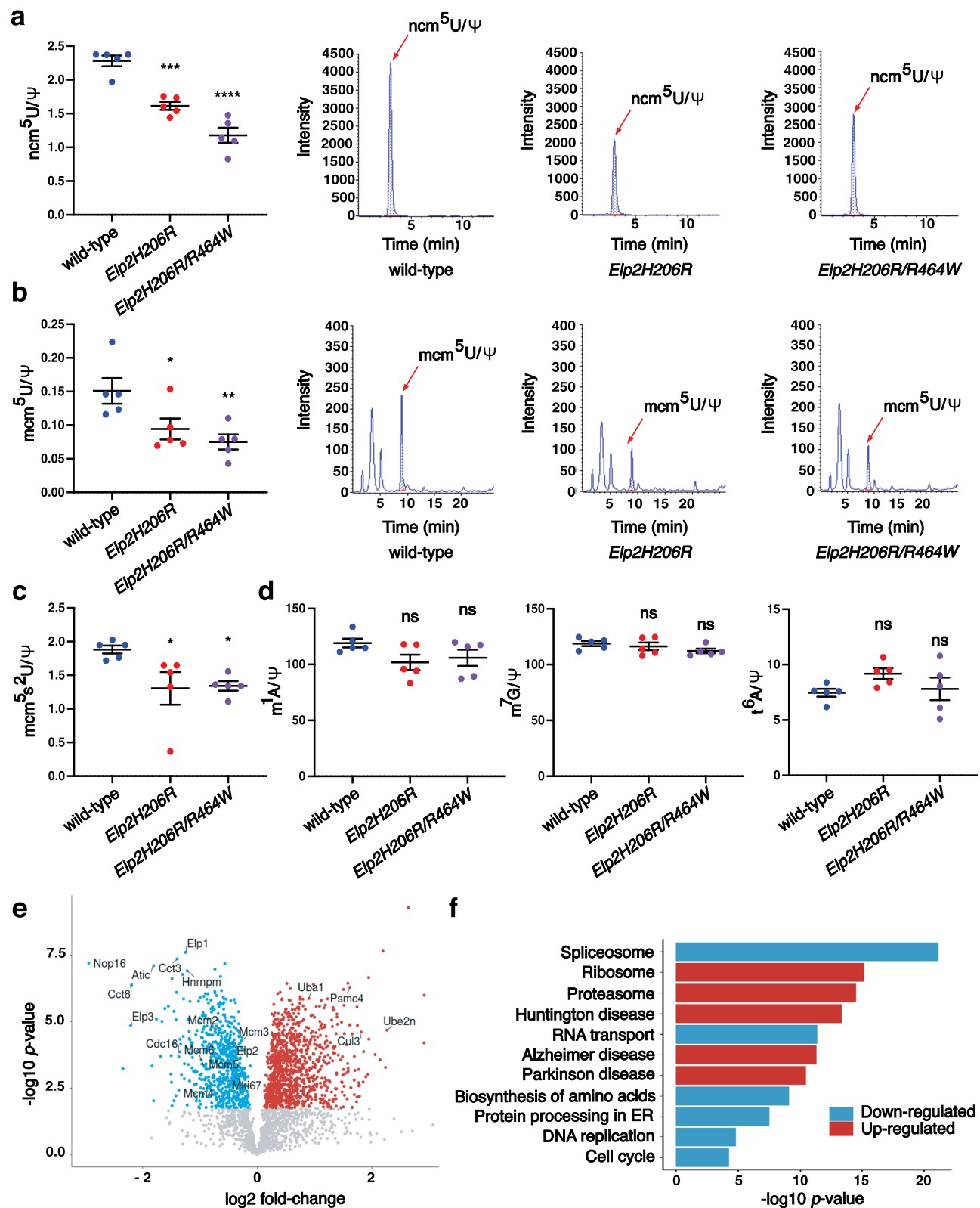

and apoptosis were found to be upregulated in mature oligodendrocytes (Supplementary Fig. 12d, e). Thus, perturbation of the function of Elongator not only deregulates the neurogenic developmental program but seems to also induce UPR in mature neurons and oligodendrocytes, which promotes neurodegeneration and myelin loss respectively.

We further assessed the consequences of the reduced tRNA modification levels on the protein expression profiles using mass spectrometry. Differential proteomic analysis of the developing cortical tissue from *Elp2* mutant and control mice identified 1159 up- and 828 downregulated proteins (Fig. 8e and Supplementary Data 3). Elp2 expression levels were reduced in the *Elp2H206R*

**Fig. 8 tRNA modification deficiency and proteostasis defects in developing brains of the *Elp2* mutant mice. a–d** High-performance liquid chromatography coupled to mass spectrometry was used to quantify the Elongator-dependent tRNA modifications 5-carbamoylmethyluridine (ncm⁵U) (**a**), 5-methoxy-carbonylmethyluridine (mcm⁵U) (**b**), 5-methoxycarbonylmethyl-2-thiouridine (mcm⁵s²U) (**c**) and Elongator-independent tRNA modifications 1-methyladenosine (m¹A), 7-methylguanosine (m⁷G) and threonylcarbamoyladenosine (t⁶A) (**d**) in 14.5 days post-coitum (dpc) *Elp2H206R* and *Elp2H206R/R464W* relative to wild-type mice. Pseudouridine (Ψ) was used as an internal normalization standard. Representative chromatograms for ncm⁵U (**a**) and mcm⁵U (**b**) are shown for each genotype. For (**a–d**) $n = 5$ animals per genotype. **e** Differential expression (DE) of peptides in the cortical extracts of *Elp2H206R* vs. wild-type 14.5 dpc embryos ($n = 3$ per genotype). Down- (blue) and upregulated (red) peptides are depicted on the plots. **f** KEGG pathway analysis of DE peptides showing functionally related peptides that are down- and upregulated in the *Elp2H206R* developing cortex. Statistical analysis: one-way ANOVA ($\alpha = 0.05$) with a Dunnett's post-hoc test. Statistically significant differences are indicated (*$p \leq 0.05$; **$p \leq 0.01$; ***$p \leq 0.001$; ****$p \leq 0.0001$; ns not significant: for m¹A $p = 0.1253$ for *Elp2H206R* and $p = 0.2647$ for *Elp2H206R/R464W*; for m⁷G $p = 0.7776$ for *Elp2H206R* and $p = 0.2012$ for *Elp2H206R/R464W*; for t⁶A $p = 0.1689$ for *Elp2H206R* and $p = 0.9094$ for *Elp2H206R/R464W*). Data represent mean ± SEM. Source data are provided as a Source Data file.

brain, along with Elp1 and Elp3, being among the most significantly downregulated proteins. This was concordant with western blot analysis of the brain extracts revealing lower expression of the Elp123 sub-complex in both *Elp2H206R* and *Elp2H206R/R464W* mice relative to control, whilst Elp456 sub-complex did not seem to be affected based on the Elp4 levels (Supplementary Fig. 13). KEGG pathway enrichment analysis showed molecular pathways involved in protein synthesis to be the most significantly downregulated in the *Elp2* mutant samples, whilst pathways implicated in protein misfolding and degradation were found amongst the most significantly enriched categories (Fig. 8f). In accordance with the previous study identifying long, AA-biased genes as specific Elongator targets in peripheral neurons[62], we found that the proteome of the developing cortex in the *Elp2* mutants was biased towards shorter proteins (Supplementary Fig. 14a). Remarkably, downregulated proteins in the *Elp2H206R* cortex were enriched in AA-ending codons, whilst upregulated proteins were biased towards the AG-ending codons (Supplementary Fig. 14b). Taken together, our data suggest that the *Elp2* mutations perturb cell proliferation and protein homeostasis in the developing brain via loss of Elongator-dependent tRNA modifications, which likely impairs the translation of large AA-biased transcripts and globally changes the proteomic profile.

## Discussion

Next-generation sequencing studies have suggested rare *ELP2* variants to be associated with ID[19–21], which led us to identify a number of individuals with *ELP2* variants and to define the role of this gene in the neurodevelopment and associated pathologies.

Here we show that not only does the murine *Elp2* closely resemble its human ortholog, but the variations in sequence as identified in the affected individuals induce similar effects in purified mouse and human Elp2 proteins. The phenotype of the mouse models with either homozygous *Elp2H206R* or *Elp2H206R/R464W* variants, recapitulates the majority of cardinal features of the affected individuals. This includes growth and vocalization impairment, developmental delay, hypotonia, stereotypic behavior, and motor defects. Our data provide evidence that the *Elp2* gene mutations are the primary genetic driver of the severe clinical phenotype.

During the course of our investigations, we have successfully produced and tested the recombinant functional human ELP123 complex. This presented us with a unique opportunity to assess the influence of individual *ELP2* patient variants on the stability, assembly, and activity of the ELP123 complex. The recombinant, purified ELP123 complexes carry the variants in both of their *ELP2* copies, which creates a slightly different situation than what we see in the compound heterozygous patients. The two ELP2 molecules are located far apart from each other in the assembled complex and we do not expect that the two proteins directly

communicate with each other. As the level of the reduced activity is almost identical for all variants, we conclude that various combinations of *ELP2* mutations in the same ELP123 complex would have similar additive effects. As expected, we do not observe completely diminished activity of the mutated complexes in vitro and in vivo, creating a situation where the tRNA modification levels are reduced to a level that permits survival, but leads to severe phenotypic implications.

Neuroimaging of the *Elp2H206R* murine brains revealed a volume reduction and aberrant microstructural integrity of the main cortical tracts predominantly in the brain structures that regulate information processing, memory, social interaction and motor skills. This is consistent with previous DT-MRI studies in ID/ASD patients that demonstrated abnormal changes in cortical tracts with reduced fractional anisotropy and greater diffusivity[63]. Tractography defects were observed in the corpus callosum of *Elp2H206R* animals, which is in accordance with the findings described in patients with ASD, ID, speech impairment and seizures[64–66]. Our data add to the growing evidence that disruption of white matter tracts in ASD might represent one of the core pathologies of the disease.

The connectome-wise comparisons revealed that the mutant brains had significantly increased connections in the limbic system, the brain structure functionally involved in the emotional and behavioral response. Abnormalities across many limbic structures have been reported for ASD[67,68]. Aberrant basal ganglia connectivity observed in the *Elp2H206R* mice likely contributes to the repetitive and stereotyped behaviors of the patients with ASD[69]. Based on the previously described SN connectivity profile in ASD[70,71], we can attribute the increased connection in the SN found in the *Elp2* mutant mice to sensory over-responsivity, including aggressivity and self-injury observed in the patients. Decreased connectivity of the DMN found in the *Elp2H206R* mice has been increasingly implicated in social dysfunction in ASD[72]. Further structural MRI and connectome assessment in the patients with *ELP2* mutations is necessary to confirm our results in mice and look at the specificity of fiber tracking in the early stages of ASD when there is diagnostic uncertainty.

Structural MRI and histological analyses identified microcephaly in the *Elp2H206R* mouse brain, which parallels the clinical phenotype and is consistent with the reduced brain size observed in mice upon a conditional loss of Elp1[28] and Elp3[61]. Conditional deletion of the Elp1 subunit in mouse cortical progenitors leads to microcephaly by causing a reduction in IP rather than AP number, resulting in suppressed indirect neurogenesis[61]. In contrast, our study indicates that neurogenesis defects arise early at the level of APs and cycling progenitors of the LGE in the case of *Elp2* genomic mutations. Although the IPs are also affected in *Elp2H206R* mice, this seems to be a consequence of the AP reduced number rather than the fate decision of the dividing

cells. Nevertheless, further research is warranted to explore direct versus indirect neurogenesis program in the developing brain of the *Elp2* mutants. We show that smaller cerebral cortex in these mice results from an overall reduction of the neuronal layers caused by premature depletion of the progenitor pool due to cell-cycle arrest and increased cell death. Neurogenesis defects extend to interneurons in addition to cortical projection neurons. Their impaired development is likely a consequence of cytoskeleton defects affecting their migration and branching as previously reported[73]. However, as the previous studies included a conditional KO approach, it is important to perform similar analyses using mouse models of clinically relevant germline mutations in the Elongator subunits to confirm these findings.

Despite up-regulation of the ubiquitin-proteasome pathway in proliferating neural progenitors of *Elp2H206R* mice, the progenitors seem to avoid the UPR and ER stress-triggered death, in contrast to PNs and oligodendroglia. This raises the intriguing question whether UPR-induced cell death selectively affects PNs and oligodendrocytes in the brain. It is also interesting that PN development seems not to be affected by the *Elp2* mutations, but this is a degenerative process, similar to what we have previously found in the *Elp6* mutant mice[26]. Future studies need to be directed towards the analyses of single-cell types and their specialized proteomes.

Our study has shown a critical link between translation kinetics and the pathology of a monogenic complex neurodevelopmental phenotype that includes ID and ASD. We have demonstrated that key stages of brain development are coordinated by the tRNA modification activity of the Elongator complex, which is essential for maintaining protein homeostasis in the developing cortex and post-mitotic brain cells. As previously proposed[62], this activity seems to be specifically important for the translation of long transcripts, which are biased towards the use of AA-ending codons. Our data indicate that the human (and mouse) brain is particularly sensitive to tRNA modification deficiency leading to perturbed proteostasis and consequent defects in the proper wiring and connectivity within the brain necessary for cognition, memory and adaptive behavior. Using patient-derived germline mutations, we have directly shown the central role of the Elongator tRNA modifying complex in the developing and homeostatic brain in both humans and mice.

## Methods

**Patient ascertainment and phenotyping**. Patients with homozygous or compound heterozygous variants in the *ELP2* gene were recruited from research or diagnostic programs in Europe, Australia, and the US. Grouping of clinical data for this series was facilitated by GeneMatcher[74]. All patients were clinically evaluated by neurologists and clinical geneticists at their respective tertiary healthcare centers, and clinical information was collected from medical files in combination with standardized questionnaires and face-to-face interviews with patients and their families. Seizures were classified according to the International League Against Epilepsy classification[75]. Where available, MRI images were reviewed by an expert neuroradiologist (C.G.M.). Written informed consent was obtained from all participants. The study was approved by the Ethical Committee of Regional Zealand (project license number SJ-91), Melbourne Health Human Research Ethics Committee (project license number HREC/16/MH251), Colorado Multiple Institutional Review Board (project license number 19-0751), and Johns Hopkins University School of Medicine Institutional Review Board (project license number NA_00045735).

**DNA constructs**. Genes encoding Elp2 from *Homo sapiens* (*hElp2*) and *Mus musculus* (*mElp2*) were cloned into pFastBac1 HTa plasmid with 6 × His tag sequence localized on the 5′ side of the inserted gene. *hElp123* construct was generated using Gibson assembly method. In detail, *hElp1* gene was cloned into pFastBac1, *hElp2* into pFastBac1 HTa with the 6 × His tag sequence and *hElp3* into pFastBac1 with an additional Twin-Strep-tag sequence on the 3′ side of the gene. All three genes were amplified in PCR with primers adding overhangs on 5′ and 3′ sides of each gene and then assembled within pBig1a plasmid using standard protocol and primers[42]. Mutations in *hElp2*, *mElp2*, and *hElp123* were introduced by QuickChange mutagenesis. A list of primers used for mutagenesis is provided in Supplementary Table 5. All genetic constructs were moved to insect cells using standard Bac-to-Bac protocols.

**Recombinant protein production and purification**. For protein expression, Super Sf9-1 cells were infected with MOI = 1 and grown for 3 days at 27 °C on a shaking platform. Subsequently, insect cells were lysed in Lysis Buffer (for hElp2 and mElp2: 50 mM HEPES, 300 mM NaCl, 2 mM MgCl2, 2 mM 2-Mercaptoethanol, 5% glycerol, 10 mM imidazole, DNase I, protease inhibitors; for hElp123: 50 mM HEPES, 100 mM NaCl, 2 mM MgCl2, 2 mM DTT, 5% glycerol, DNase I, protease inhibitors) by 3 rounds of freezing and thawing in liquid nitrogen and sonication, followed by centrifugation (4 °C, 1 h, 80,000 rcf). hElp2 and mElp2 supernatants were purified on NiNTA agarose beads (Qiagen) followed by overnight tag cleavage in Dialysis Buffer (50 mM HEPES, 300 mM NaCl, 2 mM 2-Mercaptoethanol, 10 mM imidazole, pH 8.0). On the next day, the protein sample was rebound to NiNTA beads and flow through was run on S200 Increase 10/300 GL column (GE Healthcare) in Final Buffer (20 mM HEPES, 300 mM NaCl, 5 mM DTT, pH 8.0). hElp123 variants were purified using StrepTrap HP 5 ml column (GE Healthcare) eluted in Strep Elution Buffer (50 mM HEPES, 100 mM NaCl, 1 mM DTT, 5% Glycerol, 2.5 mM d-desthiobiotin, pH 8.0), followed by affinity chromatography on HiTrap Heparin HP 5 ml column (GE Healthcare) eluted in a gradient of Heparin Elution Buffer (50 mM HEPES, 1000 mM KCl, 1 mM DTT, pH 8.0). Lastly, eluates were run on Superose 6 Increase 10/300 GL column (GE Healthcare) in Final Buffer (20 mM HEPES, 150 mM NaCl, 5 mM DTT, pH 8.0).

**Structure modeling**. hElp1, hElp2, and hElp3 protein structure models were predicted by homology to previously solved structures with Phyre2 server[76]. hElp123 structure was obtained by aligning resultative structures of human proteins to yeast Elp123 Cryo-EM structure (PDB 6QK7) in PyMOL[77].

**Thermal shift assay**. Protein samples were concentrated to 0.65 g/l (hElp2) and 0.3 g/l (mElp2) in 20 mM HEPES, 300 mM NaCl, 5 mM DTT, pH 8.0 and mixed with 25 × SYPRO Orange (ThermoFisher) on a 96-well qPCR plate. CFX96 Real-Time System C1000 Touch Thermal Cycler (Biorad) was used to gradually heat samples from 4 to 98 °C with a heating rate of 0.2 °C/10 s. The fluorescence intensity was measured at probe-specific excitation (470 nm) and emission (570 nm) wavelengths.

**Microscale thermophoresis**. Experiments were conducted using a constant 100 nM concentration of Cy5-labled tRNA$^{Arg}_{UCU}$ or tRNA$^{Arg}_{ACG}$ in MST Buffer (20 mM HEPES, 150 mM NaCl, 5 mM DTT, pH 8.0, 0.0125% Tween 20) and incubated with 16 serial dilutions of hElp123$_{WT}$ (starting from 12 μM). Measurements were performed at 100% excitation power in Premium Coated capillaries on the Monolith NT.115 (Nanotemper Technologies) at 25 °C. Obtained data were analyzed and dissociation constant values were calculated using MO. Affinity software (Nanotemper Technologies).

**Acetyl-CoA hydrolysis activity assay**. Each of the hElp123 variants in concentration of 0.15 μM was mixed with 2 μM tRNA and 100 μM ACO and incubated for 3 h at 37 °C. Next, the samples were passed through a 3 kDa cutoff concentrator (EMD Millipore). Flow-through was collected and pipetted into a 96-well plate. ACO quantity in each sample was determined with Acetyl-CoA Assay Kit (Sigma) according to the manufacturer's instructions. Fluorescence intensity was measured on a plate reader (TECAN) at probe-specific excitation (535 nm) and emission (587 nm) wavelengths.

**Animals and genotyping**. All experimental procedures on mice were approved by the University of Queensland Molecular Biosciences Ethics Committee (project license numbers IMB/533/17 and IMB/001/18). Mice were housed under a 12-h light/dark cycle in a barrier facility with food and water provided ad libitum. All mice were housed at room ambient temperature (22 ± 2 °C) with relative humidity (30–60%). Animals were initially maintained on C57BL/6 genetic background and transitioned mid-experimentation onto the DBA2/J background. No gender-related phenotypic differences were observed in all mice strains. *Elp2* knock-out (KO) and heterozygous *Elp2H206R* and *Elp2R464W* mice were generated via CRISPR/Cas9-mediated gene editing using the gRNA sequence 5′-TCTTTCTCTCTGCGGA CACG-3′ for introducing the *Elp2*KO and *Elp2H206R* mutations, and 5′-AAATTTTCCACAAAATTCCG-3′ for the *Elp2R464W* mutation.

Genomic DNA for genotyping purposes was obtained from tail tips using QuickExtract DNAextraction solution (Epicenter) as per the manufacturer's instructions. Animals were genotyped via restriction digestion of PCR-amplified genomic material, followed by agarose gel electrophoresis for visualization of DNA bands of different size. A list of primers used for genotyping the *Elp2* mutant mice is provided in Supplementary Table 5. Cycling conditions for genotyping *Elp2H206R* and *Elp2*KO mice were as follows: 94 °C for 2 min, 35 cycles of denaturation at 94 °C for 20 s, annealing at 67 °C for 20 s and extension at 72 °C for 45 s, followed by final extension at 72 °C for 5 min. Amplified product of 520 bp was digested by using EagI restriction enzyme. For genotyping *Elp2R464W* animals, we used the following cycling conditions: 94 °C for 2 min, 35 cycles of

denaturation at 94 °C for 20 s, annealing at 61 °C for 20 s and extension at 72 °C for 60 s, followed by final extension of 72 °C for 5 min. This yielded a PCR product of 929 bp that was digested by AvrII enzyme. We sequenced the PCR-amplified products using the Sanger sequencing method to further confirm the genotypes.

**Behavioral testing and analysis**. The behavioral analysis of mouse phenotype was performed according to the modified SHIRPA screen (v1.04: 2002.3)[78] at P21 (C57BL/6 mice) or between P50 and P60 (DBA2/J mice). Each mouse was observed in a transparent cylindrical viewing jar (15 cm diameter) for 5 min, and then transferred to an arena (30 cm × 30 cm) that consisted of nine evenly sized squares (10 cm × 10 cm). This was followed by a series of anatomical and neurological measures[79]. Limb clasping was assessed on a scale 0–3 per scoring system previously published[80].

For the grooming analysis, each mouse was recorded in its home cage for 30 min for cumulative time spent grooming all body regions and the absolute number of bouts.

To induce ultrasonic vocalization (USV) in pups, they were isolated from their nest at P8. Pups were removed individually and placed into an isolation container made of Styrofoam (260 mm × 180 mm × 185 mm) to act as a sound attenuation box studio. Pups were placed on a preheated heating pad at 35 °C for the duration of testing. USV emission was recorded by an ultrasonic microphone (Ultramic 250 K, Dodotronic) attached to the lid of the sound-attenuating box, 8 cm above the arena floor, and connected via USB port to a laptop. Acoustic data were recorded for 5 min using Audacity v2.1.3 (available at http://audacityteam.org/; the name Audacity is a registered trademark of Dominic Mazzoni) software with a sampling rate of 250,000 Hz. For acoustical analysis, USV recordings were transferred to Raven Pro v1.5 (Cornell Laboratory of Ornithology) software. Total number of calls, call duration, and peak frequency, and amplitude were calculated for a 5 min recording session.

All behavioral analyses were performed by researchers blinded to the genotype of the animals.

**Tissue and embryo collection**. Experimental animals were anaesthetized using Zoletil (50 mg/kg, i.p.) and Dormitor (1 mg/kg, i.p.) and transcardially perfused with phosphate-buffered saline (PBS), followed by 4% paraformaldehyde (PFA) solution. Following dissection, brain tissue was drop-fixed in 4% PFA at 4 °C for 12 h under constant agitation. The tissue was then washed twice with PBS and left in PBS overnight. Brains were further processed for either MRI or histological analysis. For histological studies, brains were processed in the tissue processor (Leica TP1020) over 15 h as per the user's guide, subsequently embedded in paraffin and sectioned at 8 μm in the sagittal plain using microtome (Leica RM2235). Sections were transferred to glass slides and dried overnight at 45 °C.

Embryo collection was carried out on the mornings of the designated days between 9 dpc and 18.5 dpc. Pregnant dams were injected intraperitoneally with BrdU labeling reagent (1 ml per 100 g body weight; Invitrogen). Twenty-four hours after BrdU injection, the animals were euthanized by carbon dioxide inhalation. Using a dissection microscope, individual embryos were isolated from the uterus and their heads were harvested. The tissue was drop-fixed in 4% PFA for 2–6 h at 4 °C, subsequently washed in PBS twice and left in PBS overnight. Heads of embryos were cryo-embedded and coronally sectioned at 10 μm using a cryostat (Leica CM3050 S). Sections were transferred onto glass slides and stored at −80 °C.

**MRI and analyses**. Volumetric and diffusion MRI analyses were performed on brains obtained from P21–23 C57BL/6J mice using 4% PFA transcardial perfusion. Prior to MRI, the brains were washed with 0.2% (v/v) gadopentetate dimeglumine (Magnevist; Bayer, Leverkusen, Germany) in 0.1 M PBS for 4 days to enhance MRI contrast[81]. MRI was performed via 16.4 T vertical bore microimaging system (Bruker Biospin, Rheinstetten, Germany; ParaVision v6.01) with 15 mm linear surface acoustic wave coil (M2M, Brisbane, Australia) and Micro2.5 imaging gradient.

Three-dimensional (3D) T1/T2*-weighted fast low angle shot structural images were acquired using gradient echo imaging with parameters as follows: repetition time (TR) = 50 ms, echo time (TE) = 12 ms, bandwidth = 50 kHz, field of view (FOV) = 19.6 × 11.4 × 8.4 mm, matrix size = 654 × 380 × 280. The resulting isotropic image resolution was 30 μm with an acquisition time of 30 min per brain.

Three-dimensional diffusion-weighted image (DWI) data were acquired with a Stejskal-Tanner[82] DWI spin-echo sequence using the following parameters: TR = 200 ms, TE = 23 ms, $\delta/\Delta$ = 2.5/12 ms (where δ is the duration of the half-sine diffusion pulse and Δ is the time between pulses), bandwidth = 50 kHz, FOV = 19.6 × 11.4 × 8.4 mm, matrix size = 196 × 114 × 84, image resolution = 100 μm, 30 direction diffusion encoding with $b$-value = 5000 s/mm$^2$ and two $b$ = 0 images with an acquisition time of 17 h per brain. DWI datasets were zero-filled by a factor of 1.5 in all dimensions and Fourier transformed for improved fiber tracking[83] to 80 μm isotropic resolution.

Advanced normalization tools (ANTS) affine and diffeomorphic registrations were used to create separate three-dimensional templates for the wild-type and Elp2H206R animal groups. The Brookhaven National Laboratory 3D MRI C57BL/6J adult mouse brain atlas[84] was then registered to each sample and normalised back into the original sample space using ANTS (inverse warp and affine

transformation). The atlas-registered samples were segmented into 20 distinct brain regions and ITK-SNAP software enabled the acquisition of volumetric statistics for each region.

For the tractography and structural connectome analyses, DWI datasets were first intensity-bias corrected with ANTS N4BiasFieldCorrection. DWIs were subsequently processed using MRtrix3 software for fiber orientation distribution modeling under the constrained spherical deconvolution method. Region-specific tractography was performed for the hippocampal commissure, anterior commissure, corpus callosum, internal capsule, and cerebellum. Regions of interest (ROIs) were manually seeded from mid-sagittal and coronal sections for all tracts except the cerebellum, and fiber tracts were generated using the iFOD2 algorithm[85] at 100 seeds per voxel before threshold masking all tracts at 10%. Cerebellar tracts were seeded from the entire cerebellum segmentation assigned during volumetric atlas registration, and fiber tracking was executed at 2 seeds per voxel. For structural connectome mapping, whole-brain tractography was generated at 10 seeds per voxel and segmented according to the modified Centre for Advanced Imaging (CAI) - John Hopkins University MRI atlas[86]. Like all connectivity networks, a structural connectome is comprised of hubs, called nodes, joined by a series of connections, called edges. Nodes were defined according to the ROIs assigned from atlas registration, while edges were computed via probabilistic tractography; a statistical technique whereby tracts are generated between all pairs of nodes with one used as a seed and the other as a target and vice versa. Connectivity between nodes was calculated as the number of streamlines successfully reaching the target node from the streamlines leaving the seed node. The resulting symmetrical connectivity matrix for each brain was comprised of 106 nodes and the Network-Based Statistic (NBS) Toolbox[87] enabled connection-wise comparison of wild-type and Elp2H206R matrices. The extent component (total number of connections) was used for NBS analyses under a range of primary thresholds ($t$ = 2.5–3.5) such that false positives were minimised[86]. The final connectivity comparisons were presented using the primary threshold with the highest statistical significance (lowest family wise error rate corrected $p$-value); $t$ = 3.2 and $t$ = 3.4 for increased and decreased network connectivity respectively.

**H&E staining, imaging, and analyses**. Following deparaffinization, brain tissue was stained in Haematoxylin (Sigma Aldrich) for 3 min and the excess of the stain was removed by quick immersion of slides in 1% HCl solution, followed by another short immersion of slides in 0.1% LiCO₃ solution. Next, slides were stained with Eosin Y solution (Sigma Aldrich) for 30 s and dehydrated in a series of ethanol dilutions (70, 90, and 100% ethanol for 30 s each) and xylene for 10 min. Slides were mounted with Entellan mounting medium (ProSciTech) and presented for 2 h at room temperature. Images were obtained using Olympus BX-51 upright brightfield microscope. Whole-brain images were acquired as tiled image stacks. Cortical layers thickness relative to section thickness and cerebellar area relative to whole-brain area were measured using ImageJ software[88]. Measurements of length and area are indicated by lines on the respective figures.

**Immunofluorescence staining, imaging, and analyses**. Paraffin-embedded tissue sections underwent deparaffinization, whilst cryo-embedded sections were thawed at room temperature for 20 min and then hydrated in PBS for 10 min. Heat-induced antigen retrieval was performed using citrate buffer-based unmasking solution (Abacus) at 100 °C for 10 min. For unspecific binding of antibodies, M.O.M. blocking reagent (Vector) was used for primary antibodies raised in mouse, and 4% horse serum (Life Technologies) for the antibodies raised in species other than mouse. Slides were incubated with following primary antibodies: Acetyl-α-tubulin (1:100; T6793), APC (1:100; ab72040), BrdU (1:100; ab6326), CC3 (1:50; ab2302), CHOP (1:100; ab11419), Ctip2 (1:100; ab18465), Cux1 (1:100; ab54583), Dcx (1:250; ab2253), Elp2 (1:100; ab154643), Ki67 (1:100; ab15580), MBP (1:100; ab40390), NeuN (1:100; MAB377), Parvalbumin (1:200; ab11427), Pcp2 (1:100; sc-49072), PDGFRα (1:100; AF1062), Sox2 (1:200; ab97959; 53-9811-82), Tbr2 (1:200; ab23345), Ub (1:100; ab7780), followed by incubation with an AF488, AF594 or AF647-labeled donkey anti-mouse, anti-rat, anti-rabbit, anti-guinea pig or anti-goat antibody (1:250; Invitrogen), and counterstained with DAPI (Sigma Aldrich). Images were captured using Zeiss LSM 710 upright confocal microscope as Z-stacks and presented as the sum of the Z-projection. Whole-cerebellum images were acquired as tiled image stacks.

Image processing and quantification of cortical thickness, cell number, cell density, MBP, and acetyl-α-tubulin mean pixel intensity were performed by using ImageJ software[88]. For the relative thickness of cortical layers, the thickness of the Cux1⁺ and Ctip2⁺ layers were measured relative to the overall cortical thickness (from the top cortical layer to the white matter; layers I–VI). Length measurements are indicated by lines on the respective figures. Sizes of the sections on which cell number and cell density quantifications were performed are specified in figures.

**Neurosphere assay**. Brains of 14.5 dpc mice were isolated from the skull and dissected manually on the coronal plane to expose the lateral ventricles at the level of the corpus callosum. The cortical tissue was carefully removed using forceps, then cut into fine pieces before enzymatic digestion was performed by incubation in 0.05% trypsin at 37 °C for 15 min to obtain a single-cell suspension. Cells were dissociated and resuspended in 5 ml of Neurocult Basal Medium Mouse/Rat

(Stemcell), supplemented with 10% Neurocult Basal Proliferation Supplement (Stemcell), 20 ng/ml epidermal growth factor (Preprotech), 10 ng/ml basic fibroblast growth factor (Preprotech), 2 μg/ml heparin and 0.5% penicillin and streptomycin. The cell suspension was then run through a 50 μm filter. Cells were cultured at 37 °C and 5% $CO_2$. The cells were plated at a concentration of $2.5 \times 10^5$ cells in 5 ml of medium in a T-25 flask for 7 days. The total number of spheres was counted, and sphere diameter measured after 7 days. The neurospheres were then dissociated and passaged at a density of $2.5 \times 10^5$ cells in 5 ml of medium. This was repeated for another 3 passages.

**Golgi-Cox staining and neuron morphology analyses**. Animals were transcardially perfused at P60 using PBS followed by 0.04% PFA. Brains were dissected, impregnated, and stained using the FD Rapid GolgiStain Kit (FD Neuro Technologies, Columbia, USA) according to the manufacturer's instructions. Coronal brain slices were collected at 150 μm in 30% sucrose using a vibratome (Leica VT1000S). Golgi-Cox-stained sections were imaged using Zeiss Axio Imager widefield fluorescence microscope for Sholl and dendritic spine analyses. Upper-layer (II and III) cortical pyramidal neurons of the somatosensory cortex were selected for all morphological analyses. Axons and basal dendritic trees were manually traced using Simple Neurite Tracer[89]. The total dendritic length was measured, and the branching complexity quantified via the Sholl Analysis plugin[90] distributed within the FIJI software package[91]. For dendritic spine analyses, the number of clearly defined spines along a 10 μm stretch of each neuron's tertiary dendritic shaft was counted.

**Transcriptome analysis**. Cortical tissue was dissected from 14.5 dpc embryos and RNA was extracted using RNeasy Micro Kit (Qiagen) according to manufacturer's instructions. Prior to sequencing, the quality of RNA samples was assessed by Agilent 2100 bioanalyzer (Agilent Technologies, USA). RNA-sequencing was performed using the Illumina NovaSeq 6000 platform. Image analysis was performed in real time by the NovaSeq Control Software (NCS) v1.6.0 and Real-Time Analysis (RTA) v3.4.4, running on the instrument computer. RTA performs real-time base calling on the NovaSeq instrument computer. The Illumina bcl2fastq 2.20.0.422 pipeline was used to generate the sequence data. The sequence reads were aligned against the mouse genome (Build version mm10). The STAR aligner (v2.5.3a) was used to map reads to the genomic sequences. The counts of reads mapping to each known gene were summarized at gene level using the feature-Counts v1.5.3 utility of the subread package (available at http://subread.sourceforge.net/). The transcripts were assembled with the StringTie v1.3.3 tool (available at http://ccb.jhu.edu/software/stringtie/) utilizing the reads alignment with M6 and reference annotation-based assembly option (RABT). This generated assembly for known and potentially novel transcripts. The Gencode annotation containing both coding and non-coding annotation for mouse genome version GRcm38 (Ensemble release 81) were used as a guide (available at http://www.gencodegenes.org/mouse_releases/6.html). Differential gene expression was analyzed using edgeR (v3.26.0) in R 3.6.0. The default TMM normalization method of edeR was used to normalize the counts between samples. A generalized linear model was then used to quantify the differential expression between the groups. A CPM cut-off of 0.5 was used. Genes with less than this value in at least six samples were removed from the dataset. After controlling for sex by incorporating it into the design model as a batch-effect, we found 917 (536 upregulated [greater expression in mutant samples], 381 downregulated [greater expression in control samples]) out of 14,266 genes being differentially expressed in the comparison of mutant vs. control.

**Incorporating epigenetic repressive tendency to identify candidate genes in the transcriptome data**. To narrow down a candidate gene set responsible for *Elp2* mutant phenotypes, we incorporated the epigenetic repressive tendency score (RTS) into the conventional differential expression analysis output[56]. We calculated a metric that combined both significance of the DE analysis and regulatory importance of individual genes (i.e., $\frac{p-\text{value from DE analysis}}{\text{repressive tentency score}}$). We ranked genes by the p-value/RTS or p-value alone. We denoted up- and downregulations with plus and minus signs respectively.

To understand relevant biological information associated with top-prioritized genes, we performed GO enrichment analysis using DAVID on top 50 genes ranked by the modified p-value using the RTS[56]. Enrichment of selected GO terms (for biological process, cellular component and molecular function categories) and associated genes from both up- and downregulated top 50 genes were visualized with bipartite graphs.

**Protein extraction and western blotting**. Mouse brain cortex lysate was prepared by homogenization in radioimmunoprecipitation assay (RIPA) buffer (150 mM NaCl, 1% Nonidet P-40, 0.5% sodium deoxycholate, 0.1% SDS, 25 mM Tris pH 7.4) with protease and phosphatase inhibitors (Cell Signalling). Protein concentration was determined using the BCA assay (Pierce). Equal amounts of protein extract were resolved on SDS-PAGE using mini gel tank and blot system (Life Technologies). Total protein was transferred onto polyvinylidene difluoride membrane at 25 V for 90 min. After incubation in 5% skimmed milk in TBS-T (10 mM Tris pH 7.6, 150 mM NaCl, 0.1% Tween 20) for 1 hour, the membrane was washed twice

with TBS-T and incubated with antibodies against: Acetyl-α-tubulin (1:1000; T6793), α-tubulin (1:500; MA1-80017), β-actin (1:1000; ab8224), Elp1 (1:1000; 5071), Elp2 (1:500; ab154643) and Elp4 (1:500; NBP2-16322) at 4 °C overnight. Next day, the membrane was washed three times for 5 min in TBS-T and incubated with relevant horseradish peroxidase-conjugated secondary antibodies (1:2500; Abcam) for 1 h at room temperature on a shaker. Peroxidase activity was detected using SuperSignal West Pico chemiluminescent (Thermo Fisher Scientific) by ChemiDoc MP imaging system (Bio-Rad Laboratories). Densitometric analysis of western blot images was performed using ImageJ software[88].

**In-gel destaining and trypsin digestion of proteins**. Ten samples were excised from each lane of the Coomassie blue-stained SDS PAGE gel and placed into separate Eppendorf tubes for destaining. Destaining was performed by adding 500 μl of 100 mM ammonium bicarbonate/acetonitrile (1:1; vol/vol) buffer, followed by incubation for 30 min and vortexing every 10 min. The buffer was removed and 200 μl of acetonitrile was added to the gel pieces in two separate batches. The samples were left for 15–30 min in acetonitrile.

Upon removal of acetonitrile, the gel pieces were covered with 200 μl of sequence-grade trypsin of 20 ng/μl concentration in 50 mM ammonium bicarbonate pH 8 buffer (Promega Corporation). The samples were incubated at 37 °C overnight. Next day, the solution was removed and samples placed in clean tubes. This was followed by the addition of 200 μl of 5% formic acid/acetonitrile (3:1; vol/vol) and incubation for 15 min at room temperature under constant agitation. The supernatant with the trypsin solution was transferred to a clean tube for each sample and dried in a vacuum centrifuge.

Prior to high-performance liquid chromatography and mass spectrometry analysis, 12 μl of 1% (vol/vol) TFA in water was added to each tube, vortexed and/or incubated for 2 min in the sonication bath. This was followed by centrifugation for 1 min at $12,000 \times g$. Finally, the samples were transferred to an autosampler vials for analysis.

**High-performance liquid chromatography/mass spectrometry analysis and protein identification and quantification**. The tryptic peptide extracts were analyzed by nano-high-performance liquid chromatography (HPLC) and mass spectrometry on an Eksigent, Ekspert nano LC400 HPLC (SCIEX, Canada) coupled to a Triple TOF 6,600 mass spectrometer (SCIEX, Canada) with a PicoView nanoflow ion source (New Objective, USA). Two microliters of extract for each sample was injected onto a 5 mm×300 μm, C18 3 μm trap column (SGE, Australia) for 5 min at 10 μl/min. This was followed by washing the trapped tryptic peptide extracts onto analytical 75 μm × 150 mm ChromXP C18 CL 3 μm column (SCIEX, Canada) at 400 nl/min flow rate at 45 °C. For peptide elution, a linear gradient of 2–40% of 0.1% formic acid in acetonitrile over 60 min at 400 nl/min was used, which was followed by 40–90% gradient and then 90% of this solvent, each for 5 min. Following each elution, the gradient was returned to 2% of 0.1% formic acid in acetonitrile for equilibration before injecting the next sample. The ionspray voltage was set to 2600 V, declustering potential (DP) 80 V, curtain gas flow 30, nebuliser gas 1 (GS1) 30, interface heater at 150 °C. 50 ms full-scan TOF-mass spectrometry (TOF-MS) data was acquired by the spectrometer, followed by up to 30 100 ms full-scan product ion data with a rolling collision energy and in an Information Dependant Acquisition (IDA) mode. The TOF-MS data were acquired over the mass range 350–1800 and for product ion ms/ms 100–1500. Ions in the TOF-MS scan that were found to exceed a 200 counts threshold and +2 to +5 charge state were set to trigger the acquisition of product ions.

For peptide and protein quantification, data were acquired under the same HPLC and mass spectrometry experimental conditions, with the following exceptions. Data were acquired using a SWATH, product ion ms/ms all approach. The SWATH experiment was set to acquire 100 product ion spectra from m/z 350 to 1500 per scan cycle with a product ion window was set to 6 Da and collision energy from 16 to 60 V with an energy spread of 5 V. The TOF-MS scan acquisition time was set to 50 ms and each product ion scan to 25 ms.

The data were acquired and processed using Analyst TF 1.7 software (ABSCIEX, Canada). Protein identification and library compilation were carried out using Protein Pilot 5.0.2 for database searching. Peptide sequence identification and protein grouping were conducted by merging all individual IDA acquisition file from the gel fractions using the Paragon algorithm from Protein Pilot 5.0.2 against the *Mus musculus* Uniprot database complemented by the contaminant sequences from thegpm.org/crap/ with the following parameters: Cys alkylation was set to iodoacetamide, Trypsin was chosen as digestion enzyme, Special Factors: Gel-based-ID was enabled to account for potential, ID Focus: Biological modifications, Thorough ID, ProtScore (Conf) >10%, and False discovery analysis. A total of 3454 individual proteins was identified with <1% global FDR and shared peptides across multiple proteins were removed before being aligned with the SWATH acquisition files using SWATH Acquisition MicroApp 2.0 on the PeakView 2.2 software. The processing settings were set as follows for the peak area extraction in all SWATH acquisition using the local generated ion library: Number of peptides per Protein: 6, Number of Transitions per Peptides: 6, Peptide Confidence Threshold: 99%, <1% FDR. Modified peptides were excluded for the protein quantitation and XIC extraction window: 15 min with a mass width of 50 ppm and retention times were aligned using a set of high intense Trypsin autodigest peptides. The ion, peptide, and protein area were exported as a text file

and the integrated area values for each protein that was present in different gel bands for each gel lane were summed before statistical evaluations, correlation of variation, and fold change computation. Normalization evaluations were done using the Normalyzer tool[92].

**Proteome data analysis**. A total number of 12,600 peptides, including six spiked-in peptides, corresponding to 2695 proteins were identified by DIA/SWATH-MS. Peptide-level intensities were log2 transformed and normalised using the cyclic loess method. Technical variability (instrument RT and intensity variability) was estimated from spiked peptides and subtracted from peptide total variance. Differentially expressed peptides between mutant and control samples were identified using Linear models with Empirical Bayes moderation. Protein-level $p$-values were obtained by Sime's adjustment of peptide-level $p$-values (diffSplice and topSplice functions in limma). Protein log2 fold-changes were determined as trimmed mean of its constituent peptides, where the top and lower 20% of peptide log2 fold-changes were discarded. Proteins were defined as differentially expressed if they achieved FDR <0.1. All statistical and pathway enrichment analysis were performed in limma[93] R/Bioconductor package. The significance of protein length enrichment in each set of up- and downregulated proteins were quantified using Wilcoxon test in R. For each set of up- and downregulated proteins ($n = 11$ and 25 respectively), we tested if there was a difference between shorter and longer proteins in the set using Wilcoxon test. The short and long criteria were defined with respect to the median protein lengths in the study. Hence, shorter lengths refer to smaller than median, and longer refers to larger than median.

The relative synonymous codon usage (RSCU) index for all genes coding the differentially expressed proteins was calculated using the uco function in seqinr (R package, v3.4). Differences in codon usage bias between up- and downregulated proteins were calculated using Mann-Whitney U tests (wilcox.test function in R). The strength of codon usage bias ($\Delta$RSCU) was calculated as a median difference in RSCU values of up- and downregulated proteins.

**tRNA modification analysis**. Approximately 100 mg of whole-brain tissue was collected from 14.5 dpc embryos and homogenized with ceramic beads (Sapphire Bioscience) using TRIzol reagent (Life Technologies) in tissue homogenizer (Bertin Technologies). Total RNA and tRNA isolation, tRNA hydrolysis, and tRNA modification analysis via high-performance liquid chromatography coupled to mass spectrometry was performed as previously described[26]. Mass spectrometer parameters were optimized for targeted ribonucleosides using multiple injections of 0.1–1 ng of commercially obtained $m^7G$ (MetaGene) and purified $ncm^5U$, $mcm^5U$, and $mcm^5s^2U$ nucleosides (a generous gift from Sebastian Leidel, University of Bern, Switzerland). The obtained retention times were superimposed with the previously published results[94]. An analytical method was developed for the simultaneous analysis of modified ribonucleosides of which six compounds of interest were targeted for quantification ($ncm^5U$, $mcm^5U$, $mcm^5s^2U$, $\Psi$, $m^1A$, $m^7G$, and $t^6A$). Pseudouridine ($\Psi$) served as an internal normalization standard. MultiQuant v2.1.1 (AB Sciex) software was used for peak assignment, quantification, and normalization.

**Statistical analysis**. Statistical analyses were completed using Prism v8.0.1 (GraphPad) software. Number of replicates, corresponding statistical tests, and statistically significant differences are indicated in figure legends. Where appropriate, multiple comparisons were corrected via the Holm-Sidak method. Differences between groups were considered significant for $p \leq 0.05$.

**Reporting summary**. Further information on research design is available in the Nature Research Reporting Summary linked to this article.

## Data availability

The *ELP2* variants identified in the patients have been deposited for public access within ClinVar[95] with accession numbers SCV000262876.5 (https://www.ncbi.nlm.nih.gov/clinvar/variation/225008/), SCV000262928.5 (https://www.ncbi.nlm.nih.gov/clinvar/variation/225038/), SCV000590102.3 (https://www.ncbi.nlm.nih.gov/clinvar/variation/432384/), SCV000589770.3 (https://www.ncbi.nlm.nih.gov/clinvar/variation/225008/), SCV001244968.1 (https://www.ncbi.nlm.nih.gov/clinvar/variation/870396/), SCV001244969.1 (https://www.ncbi.nlm.nih.gov/clinvar/variation/870397/), SCV001482303.1 (https://www.ncbi.nlm.nih.gov/clinvar/variation/1012183/), SCV001482306.1 (https://www.ncbi.nlm.nih.gov/clinvar/variation/1012184/), SCV001482307.1 (https://www.ncbi.nlm.nih.gov/clinvar/variation/870396/), SCV001482309.1 (https://www.ncbi.nlm.nih.gov/clinvar/variation/1012185/) and SCV001482310.1 (https://www.ncbi.nlm.nih.gov/clinvar/variation/1012186/). Transcriptomic data that support the findings of this study are contained in Supplementary Data 1 (DE genes) and Supplementary Data 2 (TRIAGE matrix), and proteomic data in Supplementary Data 3 (DE peptides). The raw RNA-sequencing data are available in the NCBI Sequence Read Archive with accession number PRJNA692814. The mass spectrometry proteomics data have been deposited to the ProteomeXchange Consortium via the PRIDE[96] partner repository with the dataset identifier PXD023653. All other data generated in this study are available from corresponding authors on reasonable request. Source data are provided with this paper.

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

## Acknowledgements

We thank all members of the Wainwright and Glatt laboratories for discussion and suggestions. The authors acknowledge Jérémy Potriquet from the AB Sciex company, Brisbane, Australia, for his assistance with proteome data quantification and analysis. This work was supported by the First Team Grant (FirstTEAM/2016-1/2; A.C.G. and S.G.) from the Foundation for Polish Science. In addition, we thank the MCB structural biology core facility (supported by the TEAM TECH CORE FACILITY/2017-4/6 grant from Foundation for Polish Science) for providing computational resources and support. We acknowledge the supports from the Queensland NMR Network and the National Imaging Facility (a National Collaborative Research Infrastructure Strategy capability) for the operation of 16.4 T MRI at the Centre for Advanced Imaging, the University of Queensland. Patient 6 (refer to Supplementary Table 1) was tested through the Acute Care Flagship of the Australian Genomics Health Alliance, supported by grants from the Royal Children's Hospital Foundation (2017-906) and the National Health and Medical Research Council (GNT1113531). The open-access publication of this article was funded by the BioS Priority Research Area under the program Excellence Initiative – Research University at the Jagiellonian University in Krakow.

## Author contributions

The study was designed and directed by M. Kojic, B.J.W. and S.G. T.G. produced and purified all human Elp2 variants with the help of M.G. and A.S.K. T.G. established the human Elp123 expression system together with M.G., A.S.K., and A.K. K.D. produced all murine Elp2 variants with the help of M.G., A.S.K., and A.K. T.G., A.S.K., K.D., and M.G. biochemically characterized all Elp2 proteins and conducted stability, tRNA interaction, and acetyl-CoA activity assays. A.C.G. produced and purified all native and labeled tRNAs. Patient recruitment, collection, and analysis of the clinical and genetic data were directed by R.S.M. and performed by R.K.A., C.D.F., C.G.M., J.S.C., A.F., Z.S., S.L., J.L. J.K.H., M.F.B., B.K., I.M., M.S.S. K.B. and R.S.M. M. Kojic, S.A.A., and T.H.J.B. designed and performed characterization of the mouse phenotypes. M. Kojic collected the samples, N.D.K. performed MRI imaging and A. Begg and N.D.K analyzed the MRI data. A. Begg performed Golgi-Cox staining and neuron morphology analyses. H&E and immuno-fluorescence staining and analyses, as well as western blotting was done by M. Kojic. Neurosphere essay was done and data interpreted by M. Kasherman and M.P. Elp2 and parvalbumin expression in the brain was assessed by A. Batzilla and M. Kojic. Brain tissue for transcriptome analysis was collected by M. Kojic and RNA extracted by A. Batzilla and A. Begg. L.A.G., M.K., S.H.Z., and M.J.D. directed transcriptome and pro-teome analyses. W.J.S, E.S., M.B., and N.J.P. developed TRIAGE. W.J.S., N.P., and M.B. consulted, designed, and performed the TRIAGE analysis and wrote the text describing it. A. Begg, A.J., and M. Kojic performed tRNA modification analyses. S.M. developed a tRNA modification analysis method together with A.J. and M. Kojic. M. Kojic collected protein samples and A.J. performed protein digestion, analysis, and quantification. S.H.Z. and M.J.D. analyzed the differentially expressed peptides, protein length, and codon usage. The manuscript was written by M. Kojic and revised by B.J.W., S.G., and M.G., with all authors discussing the results and refining and approving the final version of it.

## Competing interests

The authors declare no competing interests.

## Additional information

[1]The University of Queensland Diamantina Institute, Translational Research Institute, The University of Queensland, Brisbane, QLD, Australia. [2]Institute for Molecular Bioscience, The University of Queensland, Brisbane, QLD, Australia. [3]Malopolska Centre of Biotechnology, Jagiellonian University, Krakow, Poland. [4]Postgraduate School of Molecular Medicine, Warsaw, Poland. [5]Centre for Advanced Imaging, The University of Queensland, Brisbane, QLD, Australia. [6]Bioinformatics Division, Walter and Eliza Hall Institute of Medical Research, Melbourne, VIC, Australia. [7]Department of Medical Biology, Faculty of Medicine, Dentistry and Health Sciences, The University of Melbourne, Melbourne, VIC, Australia. [8]School of Biomedical Sciences, The University of Queensland, Brisbane, QLD, Australia. [9]School of Chemistry and Molecular Biosciences, The University of Queensland, Brisbane, QLD, Australia. [10]Department of Epilepsy Genetics and Personalized Medicine, Danish Epilepsy Centre, Dianalund, Denmark. [11]Centre for Functional and Diagnostic Imaging and Research, Hvidovre Hospital, Hvidovre, Denmark. [12]Department of Neurology and Developmental Medicine, Division of Neurogenetics, Kennedy Krieger Institute, Baltimore, MD, USA. [13]Department of Neurology, Johns Hopkins University School of Medicine, Baltimore, MD, USA. [14]Department of Pediatrics, Johns Hopkins University School of Medicine, Baltimore, MD, USA. [15]Department of Paediatrics, The University of Melbourne, Melbourne, VIC, Australia. [16]Victorian Clinical Genetics Services, Murdoch Children's Research Institute, Melbourne, VIC, Australia. [17]Australian Genomics Health Alliance, Parkville, VIC, Australia. [18]The University of Melbourne, Melbourne, VIC, Australia. [19]Department of Metabolic Medicine, Royal Children's Hospital, Parkville, VIC, Australia. [20]Department of Paediatrics, Regional Hospital Viborg, Viborg, Denmark. [21]Department of Genetics, Pitié-Salpêtrière Hospital, AP-HP, Paris, France. [22]The University of Colorado Anschutz, Children's Hospital Colorado, Aurora, CO, USA. [23]Queensland Brain Institute, The University of Queensland, Brisbane, QLD, Australia. [24]Queensland Centre for Mental Health Research, The Park Centre for Mental Health, Brisbane, QLD, Australia. [25]CSIRO-QUT Synthetic Biology Alliance, Centre for Tropical Crops and Bio-commodities, Queensland University of Technology, Brisbane, QLD, Australia. [26]The Ruprecht Karl University of Heidelberg, Heidelberg, Germany. [27]Department of Clinical Pathology, Faculty of Medicine, Dentistry and Health Sciences, The University of Melbourne, Melbourne, VIC, Australia. [28]Department for Regional Health Research, The University of Southern Denmark, Odense, Denmark. ✉email: sebastian.glatt@uj.edu.pl; b.wainwright@uq.edu.au

