## [Peer Review File · Nature Communications]

Reviewers' comments:

Reviewer #1 (Remarks to the Author):

This study by Kojic et al reports on the phenotype of humans and mice that have shared mutations in the ELP2 gene, a key subunit of the six-subunit Elongator complex. The study is timely as there is growing interest in the function of the Elongator complex in mammals as mutations in its various subunits have all (except of Elp5) been shown to be associated with neurological disorders. To its credit, this is the first detailed analysis of Elp2 in the nervous system. While there are many strengths to this study (listed below), there are some serious gaps in the data set and in the identification of underlying mechanisms, that preclude endorsing the strong conclusions the authors draw. While the breadth of the data presented is impressive, (1) the depth required to support the data are missing in some cases, as is causality often, and (2) many of the reported findings are confirmatory in that they support already published data the field has accumulated from similar studies on other Elongator subunits, including Elp1, Elp3, and Elp6 (by the authors themselves), hence this reviewer is not convinced that the data here reveal sufficiently novel information about Elongator function in health and disease to warrant publication in Nature Communication.

Strengths include:

- detailed analysis of the patients is a contribution to our understanding of the clinical phenotype of patients with various Elongator subunit mutations.
- Generation of mice with actual patient mutations is a powerful approach for generating models for interrogation of Elp2 function.
- Multidisciplinary approach to mutation analysis including human neurological exams and scans, mouse scans, mouse behavior, ELP2 structural information and protein dynamics, and neurodevelopmental and adult degenerative analyses, proteomic and transcriptome analysis of Elp2 mutant mouse brains, and finally, demonstration of reduced tRNA modification, a hallmark of Elongator function.
- Structural information regarding the location of the Elp2 mutations is very interesting and a contribution to our understanding of the structure/function relationship of the various elongator subunits.

Weaknesses:

1. Overstating conclusions:

The authors claim: "Further, we show that impaired function of the complex leads to proteome instability and consequent neurogenesis defects, myelin loss and neurodegeneration. Elp2 mutations perturb protein homeostasis by reducing expression of the genes that guide cell cycle and translation, and up-regulating protein degradation in neurons and oligodendroglia."

Importantly, the demonstration of the direct causality of these processes is not sufficiently strong, nor are the mechanistic links between these events rigorously demonstrated, to support these conclusions.

2. While the authors claim that the mutated Elp2 protein is much less stable, it becomes unclear how significant that is for the subsequent protein studies. Elp2 levels are never measured in any of the mouse models – are they reduced at all? Or all the deleterious mouse effects the result of aberrant protein function? This point must be addressed. On Pg. 7: "No significant destabilization was observed for the L98F and H206R mutations." Yet, the H206R mutation is one of the most serious mutations, and the mouse model that is used for the majority of their subsequent studies – so how meaningful is the protein stability decrement? Furthermore, the mouse H206R mutation does have a decreased melting temperature although the human does not. Which means this statement on pg. 7 is not accurate "The relative severity of the individual mutations is almost identical for murine and human Elp2 proteins."

Yet if protein stability is reduced for the "majority" of mutations, a surprising finding is that the Elp123 complex forms normally. This raises the interesting question of whether Elp2 has functions independent from its association with the other elongator subunits to form the Elongator complex. This potential independent function of Elp2 is never addressed but should be. More perplexing is

that although the complex forms normally, all 5 mutations tested had reduced ACO hydrolysis rates. So does protein stability have anything to do with the decreased ACO hydrolysis? Again, mechanistic links between these findings were not made clear.

3. Questions about brain morphogenesis calculations. Table S2 – were the relative brain weights normalized? Brains in the mutants are overall smaller, so of course each individual brain structure/nucleus will be smaller than that in the control, but are the mutants' individual structure volumes proportional to the reduction in brain mass? Same for Fig. 7 a & b – in addition to the data shown in those figures, the thickness of cortex and VZ should be normalized to thickness of the entire section (ventricle to outer cortical layer) since you have already shown the brains are smaller. The question is whether the relative proportion of mitotically-active cells compared to post-mitotic cortical layers differ between mutant and control. Reduced brain size has been shown before for Elp1 and Elp3 mutant mice and should be mentioned.

4. Dendritic branching defects (Fig. 6) has been shown before too by work from four separate labs (Weil, Tourtellotte, Lefcort and Nguyen) for Elp1 and Elp3 loss-of-function. Similarly, several of these groups (and others, e.g. in *C.elegans*) have also shown reductions of tubulin acetylation in neurons with reduction of elongator subunits. However, demonstration of alterations in spine number is new and an interesting finding.

5. For Figure 7c & d, how was the cell quantification conducted? Please describe the methods used. Also in contrast to what is claimed in the manuscript, Ki67 labels mitotically active progenitor cells throughout the cell cycle, not just in S phase. The data in Fig. 7d are informative and interesting, showing that the % of Brdu+/Ki67- cells is reduced in the mutant. However, it is unclear how to interpret the Ki67 or pHH3 data alone since again, the proportion of progenitor cells vs. post-mitotic neurons should be calculated. Since the brains are overall smaller one would predict to see fewer absolute numbers of cells in the mutant (unless cell density is perturbed?) – but the question is are all cell types reduced equally or primarily progenitors or post-mitotic neurons? For example, based on the Sox2 and Tbr2 staining, it looks like proportionally, there is an increase in progenitor cells relative to post-mitotic neurons in the mutants but this needs to be quantified. Also, given the increase seen in Brdu+/Ki67- cells, one should also check to see if the progenitors are dying e.g. with TUNEL or cleaved caspase 3.

6. The transcriptome data by itself make it difficult to ascribe causality. First of all, Elongator is required for translation, meaning: do progenitors exit the cell cycle because in the absence of Elp2 they are incapable of translating the mRNAs for the transcription factors required for neurogenesis OR are the levels of these genes reduced because there are fewer progenitors in the mutants as shown in Fig. 7? Are the transcription factors that drive neurogenesis AA-codon biased? To attribute the reduction in neurogenesis to a reduction in the Elp2-regulated transcriptome, would require showing that the key genes that mediate neurogenesis are in fact reduced in the mutant because of codon-bias. Alternatively, these genes could be downstream of a gene that is codon-biased, but at least some attempt to connect these mechanisms must be made to explain the processes that mediate the Elp2 mutant phenotype.

7. Nicely, the authors show that the Elongator-mediated mcm5U modifications are reduced on tRNAs in the mutants, which supports several previously published data on Elongator function in multiple species (yeast to humans), and sets the stage for the subsequent proteomic analysis. However, the authors fail to mine their proteomic data to determine whether the proteins that mediate neurogenesis or CNS circuit formation are in fact reduced and also translated from AA-biased codons. Without that analysis, it's hard to draw conclusions about the functional connections between the transcriptomic, proteomic and developmental changes exhibited by the Elp2 mutant mice.

8. Interpretation of mouse behavior studies (and patient phenotype) are perhaps misguided. Both the mouse and patient neurological exams indicate that these patients and mice have profound CNS deficits including pronounced motor problems. The majority of patients do not move at all, have severe hypotonia and spasticity, while the mice have tremors, gait problems, reduced grip strength, hypotonia and hind limb clasp – the latter a classic CNS descending pathway lesion. That means these patients/mice are not necessarily a classic model of Autism and/or intellectual

disability – instead their entire CNS seems to have suffered from a systematic dysfunctional development, in particular including sensorimotor function. That raises a few questions including whether the patients have additional mutations and/or copy number variants that exacerbate the phenotype, rather than a dysfunctional *Elp2* being the sole culprit. However, the fact that recapitulation of the motor problems is obtained in the *Elp2* mutant mice at least supports the *Elp2* mutation driving the CNS dysfunction in the patients.

That being said though, while it's clear the patients in addition to their motor problems also have ID, it's less clear how severely the mice display ASD or ID behaviors. Their repetitive behavior data fits with ASD, but not, to my knowledge, with ID. However, the repetitive behavior test is fairly short: the mice had a 10-minute habituation, then recorded for 10 minutes. They only groomed for 20 seconds over 10 minutes. It's also possible that the repetitive grooming is a self-soothing behavior in response to stress. As harmless as it seems, putting the mouse into a novel chamber, even with the habituation period, could be stressful and trigger grooming. Perhaps it is not an ASD-related repetitive behavior, but an abnormal stress response. This is certainly not a deal-breaker, but a longer recording period and assessment in a home-cage environment would be more informative of an ASD-type phenotype.

The authors claim an ASD-like reduction in ultrasonic vocalization (USV) communication. The experiment is reasonable – maternal separation of pups triggers USV from the pups. Some autism models have fewer USVs in that paradigm, suggesting that the mice do not communicate well. However, impairment of several neurotransmitter systems can influence USVs (5HT, GABA, opioids, cannabinoids, vasopressin, oxytocin). In summary, there's not a clear-cut pattern that ASD models always have reduced USVs. In fact, in a *Mecp2* KO model of Rett syndrome (typically considered as part of the autism spectrum) KO mice have increased USVs.

Finally, the major ASD characteristic that the authors did not address at all is social interaction. There are multiple ways to assay it, that it seems like a very reasonable experiment if one wants to interrogate ASD behaviors. It's possible that the mouse motor problems may preclude conducting the social interaction experiments but at least this point should be addressed since it is a classic hallmark of ASD.

9. Importantly, whether ER stress is truly responsible for the cell death of PC can't be determined by the data shown in Fig. S8 – there is no quantification of dying PCs and to conclude that ER stress is the cause is premature based on the preliminary data shown. Studies that demonstrate ER stress are far more comprehensive than the studies done here. Fig. S9 does have some quantification of the loss of glial cells, which is interesting. Do the APC+ oligodendrocytes express *Elp2* or is their loss an indirect effect of the loss of Purkinje cells? That would be important to determine. Also the images in both Fig. S8 and S9 should include a merged view to show overlap in expression of the markers.

10. There are some confusing inconsistencies at several points in this study. For example, between images shown of mouse tissue/structures and the associated quantified data. Fig. S8a, the P60 brains of the mutants and control do not look as dissimilar as they do in Figs. S4A at P21. Why? Additionally, for the brains depicted in Fig. 4A, the H206R mutant brain does not look smaller than the WT which raises questions as to the data shown in Fig. 4d.

In addition, it seems as if *Elp2* is not required in cerebellar progenitors in contrast to its requirement in cortical progenitors, based on the data in Fig. S8c. That is an interesting difference and warrants comments by the authors. In contrast, PC number is significantly reduced by P60 in the mutants, suggesting a degenerative process. Do the *Elp2* patients have ataxia? Or are their movement impairments so severe that ataxia can't be discerned?

A major source of confusion is the mouse lines used for the different experiments. The authors state the mice die by around P20 on the C57BL/6 background but can live up to P60-120 when put on a DBA2/J background. That raises the question of the connection between the developmental histological/immunochemical analyses done in the mice on the C57BL/6 background and the mouse behavior and adult work done in mice on the DBA2/J background. For example, is life span extended to such a dramatic extent by changing background because the neurodevelopmental

impairments identified in the C57/bl6 line do not occur in the DJA2/J cross? If so, how are we to interpret the behavioral abnormalities? The consistency in phenotype between these two different mouse lines (both with the same H206R mutations) needs to be clarified.

10. Novelty regarding our understanding of the Elongator complex function in neural development and CNS function. Many of the major findings regarding these events have been demonstrated in previous publications, most of which are not cited by the authors. Thus the conclusion of the paper as stated "our work reveals the role of epitranscriptomic processes in neural development and homeostasis which, when perturbed, lead to profound CNS defects in both humans and model systems" is not novel. While the authors show that Elp2 does participate in the same CNS steps that we already know Elp1 and Elp3 are required for, which is interesting although not at all unexpected based on a decade of work on the Elongator complex from yeast to *C.elegans* to mouse to humans, what is missing from this study is a deeper probe of the mechanisms mediating these aberrant processes.

Reviewer #2 (Remarks to the Author):

Elongator is a protein complex mainly involved in tRNA modifications and whose dysfunction has been associated with several neurological disorders, ranging from Familial dysautonomia, ALS, to intellectual disabilities. Previous animal models showed that loss or reduced Elongator activity in the nervous system affects the developmental program of neurogenesis and alters neuronal fitness in the central and autonomic nervous system, further leading to neurodegeneration. Here new pathological human variants of Elp2 are described and humanized mouse models are used to decipher the impact of those variants on brain development and maintenance. This work nicely shows that these mice suffer from microcephaly and cerebellar degeneration. Moreover, Elp2 variants impair Elongator activity leading to brain defects associated with changes in connectivity that may underlie specific behavioral defects.

This ms is interesting and overall well written but is unfortunately quite descriptive and would benefit from additional mechanistic insights. Moreover, some conclusions do not stand because data should be revisited under new light for proper interpretation.

Major concerns

The functional demonstration that Elp2 variants affect the ability of the Elp123 sub-complex to promote tRNA-induced acetyl-CoA hydrolysis *in vitro* is suggestive but not demonstrative of the negative impact of Elp2 variants on the activity of Elongator as a whole.

Regarding the behavioral screen. Some experiments, such as self-grooming should be confirmed in adult mice (p21 is adolescence). ASD is also classically associated with sociability defects and authors should assess social preference and social recognition in adult mice.

Regarding the connectome studies, these parameters should be acquired in mature brain, not in adolescents who are still undergoing brain maturation (synapse pruning and connectivity refinement).

The histological characterization of the cortex is problematic and leads to misinterpretations. The distribution of Cux1 staining across cortical layers is unexpected as all layers are positive in Figure 7a. To assess microcephaly, not only the thickness should be measured (done here), but also the absolute number of cells as well as their density. This is critical since dendritogenesis is strongly affected in mutant (see figure 6) and this can in addition to progenitor defects, lead to progressive changes of cortical thickness and neuron density after birth (progressive microcephaly). The analysis of brain weight and dimensions at different stages (during development and after birth) would help deciphering if there is indeed a worsening of the microcephaly after birth.

There are also semantic issues. First, the term "neural stem cells" is vague and incorrect as

projection neurons are generated by both VZ and SVZ cells that are named apical progenitors (APs; or radial glial cell, RGCs) and basal progenitors (BP; or intermediate progenitors), respectively.

Other issues refer to the analysis of cell proliferation. First, both VZ and SVZ progenitors are indeed cycling progenitors. Second, Ki67 immunostaining marks most cell cycle phases and is not restricted to cells entering into S-Phase. Also, the way the cell cycle exit is assessed is wrong, a 2 hours BrdU pulse does not give enough time for a cortical progenitor to exit the cell cycle, which also questions the quality of the overall histological analysis performed in the manuscript.

Increased apoptosis during cortical development has been shown to be a cause of microcephaly in animal models, it would be important to check for apoptosis markers during development and not just in the adult cerebellum and mature oligodendrocytes.

For the neurosphere assay, there is no SVZ purification step and thus cycling progenitors must come from both VZ and SVZ. The conclusion of the paragraph is thus wrong. There is also a major comment regarding the interpretation of the cortical phenotype. There is no good experimental demonstration that Elp2 mutation induces cell cycle exit and previous work reported that loss of Elongator activity in cortical progenitors leads to microcephaly as a result of a defect in the fate of VZ progenitors, thereby changing the balance between direct and indirect neurogenesis. This seems to happen also for the Elp2 humanized mice who show a significant reduction of newborn IP production (see figure 7c). In other words, it is necessary to assess direct versus indirect neurogenesis to take a final conclusion about the cellular mechanism of microcephaly. The transcriptional analysis of dissected cortices revealed reduced expression of genes that are strongly enriched in the ventral forebrain progenitors during development (top genes such as *Ascl1*, *Gsx2*..), which suggests that the tissue dissected include more than the cortex. What is the relevance of reduction of expression of these genes in the cortex? This is problematic because Elongator is also important for cortical interneurons as previously reported. Thus the molecular analysis is difficult to interpret in regard to projection neurons. This part needs a better mechanistic understanding of the phenotype.

It is also surprising that the authors focused on chronic UPR-induced neurodegeneration in the cerebellum without checking ER stress and UPR in the cortex as this pathway was previously shown and experimentally tested by other to explain the microcephaly phenotype. This needs to be addressed for Elp2 humanized mice in their cortex.

Elongator has been involved in the regulation of postmitotic interneuron migration, thus what is the phenotype the cortical interneurons and their progenitors in Elp2 humanized mice?

Minor concerns

Why switching from H206R (for axonal tracts analysis Fig5) to H206R/R464W (for dendritic complexity fig6) to H206R mouse model again (for proliferation studies, fig7)?

Typos: abstract: a series of patients

Figure S6C: the blot is poor, its exposition is too long to make quantitative measurement.

Fig S6C : WB analysis for adult or embryonic samples ?

Change in the main text that Tbr2 are in the SVZ and Sox2 in the VZ and not the opposite, and change cycling progenitors for radial glial cells :

« Analyses of Elp2H206R cortical progenitors showed that both intermediate progenitors (IPs; Tbr2+) in the ventricular zone and cycling progenitors (Sox2+) in the subventricular zone (SVZ), were affected (Figure 7c). »

Reviewer #3 (Remarks to the Author):

In the manuscript by Kojic et al., the authors found ELP2 mutations in ID and ASD patients, and performed biochemical experiments using recombinant ELP1/2/3 proteins, and physiological experiments using Elp2 mutant mice. Although preceding studies already described that ELP2 mutations were associated with ID (Ref 13-15), Kojic et al. made Elp2 mutant mice, and directly proved that Elp2 is responsible for ID, providing physiological insights into the onset of ID. The physiological analyses in this manuscript are fine, but there are many undefended conclusions made from molecular analyses. Specific comments are described below.

Major points:

- General. There is a paper published in 2018 on Elp2 KO mouse (Lu et al. BBRC. 2018. PMID: 29723529), which described a different phenotype from the authors' report. I wonder how the authors would interpret their results.

- In Figure 2, the authors performed gel filtration experiments and melting curve experiments using recombinant human ELP2 to show that recombinant ELP2 proteins harboring patient mutations are structurally less stable compared to WT ELP2. Although the results are beautiful and suggestive, the results are not sufficient to say that "Elp2 mutations perturb the protein stability" as written in the paragraph title in page 6. This is because the authors only checked the structural stability of recombinant proteins in test tubes and not within cells. To defend such a conclusion, it is necessary to check at least one of the followings: 1) comparing WT and mutant Elp2 protein steady-state levels in mouse brains, 2) comparing WT and mutant Elp2 protein steady-state levels in culture cells (if antibody to endogenous Elp2 is not available, tagged Elp2 can be used), 3) comparing decay speed of WT and mutant Elp2 in culture cells using a translation inhibitor such as cycloheximide.

- In Figure 7f-7h, the authors performed RNA-seq and Gene Ontology analysis of mRNAs that were up- or down-regulated in Elp2 mutant mouse brains. In the Methods section page 48, the authors wrote that they "found 917 (536 up-regulated, 381 down-regulated) out of 14266 genes being differentially expressed in the comparison of mutant vs control". However, in the Gene Ontology analysis in Figure 7h, the authors seemed to have analyzed only about 35 down-regulated genes and 23 up-regulated genes. The authors should show the criteria to have chosen the genes. Also, the authors should show P-values for each enriched gene ontology term. These two are prerequisites to judge if the results of gene ontology analysis are valid.

- In Figure 8d-g, the authors performed peptide mass spec and used the results to make analysis on the length and codon usage of proteins that were up- or down-regulated in Elp2 mutant mouse brains. In Figure 8f, the authors checked up- or down-regulated proteins and compared their "length". There seems to be at least two problems with this analysis. First, in Figure 8d, there are hundreds, maybe thousands of proteins that were up- or down-regulated, but in Figure 8f, only 11 up-regulated proteins and 25 down-regulated proteins were used. The authors should clearly write the criteria to have chosen the proteins. Otherwise, I cannot remove skepticism on the arbitrariness of the analysis and conclusion. Secondly, to conclude that Elp2 mutation is "specifically affecting long transcripts" as written in the Summary, statistical analysis is absolutely required. For this kind of analysis, for example, I see people comparing e.g. >100 proteins that were upregulated against >100 proteins that were downregulated, drawing a cumulative distribution curve (horizontal axis: protein length, vertical axis: cumulative protein number %), and performing a Kolmogorov-Smirnov statistical test.

- In Figure 8g, among the codons whose usage decreased in down-regulated proteins, CAA codon (Glutamine), GAA codon (Glutamate), and AAA codon (Lysine) are highlighted. These codons are decoded by tRNAs bearing mcm5s2U, and not by tRNAs bearing mcm5U (e.g. Johanson et al. MCB. 2008. PMID: 18332122). So, in addition to ncm5U and mcm5U measured in Figure 8a & 8b, it is absolutely necessary to check the mcm5s2U modification status. Also, as Elp2 is required for ncm5s2U modification, it is preferable to also measure ncm5s2U level.

- In Figure 8g, the authors discussed about NAA codons (which are decoded by mcm5s2U-bearing tRNAs). mcm5s2U is known to promote the translation of NAA codons, so this is good. On the

other hand, in Figure 8g, the authors did not discuss about codons that are decoded by mcm5U/ncm5U/ncm5s2U-bearing tRNAs, although all of these modifications are synthesized using Elp2. mcm5U in tRNAs (Arg, Gly) anticodon promotes recognition of G-ending Arg and Gly codons AGG and GGG. ncm5U in tRNAs (Val, Ser, Thr) promotes recognition of G-ending GUG, UCG, ACG codons (Johanson et al. MCB. 2008. PMID: 18332122). It is unfair and biased to only discuss about codons decoded by tRNAs bearing mcm5s2U-bearing and not discuss about codons decoded by tRNAs bearing mcm5U/ncm5U/ncm5s2U. The authors should either discuss about these codons or remove Figure 8g and its implications.

Minor points:

- In Figure 8g, as a codon whose usage decreased in down-regulated proteins, TAA codon is highlighted. However, TAA is a stop codon, not requiring a tRNA or tRNA modification. So, this is peculiar. I wonder how the authors would interpret this result.

We thank the reviewers for their time and insight. In response to these comments we have performed additional experiments, additional analyses and edited the text to provide greater clarity. Consequently we believe it is significantly improved.

Response to reviewer comments

Reviewer #1 (Remarks to the Author):

This study by Kojic et al reports on the phenotype of humans and mice that have shared mutations in the ELP2 gene, a key subunit of the six-subunit Elongator complex. The study is timely as there is growing interest in the function of the Elongator complex in mammals as mutations in its various subunits have all (except of Elp5) been shown to be associated with neurological disorders. To its credit, this is the first detailed analysis of Elp2 in the nervous system. While there are many strengths to this study (listed below), there are some serious gaps in the data set and in the identification of underlying mechanisms, that preclude endorsing the strong conclusions the authors draw. While the breadth of the data presented is impressive, (1) the depth required to support the data are missing in some cases, as is causality often, and (2) many of the reported findings are confirmatory in that they support already published data the field has accumulated from similar studies on other Elongator subunits, including Elp1, Elp3, and Elp6 (by the authors themselves), hence this reviewer is not convinced that the data here reveal sufficiently novel information about Elongator function in health and disease to warrant publication in Nature Communication.

Strengths include:

- detailed analysis of the patients is a contribution to our understanding of the clinical phenotype of patients with various Elongator subunit mutations.
- Generation of mice with actual patient mutations is a powerful approach for generating models for interrogation of Elp2 function.
- Multidisciplinary approach to mutation analysis including human neurological exams and scans, mouse scans, mouse behavior, ELP2 structural information and protein dynamics, and neurodevelopmental and adult degenerative analyses, proteomic and transcriptome analysis of Elp2 mutant mouse brains, and finally, demonstration of reduced tRNA modification, a hallmark of Elongator function.
- Structural information regarding the location of the Elp2 mutations is very interesting and a contribution to our understanding of the structure/function relationship of the various elongator subunits.

Weaknesses:

1. Overstating conclusions:

The authors claim: "Further, we show that impaired function of the complex leads to proteome instability and consequent neurogenesis defects, myelin loss and neurodegeneration. Elp2 mutations perturb protein homeostasis by reducing expression of the genes that guide cell cycle and translation, and up-regulating protein degradation in neurons and oligodendroglia." Importantly, the demonstration of the direct causality of these processes is not sufficiently strong, nor are the mechanistic links between these events rigorously demonstrated, to support these conclusions.

Response:

We would argue that there is substantial novelty to the study presented. All previous *in vivo* studies of Elongator function in the CNS have utilised either conditional knockout mouse models or in the case of our own previous study (Kojic *et al.*, *Nature Communications* 2018), an ENU mouse mutant. No previous studies of Elongator in the CNS have utilised patient-specific mutations engineered into the germ line. For example, number of studies have used Elp3 conditional alleles to create LOF in selected cells in the CNS and of course these studies have led to interesting findings. In contrast, by modelling missense patient mutations in the germ line we have been able to recapitulate the complex neuro-developmental phenotype of the patients and unlike all previous studies (including our own) we observe and analyse a spectrum of Elongator dysfunction in the CNS. Of course there is and should be some overlap with previously described Elongator function, but there are important novel outcomes as well such the link between perturbed Elp2 function and the intact neural connectome, as just one example from this work.

No previous studies that report Elongator mutation in CNS disease patients have included any functional data, let alone the extraordinarily complex analyses we present including biochemistry, mouse models, functional neurological and behavioural studies etc. Overall, this allows us to define the subset of tRNAs controlling the phenotype including the aberrant connectome and autism spectrum disorder, as well the various CNS cells and cell processes responsible.

Co-senior author Glatt is clearly an international leader in the biochemistry of the Elongator complex and its components (Glatt *et al.*, 2012/2015/2016; Dauden *et al.* 2017/2019; Krutyhołowa *et al.*, 2019; Lin *et al.*, 2019). This manuscript presents the first full heterologous reconstitution of the enzymatically active human Elp123 subcomplex. Using this experimental system, we are the first to model clinically relevant Elongator mutations *in vitro* and study their direct influence in the activity of human Elongator. Therefore, we provide unprecedented insights into the structure/function relationship of this key complex and set a new gold-standard to analyse the consequence of patient derived amino acid substitutions or other functional sites (e.g. phosphorylation sites) in Elongator subunits *in vitro*.

Whilst we have not yet mechanistically defined that perturbation of a specific protein specifically leads to a specific singular aspect of this complex phenotype we would argue that to require this level of causality sets an extraordinarily high bar for novelty in this relatively new field. Of course, over the next years we will pull apart the specific mechanisms downstream of epitranscriptomic dysregulation in the individual cellular and system components when Elongator is clearly playing a major regulatory role based upon these studies, and those of others.

In order to address the reviewer concern that the manuscript was “overclaiming” insight we have modified the abstract to which the reviewer was referring as, “Further, we show that impaired function of the complex leads to proteome instability, neurogenesis defects, myelin loss and neurodegeneration.”

Overall, we believe the objective evidence considering the state of the field supports a view that this study is not preliminary but rather a novel and comprehensive insight into the role of Elongator function in CNS neurodevelopmental disease directly relevant to the many patients who carry these mutations.

To provide further clarity around these impacts and respond in part to reviewer comment we have re-titled the manuscript: *“Mutations in the Elongator complex perturb the epitranscriptome and lead to a complex neurodevelopmental phenotype in patients including intellectual disability and autism”*.

2.a While the authors claim that the mutated Elp2 protein is much less stable, it becomes unclear how significant that is for the subsequent protein studies. Elp2 levels are never measured in any of the mouse models – are they reduced at all? Or all the deleterious mouse effects the result of aberrant protein function? This point must be addressed. On Pg. 7: “No significant destabilization was observed for the L98F and H206R mutations.” Yet, the H206R mutation is one of the most serious mutations, and the mouse model that is used for the majority of their subsequent studies – so how meaningful is the protein stability decrement? Furthermore, the mouse H206R mutation does have a decreased melting temperature although the human does not. Which means this statement on pg. 7 is not accurate “The relative severity of the individual mutations is almost identical for murine and human Elp2 proteins.”

Response: We have analyzed Elp2 protein levels in the brain tissue of *Elp2H206R*, *Elp2H206R/R464W* and wild-type mice as suggested by the reviewer. The data are included in **Supplementary Figure 12**. Elp2 expression is reduced in the brain tissue of the *Elp2* mutant animals, which is consistent with our *in vitro* data demonstrating that the *Elp2* mutations perturb the protein stability. The mutation seems to mainly affect the Elp123 subcomplex, as Elp1 levels were also found to be reduced in the mutants, whereas Elp4 expression was not significantly affected. These findings concur with our proteomics data showing the reduced Elp1, Elp2 and Elp3 levels in the brain tissue of the *Elp2* mutant mice relative to their wild-type littermates (**Figure 8e** and **Supplementary Data 2**).

Moreover our *in vitro* data on Elp2 protein biochemistry clearly show that it is very well worth distinguishing between single protein and complex stability. It is common that proteins that are in physiological conditions work as a part of a bigger complex are less stable and more fragile when they are expressed alone *in vitro*. In fact, we undertook extra efforts to establish assays and test all human *Elp2* mutations in both scenarios – the respective individual Elp2 proteins alone and in the context of its molecular partners Elp1 and Elp3. We would like to re-emphasize that our work represents the very first study that shows the expression and purification of human Elp123 and disease-related variants. Both human and mouse H206R show a slightly decreased melting temperatures – their behavior is alike. The statement has been rephrased to: *“The relative severity of the individual mutations is similar for murine and human Elp2 proteins.”*

2.b Yet if protein stability is reduced for the “majority” of mutations, a surprising finding is that the Elp123 complex forms normally. This raises the interesting question of whether Elp2 has functions independent from its association with the other elongator subunits to

form the Elongator complex. This potential independent function of Elp2 is never addressed but should be.

Response: The observation that decreased stability of proteins carrying the *Elp2* mutations does not lead to abolishment of complex formation was also surprising for us at first. But after consideration - it aligns well with the fact that both mice and human patients carrying *Elp2* mutations are not embryonically lethal. Importantly, any impairment of Elongator-dependent modification levels (even several %) was shown to lead to the severe lethal phenotypes. Thus, the conclusion that our findings would imply independent functions of Elp2 is difficult to support. As previously explained, the mutations indeed influence the thermal stability of Elp2 proteins *in vitro*, but do not impair the Elp123 complex assembly *per se*, as stated: “Therefore, the respective *Elp2* mutations do not affect Elongator at the stage of complex assembly.” Here we show for the first time the reconstituted human Elp123 complex. This provides a novel opportunity to understand its function and the consequences of patient-derived mutations on the Elongator activity. The Elongator complex can be properly produced with the presented mutations, but it cannot perform tRNA modifications as efficiently as without these substitutions.

2.c More perplexing is that although the complex forms normally, all 5 mutations tested had reduced ACO hydrolysis rates. So does protein stability have anything to do with the decreased ACO hydrolysis? Again, mechanistic links between these findings were not made clear.

Response: We believe the reviewer has not completely appreciated that we are measuring both individual parameters independently – namely protein stability (of the individual proteins) and ACO hydrolysis rates. Obviously, we would expect that the tRNA-dependent ACO hydrolysis rates of the mutants is much closer to the genuine phenotypic effect that is observed in the patients and the mice. In fact, the observation that the human *Elp2H206R* mutation shows only a slight effect on protein stability, but decreased activity is a clear indicator that the biochemical activity of the Elp123 complex is the more relevant parameter.

It should also be noted that the ACO hydrolysis is induced by tRNA that binds to Elp1 and conducted by the catalytic Elp3 subunit (as stated in the text: “...*human Elp123 (hElp123) sub-complex, which harbors the enzymatically active Elp3 subunit and is able to bind tRNAs via the C-terminus of Elp1 and the active site of Elp3.*”). For this reason alone, the data are complementary and provide added value, rather than reflecting inconsistencies in our data, analyses or conclusions. In fact, the presented data clearly show that the observed phenotype is correlated with enzymatic activity of Elongator performed by the entire complex and not the single protein.

The text has been rephrased to provide further clarity : “*Our results clearly demonstrate that the identified patient-derived mutations influence the stability of both tested mammalian Elp2 proteins and have detrimental effect on the ability of hElp123 to induce the initial step of the m^5 modification reaction.*”

3.a Questions about brain morphogenesis calculations. Table S2 – were the relative brain weights normalized? Brains in the mutants are overall smaller, so of course each individual brain structure/nucleus will be smaller than that in the control, but are the mutants' individual structure volumes proportional to the reduction in brain mass?

Response: The volumes of different brain structures were not normalized, but the absolute values were provided. We have normalized the volumetric data (% of the total brain volume) as suggested by the reviewer and highlighted the structures in the table that were significantly affected in the *Elp2H206R* mice (**Supplementary Table 2.**).

3.b Same for Fig. 7 a & b – in addition to the data shown in those figures, the thickness of cortex and VZ should be normalized to thickness of the entire section (ventricle to outer cortical layer) since you have already shown the brains are smaller.

Response: We have now included the normalized values in **Figure 7a**. The thickness of the Cux1⁺ and Ctip2⁺ layers were measured relative to the overall cortical thickness (from the top cortical layer to the white matter; layers I-VI), as explained in the Methods section. Normalized cortical thickness was also quantified for different ages (14.5 dpc, P7 and adult brain) and the data are included in **Supplementary Figure 8**.

3.c The question is whether the relative proportion of mitotically-active cells compared to post-mitotic cortical layers differ between mutant and control. Reduced brain size has been shown before for *Elp1* and *Elp3* mutant mice and should be mentioned.

Response: To address this, we quantified a proportion of apical (Sox2⁺ cells; most of these are cycling progenitors) relative to post-mitotic progenitors (Dcx⁺ cells) and found that there was a higher portion of the apical progenitors in the mutant brains (**Figure 7c**). We have already demonstrated the reduced production of newborn intermediate progenitors in the mutants relative to control (Sox2⁺Tbr2⁺ cells; **Figure 7c**). Together with the new data, this suggests that microcephaly in the *Elp2H206R* brains is not only accounted for by a reduced number of apical progenitors (Sox2⁺) and thus, intermediate progenitors (Tbr2⁺) and newborn neurons (Dcx⁺; relative to wild-type); but also the reduced ability of these apical progenitors to generate new neurons. This is likely a consequence of their premature cell-cycle exit and death via apoptosis (**Figure 7d**).

Additionally, we have now included in manuscript reference to previous studies reporting that conditional loss of *Elp1* and *Elp3* led to a reduction in brain in mice (page 19 of the manuscript).

4. Dendritic branching defects (Fig. 6) has been shown before too by work from four separate labs (Weil, Tourtellotte, Lefcort and Nguyen) for *Elp1* and *Elp3* loss-of-function. Similarly, several of these groups (and others, e.g. in *C.elegans*) have also shown reductions of tubulin acetylation in neurons with reduction of elongator subunits. However, demonstration of alterations in spine number is new and an interesting finding.

Response: Indeed the findings of the Nguyen laboratory for example on reduced branching and tubulin acetylation in the cortical neurons upon conditional *Elp1* and *Elp3* silencing⁴ are

consistent with ours (noted on page 12 of the manuscript). As the reviewer noted, alteration in the spine number of cortical neurons in an Elongator mutant is novel in the field and is of great interest given the phenotype of *Elp2* missense mutation patients.

5.a For Figure 7c & d, how was the cell quantification conducted? Please describe the methods used. Also, in contrast to what is claimed in the manuscript, Ki67 labels mitotically active progenitor cells throughout the cell cycle, not just in S phase.

Response: The methods used for the cell quantification are described in the Methods section (under “*Immunofluorescent staining, imaging and analyses*”). The statement that Ki67 labels mitotically active progenitor cells only in S-phase has now been removed.

5.b The data in Fig. 7d are informative and interesting, showing that the % of Brdu+/Ki67- cells is reduced in the mutant. However, it is unclear how to interpret the Ki67 or pHH3 data alone since again, the proportion of progenitor cells vs. post-mitotic neurons should be calculated. Since the brains are overall smaller one would predict to see fewer absolute numbers of cells in the mutant (unless cell density is perturbed?) – but the question is are all cell types reduced equally or primarily progenitors or post-mitotic neurons? For example, based on the Sox2 and Tbr2 staining, it looks like proportionally, there is an increase in progenitor cells relative to post-mitotic neurons in the mutants but this needs to be quantified.

Response: We have revised **Figure 7** and removed Ki67 and pHH3 data the reviewer refers to, as the data demonstrating a premature cell cycle exit (**Figure 7d**) is sufficient and compelling to show a proliferation defect in the cycling progenitors. Furthermore, the defects identified in a neurosphere assay (**Figure 7e**) and transcriptomic analysis (**Figure 7f-g**) support the reduced proliferation of the progenitors in the brain tissue of the *Elp2* mutants.

At this stage of embryonic development (14.5 dpc), the brains are not overall significantly smaller, but clearly reduced cortical thickness is observed (**Supplementary Figure 8**). Thus, we quantified the number of apical and intermediate progenitors and post-mitotic neurons relative to wild-type brains to see whether this is an early neurogenesis defect. Indeed, we demonstrate that early progenitors are being affected in addition to the post-mitotic ones (**Figure 7**). The cell density is not perturbed according to the new data we have included in **Figure 7a**. We quantified the proportion of progenitor cells vs. post-mitotic neurons as advised by the reviewer (refer to our **response to comment 3.c**) and found that in addition to having reduced numbers of apical progenitors relative to control, the transition of these cells to post-mitotic progenitors is reduced. This is likely due to premature cell cycle arrest in the progenitors and their increased death (**Figure 7d**), but other mechanisms affecting this transition may be involved as well. We can speculate that the expression of the genes driving this transition may be altered due to their dependence on the Elongator, however this is beyond the scope of this study to provide definitive causality.

5.c Also, given the increase seen in Brdu+/Ki67- cells, one should also check to see if the progenitors are dying e.g. with TUNEL or cleaved caspase 3.

Response: We performed these analyses as suggested by the reviewer and found increased cell death in the developing cortex of the mutant mice based on cleaved caspase-3 expression (**Figure 7d**).

6. The transcriptome data by itself make it difficult to ascribe causality. First of all, Elongator is required for translation, meaning: do progenitors exit the cell cycle because in the absence of Elp2 they are incapable of translating the mRNAs for the transcription factors required for neurogenesis OR are the levels of these genes reduced because there are fewer progenitors in the mutants as shown in Fig. 7? Are the transcription factors that drive neurogenesis AA-codon biased? To attribute the reduction in neurogenesis to a reduction in the Elp2-regulated transcriptome, would require showing that the key genes that mediate neurogenesis are in fact reduced in the mutant because of codon-bias. Alternatively, these genes could be downstream of a gene that is codon-biased, but at least some attempt to connect these mechanisms must be made to explain the processes that mediate the Elp2 mutant phenotype.

Response: As stated in the manuscript: *“We recognize that many of the DE genes are likely indirectly affected downstream targets and that key regulatory factors are commonly expressed at relatively low abundance.”* To enrich for the genes that have a strong potential to be cell identity genes, we performed TRIAGE analysis. As the reviewer noted, it is difficult to predict whether specific transcription factors that mediate neurogenesis would be reduced in the *Elp2^{H206R}* brains due to the codon usage bias, as they could simply be downstream of key regulatory genes that are codon-biased. Thus, these transcription factors would not necessarily be codon-biased themselves, and the key regulatory genes upstream of them would likely still be at low abundance despite the enrichment analysis. Based on this, we believe that attempting to analyse the codon usage bias of specific transcription factors that drive neurogenesis identified in our transcriptomic analysis may lead to over-interpretation of the data.

The down-regulated genes likely reflect a lower number of cycling progenitor cells in the mutant brains. We used the transcriptome analysis to identify cell populations (based on enriched gene expression patterns) being affected in the mutant brain. These data complements our histological analyses.

7. Nicely, the authors show that the Elongator-mediated mcm5U modifications are reduced on tRNAs in the mutants, which supports several previously published data on Elongator function in multiple species (yeast to humans) and sets the stage for the subsequent proteomic analysis. However, the authors fail to mine their proteomic data to determine whether the proteins that mediate neurogenesis or CNS circuit formation are in fact reduced and also translated from AA-biased codons. Without that analysis, it's hard to draw conclusions about the functional connections between the transcriptomic, proteomic and developmental changes exhibited by the Elp2 mutant mice.

Response: Our data clearly show that down-regulated proteins are indeed AA codon-biased and among them, we identified proteins involved in cell cycle regulation, amino acid synthesis, RNA transport, all the process crucially important for neurogenesis. Moreover, pathways implicated in protein misfolding and degradation were found amongst the most

significantly enriched categories, further supporting translational defects in the developing brain of the *Elp2* mutants. This provides a solid link between impaired function of the complex and reduced proliferative capacity of the neural progenitors resulting in microcephaly. As noted in our **response to comment 6**, key regulatory factors driving specifically neurogenesis are commonly expressed at relatively low abundance (especially in a crude brain lysate) and thus, it can be challenging to detect them at RNA and/or protein level. The proteins involved in cell proliferation and translation found to be down-regulated in our proteomic analysis can be downstream some other key proteins that mediate neurogenesis, but there is a possibility that due to their low abundance, we were not able to detect them as differentially expressed. We agree with the reviewer that a further follow up on this would be an exciting and promising approach to define the molecular drivers responsible for the phenotype. Overall our study supports a functional connection between the altered function of the Elongator and attenuation of the complete neurogenic program, which our proteomic analysis does support. As stated in the Discussion section of the manuscript: *“Future studies need to be directed towards the analyses of single cell types and their specialized proteomes.”*

8.a Interpretation of mouse behavior studies (and patient phenotype) are perhaps misguided. Both the mouse and patient neurological exams indicate that these patients and mice have profound CNS deficits including pronounced motor problems. The majority of patients do not move at all, have severe hypotonia and spasticity, while the mice have tremors, gait problems, reduced grip strength, hypotonia and hind limb claspings – the latter a classic CNS descending pathway lesion. That means these patients/mice are not necessarily a classic model of Autism and/or intellectual disability – instead their entire CNS seems to have suffered from a systematic dysfunctional development, in particular including sensorimotor function. That raises a few questions including whether the patients have additional mutations and/or copy number variants that exacerbate the phenotype, rather than a dysfunctional *Elp2* being the sole culprit. However, the fact that recapitulation of the motor problems is obtained in the *Elp2* mutant mice at least supports the *Elp2* mutation driving the CNS dysfunction in the patients.

Response: Sensorimotor abnormalities are commonly present in ASD and/or ID and occur early during development⁵⁻⁸. Of course as spectrum disorders there those who are less severely affected. Thus, it is not surprising that both the patients and *Elp2* mutant mice have a motor phenotype in addition to ID/ASD. All patients were analysed through clinical genomics services and whole-exome sequencing did not identify any candidate variants other than *ELP2*. Moreover, parents of all patients were found to be heterozygous carriers of the *ELP2* disease-causing variants confirming the homozygous single gene recessive molecular pathology. Of course WES is less sensitive to the detection of some CNV but the clinical geneticist authors of this study are sanguine that the mode of inheritance as proposed is fully supported. Modelling the specific *ELP2* mutations identified in the patients in mice not only recapitulated the motor phenotype as noted by the reviewer, but also global developmental delay with microcephaly, repetitive behaviour and abnormal vocal phenotypes, all commonly associated with ID and ASD-like behaviour in mice. Moreover, our DT-MRI analyses of the *Elp2* mutant mice revealed abnormal connectivity between some crucial brain regions involved in cognition, social interaction, motor function and

emotional and behavioral response. Based on the above, we are confident that *ELP2/Elp2* mutations are sole contributors to the phenotype.

8.b That being said though, while it's clear the patients in addition to their motor problems also have ID, it's less clear how severely the mice display ASD or ID behaviors. Their repetitive behavior data fits with ASD, but not, to my knowledge, with ID. However, the repetitive behavior test is fairly short: the mice had a 10-minute habituation, then recorded for 10 minutes. They only groomed for 20 seconds over 10 minutes. It's also possible that the repetitive grooming is a self-soothing behavior in response to stress. As harmless as it seems, putting the mouse into a novel chamber, even with the habituation period, could be stressful and trigger grooming. Perhaps it is not an ASD-related repetitive behavior, but an abnormal stress response. This is certainly not a deal-breaker, but a longer recording period and assessment in a home-cage environment would be more informative of an ASD-type phenotype.

Response: To address the reviewer's concern, we performed additional analyses where each mouse was recorded in its home cage for 30 minutes for cumulative time spent grooming all body regions and the absolute number of bouts (**Figure 4g** and **Supplementary Figure 4g**). Both the repetitive behaviour and abnormal vocal phenotypes we observed in the mutants have widely been associated with ASD features in mice⁹⁻¹². As we were limited in our ability to assess their social interaction, learning and memory due to the very severe motor phenotype, we analyzed the brain tissue of the *Elp2* mutants using DT-MRI and assessed tractography and the connectome. The neuroimaging confirmed that the affected structures correspond to the abnormalities often associated with ID and ASD (as discussed in the manuscript).

8.c The authors claim an ASD-like reduction in ultrasonic vocalization (USV) communication. The experiment is reasonable – maternal separation of pups triggers USV from the pups. Some autism models have fewer USVs in that paradigm, suggesting that the mice do not communicate well. However, impairment of several neurotransmitter systems can influence USVs (5HT, GABA, opioids, cannabinoids, vasopressin, oxytocin). In summary, there's not a clear-cut pattern that ASD models always have reduced USVs. In fact, in a *Mecp2* KO model of Rett syndrome (typically considered as part of the autism spectrum) KO mice have increased USVs.

Response: We agree with the reviewer - ASD mouse models can have both increased and reduced USV and there is no a clear-cut pattern, nevertheless, alternations in USV have been widely shown in these models. We did say that: "*Consistent with clinical findings and other ID/ASD mouse models¹², Elp2 mutant neonates had a severe communication deficit in the form of a markedly reduced number and duration of ultrasonic calls.*" So, the phenotype of the mutants is consistent with the communication deficit in the patients and other ID/ASD mouse models, but of course we agree this does not imply that all ID/ASD models have reduced USV.

8.d Finally, the major ASD characteristic that the authors did not address at all is social interaction. There are multiple ways to assay it, that it seems like a very reasonable experiment if one wants to interrogate ASD behaviors. Its possible that the mouse motor

problems may preclude conducting the social interaction experiments but at least this point should be addressed since it is a classic hallmark of ASD.

Response: Yes, we attempted to further characterize the ASD phenotype in a 3-chamber social interaction tests and ID phenotype by performing active place avoidance test (for assessing the spatial memory). However, the reviewer predicted the motor phenotype of the *Elp2* mutant mice was very severe by P50-P60 when the mice were tested. Thus, it would be difficult to say whether the animals truly have reduced social-interaction and impaired memory and learning skills, or whether this is a consequence of advanced motor defects. As explained in our **response to comment 8.b**, this prompted us to analyse their brain connectome and tractography via DT-MRI and confirm ASD/ID-like phenotype based on neuroimaging.

9. Importantly, whether ER stress is truly responsible for the cell death of PC can't be determined by the data shown in Fig.S8 – there is no quantification of dying PCs and to conclude that ER stress is the cause is premature based on the preliminary data shown. Studies that demonstrate ER stress are far more comprehensive than the studies done here. Fig. S9 does have some quantification of the loss of glial cells, which is interesting. Do the APC+ oligodendrocytes express *Elp2* or is their loss an indirect effect of the loss of Purkinje cells? That would be important to determine. Also, the images in both Fig. S8 and S9 should include a merged view to show overlap in expression of the markers.

Response: Degeneration of Purkinje neurons (PNs) reflected in their reduced number over age has already been demonstrated and quantified (**Supplementary Figure 10c; Supplementary Figure 8c** in the previous version of the manuscript). Our previous study on *Elp6* mutant mice demonstrating PN degeneration likely caused by protein misfolding and ER-stress induced apoptosis¹³, together with other studies^{14,15} have used the same immunofluorescence-based method to determine ER stress in PNs as we did in this study (analyze expression of ER-stress induced transcription marker CHOP). We did not quantify the number of PNs expressing CHOP/Ub/CC3, as these cells were rare in the brains of wild-type controls. Although one could always perform more comprehensive studies we believe that given previously published approaches it is reasonable to conclude a link between ER stress and PNs. We observed a massive PN degeneration in the *Elp2* mutants just like we have in the *Elp6* mutants¹³ and we analyzed Ub/CHOP/CC3 expression in these cells to confirm that the mechanism corresponded to the one we observed in the *Elp6* mutant animals. This led us to investigate whether a similar mechanism underlaid the myelin loss, which was confirmed in both *Elp2H206R* and *Elp2H206R/R464W* mice.

Following the suggestion from the reviewer we determined that mature oligodendrocytes (APC⁺ cells) do express *Elp2* and we included these new data in the **Supplementary Figure 11a**. Merged views are now included in **Supplementary Figure 10** and **Supplementary Figure 11 (Supplementary Figure 8 and Supplementary Figure 9** in the previous version of the manuscript respectively).

10.a There are some confusing inconsistencies at several points in this study. For example, between images shown of mouse tissue/structures and the associated quantified data. Fig. S8a, the P60 brains of the mutants and control do not look as dissimilar as they do in Figs.

S4A at P21. Why? Additionally, for the brains depicted in Fig. 4A, the H206R mutant brain does not look smaller than the WT which raises questions as to the data shown in Fig. 4d.

Response: We apologise for the confusion – this is our error since we failed to include the appropriate scale bars which were different for each of the brains in **Supplementary Figure 8a (Supplementary Figure 10a** in the new version of the manuscript). This has been corrected now. There is some variation in the brain and body size/weight among mutants, so we have replaced the image and used a more representative image of the *Elp2H206R* whole brain in **Figure 4a**.

10.b In addition, it seems as if *Elp2* is not required in cerebellar progenitors in contrast to its requirement in cortical progenitors, based on the data in Fig. S8c. That is an interesting difference and warrants comments by the authors. In contrast, PC number is significantly reduced by P60 in the mutants, suggesting a degenerative process. Do the *Elp2* patients have ataxia? Or are their movement imparities so severe that ataxia can't be discerned?

Response: We agree with the reviewer that PN development seems not to be affected in the *Elp2* mutants, but there is a subsequent degenerative process, similar to that which we demonstrated in the *Elp6* mutant mice¹³, where PN degeneration was the only CNS phenotype. In order to understand why this is the case, conditional KO studies and analyses of single cell types and their specialized proteomes are required. We have included this in the Discussion section on page 20 of the manuscript. We can only speculate at this stage that perhaps the proteome of mature PNs is AA codon-biased (thus, dependent on the Elongator), unlike the ones in the precursor cells (which is not the case when it comes to the cortical progenitors). We agree this is a fascinating question and further studies should be designed to address this.

Ataxia is present in *Elp2* Patients. Patient 1 (**Supplementary Table 1**.; the *ELP2* variants found in this patient have been modelled in mice in this study) has ataxia, whilst other patients have severe motor impairments or diplegia/paraplegia. The mice seem to have ataxia as well based on their severe motor defects (scored tremor and gait abnormalities shown in **Figure 4e** and **Supplementary Figure 4e**). The severity of the motor phenotype in the mice prevented us from performing further behavioural tests for ataxia, like rotarod or balance beam.

10.c A major source of confusion is the mouse lines used for the different experiments. The authors state the mice die by around P20 on the C57BL/6 background but can live up to P60-120 when put on a DBA2/J background. That raises the question of the connection between the developmental histological/immunochemical analyses done in the mice on the C57BL/6 background and the mouse behavior and adult work done in mice on the DBA2/J background. For example, is life span extended to such a dramatic extent by changing background because the neurodevelopmental impairments identified in the C57/bl6 line do not occur in the DJA2/J cross? If so, how are we to interpret the behavioral abnormalities? The consistency in phenotype between these two different mouse lines (both with the same H206R mutations) needs to be clarified.

Response: Changing the genetic background of the mice was done in order to be able to study the consequences of the mutations in adult animals. Nevertheless, the phenotypic features on both genetic backgrounds in mice are highly consistent (**Figure 4** and **Supplementary Figure 4**). Moreover, the phenotype of *Elp2H206R* mice is also consistent with the one of the *Elp2H206R/R464W* animals (**Supplementary Figure 4**). The neurodevelopmental impairments identified in the C57BL/6 line do occur on the DBA2/J background as well, given that in both lines we observed microcephaly (**Figure 4d** and **Supplementary Figure 4d**), reduced cortical thickness and neurogenesis defects at the histological level (comparable to the ones presented in **Figure 7**; data not included for the DBA2/J strain). Hence, we do believe that there is a consistency in the neurodevelopmental phenotype between two different mouse lines, which was the main focus of this study. In addition, there is a variation in the clinical phenotypes of the patients with the *ELP2* variants (**Supplementary Table 1.**), although the neurological phenotypes in the patients were very similar. The reason for the life span being extended on a different genetic background in mice remains unclear. Life span also varies markedly in the patients, so there are likely genetic background effects underlying the spectrum of the phenotype as well as stochastic events, as often seen in various neurodevelopmental diseases¹⁶.

11. Novelty regarding our understanding of the Elongator complex function in neural development and CNS function. Many of the major findings regarding these events have been demonstrated in previous publications, most of which are not cited by the authors. Thus, the conclusion of the paper as stated “our work reveals the role of epitranscriptomic processes in neural development and homeostasis which, when perturbed, lead to profound CNS defects in both humans and model systems” is not novel. While the authors show that *Elp2* does participate in the same CNS steps that we already know *Elp1* and *Elp3* are required for, which is interesting although not at all unexpected based on a decade of work on the Elongator complex from yeast to *C.elegans* to mouse to humans, what is missing from this study is a deeper probe of the mechanisms mediating these aberrant processes.

Response: The authors included all relevant citations in the manuscript where applicable. The growing field of the Elongator research includes many relevant studies, but it is not possible to reference them all in this manuscript given the breath of our study and the limited number of references allowed. If the reviewer feels as though we have failed to reference a particularly key study to make a point not already made then we will do our best to accommodate.

The sentence the reviewer refers to has been modified to: “*More generally, our work is the first comprehensive study of the clinically relevant Elongator mutations that reveals the role of epitranscriptomic processes in brain development and homeostasis which, when perturbed, lead to profound CNS defects in both humans and model systems.*”. With respect to the issue of novelty please see our response to Question 1.

Reviewer #2 (Remarks to the Author):

Elongator is a protein complex mainly involves in tRNA modifications and whose dysfunction has been associated with several neurological disorders, ranging from Familial

dysautonomia, ALS, to intellectual disabilities. Previous animal models showed that loss or reduced Elongator activity in the nervous system affects the developmental program of neurogenesis and alters neuronal fitness in the central and autonomic nervous system, further leading to neurodegeneration. Here new pathological human variants of Elp2 are described and humanized mouse models are used to decipher the impact of those variants on brain development and maintenance. This work nicely show that these mice suffer from microcephaly and cerebellar degeneration. Moreover, Elp2 variants impairs Elongator activity leading to brain defects associated with changes in connectivity that may underlie specific behavioral defects.

This ms is interesting and overall well written but is unfortunately quite descriptive and would beneficiate from additional mechanistic insights. Moreover, some conclusions do not stand because data should be revisited under new light for proper interpretation.

Major concerns

12. The functional demonstration that Elp2 variants affects the ability of the Elp123 sub-complex to promote tRNA-induce acetyl-CoA hydrolysis in vitro is suggestive but not demonstrative of the negative impact of Elp2 variants on the activity of Elongator as a whole.

Response: We believe these findings are definitive. We have previously determined the high-resolution Cryo-EM structure of the yeast Elp123 subcomplex and have shown that Elp123 is sufficient to bind tRNAs and to induce the hydrolysis of ACO. We are very much aware that the Elp456 ring is involved in the modification reaction as well, as we have previously solved the high-resolution crystal structure of the yeast Elp456 complex and showed its positioning within the full complex using negative EM. We have recently also demonstrated that the presence of the ring reduces the binding affinity of tRNAs to the complex and believe that the ring is crucial for clearance of modified tRNAs from the catalytically active Elp123 subcomplex. Here, we have now managed to produce the human Elp123 complex and use the same well-validated experimental system to test the specific effect of patient derived mutations in Elp2 on the initial activation step. Therefore, we unambiguously claim that: *“identified patient-derived mutations (...) affect the ability of hElp123 to induce the initial step of the cm⁵ modification reaction”*. Our presented work in this manuscript also pioneers future studies that will use our system to test the direct influence of mutations on the activity of the complex.

It should also be noted that the ACO hydrolysis is induced by tRNA that binds to Elp1 and conducted by the catalytic Elp3 subunit. For this reason alone, the data are complementary and provides added value, rather than reflect inconsistencies in our data, analyses or conclusions.

In summary, the performed experiments are sufficient to link observed decreased level of tRNA modification in mutated mice with decreased enzymatic activity of the Elongator complex and with decreased stability. Our claims are fully supported by the data presented in the manuscript.

13. Regarding the behavioral screen. Some experiments, such as self-grooming should be confirmed in adult mice (p21 is adolescence). ASD is also classically associated with sociability defects and authors should assess social preference and social recognition in adult mice.

Response: Data on self-grooming in the DBA2/J strain were obtained using P60 mice as specified in the figure legend (**Figure 4g**). We are not able to obtain this data using the C57BL/6 strain as these mice do not survive post P21-P23 (**Supplementary Figure 4**).

Please refer to our **response to comment 8.d** (Reviewer 1) in regard to the social interaction test.

14. Regarding the connectome studies, these parameters should be acquire in mature brain, not in adolescents who are still undergoing brain maturation (synapse pruning and connectivity refinement).

Response: ID and ASD are neurodevelopmental disorders and thus, we performed these analyses at P21, when the neuronal connections are being established in the developing brain. At this age, neurogenesis has been completed, synapse initiation, maturation and pruning have been achieved, but synaptic morphological changes still take place¹⁹, which reflects on the connectome alternations. We could perform all the analyses in the more mature brain, but we do not believe this reflects a clinically relevant or particularly informative approach. Our DT-MRI imaging showed that the mice have reduced connectivity between brain regions involved in cognition, social interaction, motor function and emotional and behavioural response in comparison to their healthy littermates, which correlates to the ID/ASD clinical presentation.

15. The histological characterization of the cortex is problematic and leads to misinterpretations. The distribution of Cux1 staining across cortical layers is unexpected as all layers are positives in Figure 7a. To assess microcephaly, not only the thickness should be measured (done here), but also the absolute number of cells as well as their density. This is critical since dendritogenesis is strongly affected in mutant (see figure 6) and this can in addition to progenitor defects, lead to progressive changes of cortical thickness and neuron density after birth (progressive microcephaly). The analysis of brain weight and dimensions at different stages (during development and after birth) would help deciphering if there is indeed a worsening of the microcephaly after birth.

Response: We thank the reviewer for comment. We re-imaged the Cux1-labelled brain sections indicated by the reviewer and adjusted the image exposure to a more appropriate level. As a result, Cux1 staining was restricted to the upper cortical layers (**Figure 7a**). As suggested by the reviewer, absolute number of cells as well as their density was quantified, and the data is included in **Figure 7a**. The organization of the upper and lower cortical layers seem to be preserved and no changes were observed in cell density. Thus, the overall reduction in the cortical thickness is due to the neurogenesis defect and reduced neuronal levels rather than additional reduction in neuron density. We quantified relative cortical thickness at different stages (14.5 dpc, P7 and adult brain; **Supplementary Figure 8**) as suggested. Given that the relative measures were preserved across different stages,

microcephaly does not seem to progress after birth, but is mainly caused by early neurogenesis defects.

16. There are also semantic issues. First, the term “neural stem cells” is vague and incorrect as projection neurons are generated by both VZ and SVZ cells that are named apical progenitors (APs; or radial glial cell, RGCs) and basal progenitors (BP; or intermediate progenitors), respectively.

Response: The changes were incorporated in the manuscript and the term “neural stem cells” removed.

17. Other issues refer to the analysis of cell proliferation. First, both VZ and SVZ progenitors are indeed cycling progenitors. Second, Ki67 immunostaining marks most cell cycle phases and is not restricted to cells entering into S-Phase. Also, the way the cell cycle exit is assessed is wrong, a 2 hours BrdU pulse does not give enough time for a cortical progenitor to exit the cell cycle, which also questions the quality of the overall histological analysis performed in the manuscript.

Response: The statements that cycling progenitors come from SVZ and that Ki67 labels mitotically active progenitor cells only in S-phase are removed. We repeated the experiment and quantified a percentage of BrdU⁺ Ki67⁻ cells out of total BrdU⁺ cells after a 24-hour pulse of BrdU. These data are now included in **Figure 7d**.

18. Increased apoptosis during cortical development has been shown to be a cause of microcephaly in animal models, it would be important to check for apoptosis markers during development and not just in the adult cerebellum and mature oligodendrocytes.

Response: To address this, we have analysed apoptosis levels and found increased cell death in the developing cortex of the mutant mice based on cleaved caspase-3 expression (**Figure 7d**).

19.a For the neurosphere assay, there is no SVZ purification step and thus cycling progenitors must come from both VZ and SVZ. The conclusion of the paragraph is thus wrong.

Response: We thank the reviewer for bringing this to our attention. We have changed the sentence to: “*Next, we isolated neural progenitors from the developing cortices and cultured them in vitro in a neurosphere assay.*” in order to be more accurate, as there was no SVZ purification. Nevertheless, we believe the conclusions are valid since we specifically dissected out cortical structures of mutant and control 14.5 dpc embryos and cultured cells in a neurosphere assay, in which neurospheres are formed by cycling progenitors²⁰. We found a significant decrease in the number and size of the spheres derived from the *Elp2H206R* embryos, pointing to a proliferation defect and therefore, a decreased ability of the cortical progenitors to generate neurons.

19.b There is also a major comment regarding the interpretation of the cortical phenotype. There is no good experimental demonstration that *Elp2* mutation induces cell cycle exit and

previous work reported that loss of Elongator activity in cortical progenitors leads to microcephaly as a result of a defect in the fate of VZ progenitors, thereby changing the balance between direct and indirect neurogenesis. This seems to happen also for the *Elp2* humanized mice who show a significant reduction of newborn IP production (see figure 7c). In other words, it is necessary to assess direct versus indirect neurogenesis to take a final conclusion about the cellular mechanism of microcephaly.

Response: We believe our study clearly demonstrates a premature cell cycle exit in the brain cortex of the *Elp2* mutant mice based on BrdU/Ki67 analyses (**Figure 7d**), indicating that the mutation indeed induces cell cycle arrest. Moreover, we used a neurosphere assay to further support our findings related to the proliferation defects in the *Elp2* mutants (**Figure 7e**). Gene ontology analyses of DE genes in mutant vs. control developing brain tissue also showed that cell proliferation was found among significantly down-regulated categories (**Figure 7h**).

Although *Elp2* mutant mice have a significant reduction in newborn IPs, this can be simply a result of having a reduced number of APs (**Figure 7b**), but also their premature cell cycle exit and death (**Figure 7d**). The study the reviewer refers to did not observe a reduction in the AP but only IP number, which led them to investigate direct vs. indirect neurogenesis¹⁷. The different findings in this study related to the AP number can be potentially explained by using a conditional KO vs. a germline mutation in our study. We have addressed this in the Discussion section on page 20, where we also explained that: *“Nevertheless, further research is warranted to explore direct versus indirect neurogenesis program in the developing brain of the Elp2 mutants.”* This is not in the scope of our study but we have demonstrated an overall reduction of the neuronal layers caused by premature depletion of the progenitor pool due to cell-cycle arrest and increased cell death. Of course, we cannot completely exclude various other mechanisms contributing to it, like reduced indirect neurogenesis.

19.c The transcriptional analysis of dissected cortices revealed reduced expression of genes that are strongly enriched in the ventral forebrain progenitors during development (top genes such as *Ascl1*, *Gsx2*..), which suggests that the tissue dissected include more than the cortex. What is the relevance of reduction of expression of these genes in the cortex? This is problematic because Elongator is also important for cortical interneurons as previously reported. Thus the molecular analysis is difficult to interpret in regard to projection neurons. This part needs a better mechanistic understanding of the phenotype.

Response: As the reviewer noted, we analyzed the transcriptomes of the mutant and wild-type forebrains structures, not only cortices (we revised this statement in the manuscript on page 15). The TRIAGE analysis enabled us to enrich for the genes that have a strong potential to be cell identity genes and the transcriptome analysis was used to identify cell populations being affected in the mutant brain. *Gsx1*, *Gsx2* and *Ascl1* were found among the most down-regulated genes in the *Elp2^{H206R}* developing brain, which suggests that there is a very early neurogenesis defect originating from the lateral ganglionic eminence (LGE) progenitors (this is now included in the manuscript on page 15).

We followed up the reviewer's advice and further analyzed our DE and TRIAGE data in order to see whether other markers specific for interneurons were up/down expressed. In addition to already highlighted down-regulated genes in the DE and TRIAGE analyses plots, we added *Dlx2* and *Dlx5* genes as significantly reduced in both analyses. Given that *Dlx2*, *Dlx5* and *Ascl1* mark progenitors of cortical interneurons (whilst *Gsx1*, *Gsx2* and *Ascl1* are expressed in early neural progenitors), our data suggest a reduction in interneuron progenitors in the mutant mice at this stage of the brain development.

This prompted us to analyze mature interneurons in adult brains of the *Elp2H206R* mice and indeed, parvalbumin-expressing interneurons were found to be reduced in number in both cerebral cortex and hippocampus (**Supplementary Figure 9**). Thus, the *Elp2* mutations do not affect only neurogenesis in the murine brain, but also development of interneurons. The mechanism of their impaired development is likely based on cytoskeleton defects affecting their migration and branching as previously reported¹⁸. However, as the previous studies included a targeted conditional KO approach, it is important to perform similar analyses using mouse models of clinically relevant germline mutations in the Elongator subunits.

20. It is also surprising that the authors focused on chronic UPR-induced neurodegeneration in the cerebellum without checking ER stress and UPR in the cortex as this pathway was previously shown and experimentally tested by other to explain the microcephaly phenotype. This needs to be addressed for *Elp2* humanized mice in their cortex.

Response: The study to which the reviewer found that a conditional *Elp3* KO in cortical neurons led to the microcephaly phenotype as a result of elevated ER stress and UPR, by performing transcriptome analyses of 14.5 dpc forebrain tissue¹⁷. We performed the same analysis in this study and did not find upregulated UPR and ER stress-related transcripts. Accordingly, the microcephaly phenotype in the *Elp2* mutant mice does not seem to be a consequence of ER stress and UPR.

21. Elongator has been involved in the regulation of postmitotic interneuron migration, thus what is the phenotype the cortical interneurons and their progenitors in *Elp2* humanized mice?

Response: Please refer to our **response to comment 19.c**.

Minor concerns:

22. Why switching from H206R (for axonal tracts analysis Fig5) to H206R/R464W (for dendritic complexity fig6) to H206R mouse model again (for proliferation studies, fig7)?

Response: Defective neuronal morphogenesis (**Figure 6**) was found in both models (*Elp2H206R* and *Elp2H206R/R464W*), but we decided to show data for the *Elp2H206R/R464W* mice to demonstrate a consistency between the two models in regards to some of the neurodevelopmental defects we analysed in both models (although, the study focused mainly on the *Elp2H206R* mice).

23. Typos: abstract: a series of patients

Response: This has been edited.

24. Figure S6C: the blot is poor, its exposition is too long to make quantitative measurement.

Fig S6C : WB analysis for adult or embryonic samples ?

Response: The blot has been repeated. Western blot analysis was performed using adult brain tissue extracts. This is now included in the Figure legend (**Supplementary Figure 6c**).

25. Change in the main text that Tbr2 are in the SVZ and Sox2 in the VZ and not the opposite, and change cycling progenitors for radial glial cells :

« Analyses of Elp2H206R cortical progenitors showed that both intermediate progenitors (IPs; Tbr2+) in the ventricular zone and cycling progenitors (Sox2+) in the subventricular zone (SVZ), were affected (Figure 7c). »

Response: The changes are incorporated in the manuscript.

Reviewer #3 (Remarks to the Author):

In the manuscript by Kojic et al., the authors found ELP2 mutations in ID and ASD patients, and performed biochemical experiments using recombinant ELP1/2/3 proteins, and physiological experiments using Elp2 mutant mice. Although preceding studies already described that ELP2 mutations were associated with ID (Ref 13-15), Kojic et al. made Elp2 mutant mice, and directly proved that Elp2 is responsible for ID, providing physiological insights into the onset of ID. The physiological analyses in this manuscript are fine, but there are many undefended conclusions made from molecular analyses. Specific comments are described below.

Major points:

26. General. There is a paper published in 2018 on Elp2 KO mouse (Lu et al. BBRC. 2018. PMID: 29723529), which described a different phenotype from the authors' report. I wonder how the authors would interpret their results.

Response: The study to which the reviewer refers to did not describe how the Elp2KO mouse was made (only mentioned "Elp2 transgenic knockout") and whether this is a germline or conditional Elp2KO²¹. In addition, the study did not describe the overall phenotype of the animals but focussed on the renal pathology only. Thus, it is difficult to compare these two Elp2KO mouse models and comment on potential phenotypic differences. Previous studies from a number of groups including our own have demonstrated that germline Elp1KO²², Elp3KO²³ and Elp6KO¹³ resulted in early embryonic lethality in mice. Yet, in Lu *et al.* (2018), Elp2KO mice survived postnatally. Thus, without confirming that this was a germline Elp2KO, we cannot comment on the model other than

to note that particular finding is at odds with the literature and all of our own findings across multiple Elp gene knockouts.

Here we demonstrated that loss of Elp2 is lethal in two different mouse strains (C57BL/6 - please refer to **Supplementary Figure 3**, and DBA2/J – data not shown), which supports numerous *in vitro* and *in vivo* studies demonstrating that loss of any of the Elongator subunits is detrimental for its function and results in embryonic lethality in mice.

27. In Figure 2, the authors performed gel filtration experiments and melting curve experiments using recombinant human ELP2 to show that recombinant ELP2 proteins harbouring patient mutations are structurally less stable compared to WT ELP2. Although the results are beautiful and suggestive, the results are not sufficient to say that "Elp2 mutations perturb the protein stability" as written in the paragraph title in page 6. This is because the authors only checked the structural stability of recombinant proteins in test tubes and not within cells. To defend such a conclusion, it is necessary to check at least one of the followings: 1) comparing WT and mutant Elp2 protein steady-state levels in mouse brains, 2) comparing WT and mutant Elp2 protein steady-state levels in culture cells (if antibody to endogenous Elp2 is not available, tagged Elp2 can be used), 3) comparing decay speed of WT and mutant Elp2 in culture cells using a translation inhibitor such as cycloheximide.

Response: We have analyzed Elp2 protein levels in the brain tissue of *Elp2H206R*, *Elp2H206R/R464W* and wild-type mice as advised by the reviewer. Please refer to our **response to comment 2.a (reviewer 1)**.

28. In Figure 7f-7h, the authors performed RNA-seq and Gene Ontology analysis of mRNAs that were up- or down-regulated in Elp2 mutant mouse brains. In the Methods section page 48, the authors wrote that they "found 917 (536 up-regulated, 381 down-regulated) out of 14266 genes being differentially expressed in the comparison of mutant vs control". However, in the Gene Ontology analysis in Figure 7h, the authors seemed to have analyzed only about 35 down-regulated genes and 23 up-regulated genes. The authors should show the criteria to have chosen the genes. Also, the authors should show P-values for each enriched gene ontology term. These two are prerequisites to judge if the results of gene ontology analysis are valid.

Response: The GO enrichment analysis was done on the top 50 genes identified by the TRIAGE-based score (i.e. DE genes that have a strong RTS value as described in the Methods section). Lists of these top 50 gene are included in **Supplementary Data 1**. For the panel, we selected relevant GO terms (i.e. 10 terms for down-regulated and 9 terms for up regulated) and identified genes linked with the terms. We provided p-values (and other relevant statistics) for these terms in **Supplementary Data 1**.

29. In Figure 8d-g, the authors performed peptide mass spec and used the results to make analysis on the length and codon usage of proteins that were up- or down-regulated in Elp2 mutant mouse brains. In Figure 8f, the authors checked up- or down-regulated proteins and compared their "length". There seems to be at least two problems with this analysis. First, in Figure 8d, there are hundreds, maybe thousands of proteins that were up- or down-

regulated, but in Figure 8f, only 11 up-regulated proteins and 25 down-regulated proteins were used. The authors should clearly write the criteria to have chosen the proteins. Otherwise, I cannot remove skepticism on the arbitrariness of the analysis and conclusion. Secondly, to conclude that *Elp2* mutation is "specifically affecting long transcripts" as written in the Summary, statistical analysis is absolutely required. For this kind of analysis, for example, I see people comparing e.g. >100 proteins that were upregulated against >100 proteins that were downregulated, drawing a cumulative distribution curve (horizontal axis: protein length, vertical axis: cumulative protein number %), and performing a Kolmogorov-Smirnov statistical test.

Response: The volcano plot in **Figure 8e** (**8d** in the first version of the manuscript) is based on differentially expressed peptides (FDR < 0.05) where p-values were determined by Empirical Bayes moderated t-statistics and corrected for multiple hypothesis testing using Benjamini-Hochberg (BH) procedure, as per defaults of limma. The peptide-level peptide p-values were then converted to protein-level p-values using Sime's adjustment and were subjected to multiple test corrections using the BH procedure, using topSplice and diffSplice framework in limma. This resulted in 11 up- and 25 down-differentially expressed (DE) proteins at FDR 0.1, as noted by the reviewer. The barcode plot of protein lengths (**Figure 8g**; **Figure 8f** in the previous version of the manuscript) is based on DE proteins identified by Sime's adjustment (36 in total, 11 up- and 25 down-regulated). The barcode enrichment plot suggested that there was an enrichment for shorter proteins in up- and longer proteins in down-regulated proteins in the mutants. We quantified the significance of enrichment in each set of up- and down-regulated proteins using Wilcoxon test in R. For each set of up- and down-regulated proteins (n = 11 and 25 respectively), we tested if there was a difference between shorter and longer proteins in the set using Wilcoxon test. The short and long criteria were defined with respect to the median protein lengths in the study. Hence, shorter lengths refer to smaller than median, and longer refers to larger than median. We have now included the p-values obtained from Wilcoxon tests in the barcode enrichment plots (**Figure 8g**).

30. In Figure 8g, among the codons whose usage decreased in down-regulated proteins, CAA codon (Glutamine), GAA codon (Glutamate), and AAA codon (Lysine) are highlighted. These codons are decoded by tRNAs bearing mcm5s2U, and not by tRNAs bearing mcm5U (e.g. Johanson et al. MCB. 2008. PMID: 18332122). So, in addition to ncm5U and mcm5U measured in Figure 8a & 8b, it is absolutely necessary to check the mcm5s2U modification status. Also, as *Elp2* is required for ncm5s2U modification, it is preferable to also measure ncm5s2U level.

Response: Whilst mcm⁵s²U has been widely analyzed in various mammalian tissues and the pathways involved in its biogenesis have mainly been identified²⁴, including finding of the fourth thiolated tRNA in humans (Arg^{UCU})²⁵, the presence and biogenesis of ncm⁵s²U in mammalian cells have not been reported so far. Thus we are unaware of such a modification described by the reviewer.

However, we did quantify mcm⁵s²U in the brain tissue of the *Elp2* mutant mice and controls and found lower levels of this modification in addition to the reduction of ncm⁵U and mcm⁵U. The data are now included in **Figure 8c**. We based our tRNA analysis using the

HPLC/MS method on the previously established HPLC and MS parameters for tRNA modification analysis in mammalian cells which did not include $\text{mcm}^5\text{s}^2\text{U}$ ²⁶. Nevertheless, we demonstrated a significant reduction in mcm^5U and ncm^5U modifications that directly depend on the activity of the Elongator complex and showed them to be reduced in the brain tissue of the *Elp2* mutants. This is consistent with other research in the field exploring the consequences of Elongator mutations on its tRNA modification role in mammalian cells^{13,23,25,27,28}.

31. In Figure 8g, the authors discussed about NAA codons (which are decoded by $\text{mcm}^5\text{s}^2\text{U}$ -bearing tRNAs). $\text{mcm}^5\text{s}^2\text{U}$ is known to promote the translation of NAA codons, so this is good. On the other hand, in Figure 8g, the authors did not discuss about codons that are decoded by $\text{mcm}^5\text{U}/\text{ncm}^5\text{U}/\text{ncm}^5\text{s}^2\text{U}$ -bearing tRNAs, although all of these modifications are synthesized using Elp2. mcm^5U in tRNAs (Arg, Gly) anticodon promotes recognition of G-ending Arg and Gly codons AGG and GGG. ncm^5U in tRNAs (Val, Ser, Thr) promotes recognition of G-ending GUG, UCG, ACG codons (Johanson et al. MCB. 2008. PMID: 18332122). It is unfair and biased to only discuss about codons decoded by tRNAs bearing $\text{mcm}^5\text{s}^2\text{U}$ -bearing and not discuss about codons decoded by tRNAs bearing $\text{mcm}^5\text{U}/\text{ncm}^5\text{U}/\text{ncm}^5\text{s}^2\text{U}$. The authors should either discuss about these codons or remove Figure 8g and its implications.

Response: We believe the reviewer is referring to studies performed in yeast (*S.cerevisiae*)²⁹ and there is strong evidence emerging from recent studies indicating that the decoding process dependent on the Elongator-mediated tRNA modifications differs fundamentally between yeast and mammalian cells^{25,30}. The evident ribosome slowdown has been shown for AAA and CAA codons, but not for GAA codon (found among down-regulated in our data) in yeast cells lacking the Ncs2 protein required for subsequent thiolation reaction to form the $\text{mcm}^5\text{s}^5\text{U}$ -bearing tRNAs. Moreover, only these two tRNAs (Lys^{UUU}, Gln^{UUG}), not tRNA-Glu^{UUC} are required to rescue the observed phenotype in yeast. CTT and CTA codons found to be enriched in down-regulated proteins in our study are not modified in yeast. As noted above, a fourth tRNA-Arg^{UCU} with the $\text{mcm}^5\text{s}^2\text{U}$ modification has been identified in humans²⁵. Therefore, we have found this result particularly interesting, supporting clear discrepancies between yeast and mammals.

As pointed out earlier (**response to comment 30.**), the $\text{ncm}^5\text{s}^2\text{U}$ modification has not been described in the literature up to date as far as we have been able to determine. The mcm^5 moiety is a substrate for the thiolation reaction to form the $\text{mcm}^5\text{s}^5\text{U}$. Therefore, we believe this figure is of a particular value to the field.

Minor points:

32. In Figure 8g, as a codon whose usage decreased in down-regulated proteins, TAA codon is highlighted. However, TAA is a stop codon, not requiring a tRNA or tRNA modification. So, this is peculiar. I wonder how the authors would interpret this result.

Response: This codon was highlighted in the figure but in a grey colour, indicating that its' usage bias among down-regulated proteins was not significant. We have now removed the stop codon from the plot (**Figure 8h**).

References

- 1 Cheishvili, D. *et al.* IKAP/Elp1 involvement in cytoskeleton regulation and implication for familial dysautonomia. *Human Molecular Genetics* **20**, 1585-1594, doi:10.1093/hmg/ddr036 (2011).
- 2 Jackson, M. Z., Gruner, K. A., Qin, C. & Tourtellotte, W. G. A neuron autonomous role for the familial dysautonomia gene *ELP1* in sympathetic and sensory target tissue innervation. *Development* **141**, 2452-2461, doi:10.1242/dev.107797 (2014).
- 3 Hunnicutt, B. J., Chaverra, M., George, L. & Lefcort, F. IKAP/Elp1 is required in vivo for neurogenesis and neuronal survival, but not for neural crest migration. *PLoS One* **7**, e32050, doi:10.1371/journal.pone.0032050 (2012).
- 4 Creppe, C. *et al.* Elongator controls the migration and differentiation of cortical neurons through acetylation of α -tubulin. *Cell* **136**, 551-564 (2009).
- 5 De Jong, M., Punt, M., De Groot, E., Minderaa, R. B. & Hadders-Algra, M. Minor neurological dysfunction in children with autism spectrum disorder. *Dev Med Child Neurol* **53**, 641-646, doi:10.1111/j.1469-8749.2011.03971.x (2011).
- 6 Parma, V. & de Marchena, A. B. Motor signatures in autism spectrum disorder: the importance of variability. *J Neurophysiol* **115**, 1081-1084, doi:10.1152/jn.00647.2015 (2016).
- 7 Jeste, S. S. The neurology of autism spectrum disorders. *Curr Opin Neurol* **24**, 132-139, doi:10.1097/WCO.0b013e3283446450 (2011).
- 8 Flint, J. Genetic basis of cognitive disability. *Dialogues Clin Neurosci* **3**, 37-46 (2001).
- 9 Chao, H.-T. *et al.* Dysfunction in GABA signalling mediates autism-like stereotypies and Rett syndrome phenotypes. *Nature* **468**, 263-269, doi:10.1038/nature09582 (2010).
- 10 Schmeisser, M. J. *et al.* Autistic-like behaviours and hyperactivity in mice lacking ProSAP1/Shank2. *Nature* **486**, 256-260, doi:10.1038/nature11015 (2012).
- 11 Silverman, J. L., Tolu, S. S., Barkan, C. L. & Crawley, J. N. Repetitive self-grooming behavior in the BTBR mouse model of autism is blocked by the mGluR5 antagonist MPEP. *Neuropsychopharmacology* **35**, 976-989, doi:10.1038/npp.2009.201 (2010).
- 12 Scattoni, M. L., Crawley, J. & Ricceri, L. Ultrasonic vocalizations: a tool for behavioural phenotyping of mouse models of neurodevelopmental disorders. *Neurosci Biobehav Rev* **33**, 508-515, doi:10.1016/j.neubiorev.2008.08.003 (2009).
- 13 Kojic, M. *et al.* Elongator mutation in mice induces neurodegeneration and ataxia-like behavior. *Nature communications* **9**, 3195 (2018).
- 14 Yang, Y. *et al.* Disruption of Tmem30a results in cerebellar ataxia and degeneration of Purkinje cells. *Cell Death & Disease* **9**, 899, doi:10.1038/s41419-018-0938-6 (2018).
- 15 Lee, J. W. *et al.* Editing-defective tRNA synthetase causes protein misfolding and neurodegeneration. *Nature* **443**, 50-55, doi:10.1038/nature05096 (2006).
- 16 Mullis, M. N., Matsui, T., Schell, R., Foree, R. & Ehrenreich, I. M. The complex underpinnings of genetic background effects. *Nature Communications* **9**, 3548, doi:10.1038/s41467-018-06023-5 (2018).
- 17 Laguesse, S. *et al.* A dynamic unfolded protein response contributes to the control of cortical neurogenesis. *Developmental cell* **35**, 553-567 (2015).

- 18 Tielens, S. *et al.* Elongator controls cortical interneuron migration by regulating actomyosin dynamics. *Cell Res* **26**, 1131-1148, doi:10.1038/cr.2016.112 (2016).
- 19 Farhy-Tselnicker, I. & Allen, N. J. Astrocytes, neurons, synapses: a tripartite view on cortical circuit development. *Neural Development* **13**, 7, doi:10.1186/s13064-018-0104-y (2018).
- 20 Azari, H. & Reynolds, B. A. In Vitro Models for Neurogenesis. *Cold Spring Harb Perspect Biol* **8**, a021279, doi:10.1101/cshperspect.a021279 (2016).
- 21 Lu, S., Fan, H. W., Li, K. & Fan, X. D. Suppression of Elp2 prevents renal fibrosis and inflammation induced by unilateral ureter obstruction (UUO) via inactivating Stat3-regulated TGF- β 1 and NF- κ B pathways. *Biochem Biophys Res Commun* **501**, 400-407, doi:10.1016/j.bbrc.2018.04.227 (2018).
- 22 Chen, Y.-T. *et al.* Loss of mouse Ikbkap, a subunit of elongator, leads to transcriptional deficits and embryonic lethality that can be rescued by human IKBKAP. *Molecular and cellular biology* **29**, 736-744 (2009).
- 23 Bento-Abreu, A. *et al.* Elongator subunit 3 (ELP3) modifies ALS through tRNA modification. *Human molecular genetics* **27**, 1276-1289 (2018).
- 24 Hawer, H. *et al.* Roles of Elongator Dependent tRNA Modification Pathways in Neurodegeneration and Cancer. *Genes (Basel)* **10**, 19, doi:10.3390/genes10010019 (2018).
- 25 Yoshida, M. *et al.* Rectifier of aberrant mRNA splicing recovers tRNA modification in familial dysautonomia. *Proceedings of the National Academy of Sciences* **112**, 2764-2769, doi:10.1073/pnas.1415525112 (2015).
- 26 Su, D. *et al.* Quantitative analysis of ribonucleoside modifications in tRNA by HPLC-coupled mass spectrometry. *Nature protocols* **9**, 828 (2014).
- 27 Karlsborn, T. *et al.* Elongator, a conserved complex required for wobble uridine modifications in eukaryotes. *RNA Biol* **11**, 1519-1528, doi:10.4161/15476286.2014.992276 (2014).
- 28 Waszak, S. M. *et al.* Germline Elongator mutations in Sonic Hedgehog medulloblastoma. *Nature*, 1-6 (2020).
- 29 Johansson, M. J., Esberg, A., Huang, B., Björk, G. R. & Byström, A. S. Eukaryotic wobble uridine modifications promote a functionally redundant decoding system. *Mol Cell Biol* **28**, 3301-3312, doi:10.1128/mcb.01542-07 (2008).
- 30 Nedialkova, D. D. & Leidel, S. A. Optimization of codon translation rates via tRNA modifications maintains proteome integrity. *Cell* **161**, 1606-1618 (2015).

Reviewer #1 (Remarks to the Author):

I appreciate the detailed response the authors have made to the reviewers' concerns and for the changes they have made to the manuscript. I won't reiterate the strengths of the paper that I stated in my initial review, they still stand. Some of the concerns raised in my original review also still stand.

Response to the author's responses:

1. Their study is novel and an important contribution in that it is the first investigation of Elp2 in the human and mouse nervous system in health and disease. However, it is not the first description of "Elongator" in the human and mouse nervous system and this point is essential. Several mouse studies have analyzed the nervous system of Elp1 and Elp3 mutant mice and identified many of the same major deficits in these mice, including behavioral deficits, so to state that "no previous studies... have included any functional data" is simply not an accurate description of the literature.

Moreover, these are not the first data to analyze the CNS of patients with a genetic mutation in an Elongator subunit. The study by Axelrod et al, 2010 titled "Neuroimaging supports central pathology in familial dysautonomia" uses diffusion tensor imaging and shows reduced white matter tracts, frontal lobe atrophy etc in patients with the FD mutation in ELP1. There are several other studies showing intellectual impairment in some FD patients.

2. As for the structural data – absolutely the data from Dr. Glatt's team is superb and yet one more important contribution from his group to our understanding of the structure and function of the Elp123 complex.

3. In response to reviewer 2, this following statement was made by the authors "thus the Elp2 mutations do not only affect neurogenesis in the murine brain but also development of interneurons". This statement is non-sensical.

Comments on the revised manuscript.

1. The word "Elongator" in the title should be modified to "Elp2" to most accurately define the novelty of the study. Other people have already examined the Elongator complex subunits in the CNS of both humans and mice. Patients with ELP3 and ELP1 mutations/copy number variants do not all report ID nor ASD nor do the mouse models, and yet presumably their "Elongator" complex is also disrupted. This paper is on Elp2 and that should be reflected in the title.

2. Abstract: The authors state that these data show a link between "translation kinetics and brain development". No translation kinetics are interrogated in this study. tRNA modification is done as is a proteomic analysis – from those two investigations nothing can be concluded about "translation kinetics"

3. Pg. 4. Since the authors are not thoroughly citing the Elongator literature, they should restrict their language to an analysis of Elp2. Following up on the comments made at the beginning of this review, this is not the first study to demonstrate CNS pathology and behavioral impairments in patients or mice with mutations in Elongator subunits.

4. Fig. 4. In the initial review, a question was raised about whether the increased grooming is a stress response to a new environment and suggested that 30 minute observation in the home cage be conducted. The authors state that data is now in Fig. 4 and in supplemental Fig. 4 and "30 minutes in the home cage" is described in the Methods, but it's not clear in the figures or legends themselves that the time period depicted reflects that 30 minutes period in the home cage. Please clarify.

5. Fig. 7d Pg. 14. The authors conclude that progenitors are dying yet to definitely make that statement they would need to quantify the number of cleaved Caspase 3+/Ki67+ cells and show they are increased in the mutants. Also during a 24 hr BrdU pulse, many BrdU+ neurons will be

included when calculating the number of progenitors that have exited the cell cycle.

6. The conclusions about the role of protein degradation and ER stress in cell death are still preliminary based on the data included here. To definitively invoke a role for ER stress in cell death, see Laguesse et al 2015 as an example (amongst many) of a more systematic and rigorous analysis of UPR in cell death. Based on Chop and Ub staining alone, one can "suggest" that ER stress is potentially contributing the death of the adult purkinje neurons.

Minor:

1. Supplementary table 2 – says statistically significant changed structures are in red; ventricles are listed which were supposedly enlarged in the mutant, but the p value is 0.95 – so there is a mistake somewhere.

2. Pg. 16. Myelin "sheets" should be myelin "sheath". But its unclear what you quantified with the MBP stain – the commissures which are myelinated, or the scattered MBP stain in the brain - better to just say you saw a reduction in MBP.

3. Again, the last sentence in the Discussion needs to be modified: This study is not the first to show "the central role of the elongator" in the CNS. As just one more example, see the study by Karlsborn et al., 2014 entitled "Familial dysautonomia patients have reduced levels of the modified wobble nucleoside mcm(5)s(2)U in tRNA".

Reviewer #3 (Remarks to the Author):

To our previous comments #26, #28, #30, #31, and #32, the authors sufficiently responded in data and words. In addition, in response to #27, the authors added an important data in Supplementary Figure 12, showing decreased Elp3 mutant protein levels compared to WT Elp3. Related to the previous comments #27 and #29, but I still have two concerns listed below.

Major point:

- In response to the comment #29, the authors responded by showing their criteria to have chosen 35 up/down-regulated proteins in Elp3 mutant/WT, and by performing statistical analysis. The response itself is OK, but now that it is clear that only 36 up/down-regulated proteins were chosen, I have to point out another thing. In the abstract, the authors wrote "specifically affecting long transcripts enriched in AA-ending codons". This is a very general conclusion and a very strong conclusion. On the other hand, this observation is made on the analyses of only 36 proteins that were up/down-regulated. Amino acids encoded by AA-ending codons are present in almost all proteins of the proteome, and thousands of other proteins translated from long transcripts having many AA-ending codons were not up/down-regulated enough to be used for the analyses in Figures 8g and 8h. So, it is not appropriate to make a strong, general conclusion from Figures 8g and 8h. Nevertheless, these data may be somewhat suggestive and may be used to make discussions. And, the authors at least have enough data to say that Elp3 mutant brains show aberrant proteostasis and the unfolded protein response, which serve as a good explanation for mutant brain dysfunction. Thus, I think that Figures 8g and 8h should be removed from the main figure, and relevant words need to be removed from the abstract, and instead can be discussed in the Discussion section.

Minor point:

- In response to the comment #27, the authors added a nice data showing the decreased Elp3 mutant protein compared to WT Elp3 protein. This is good. One small concern is that the loading control Actin bands are saturated in the western blot, thus not serving as a sufficient loading control. Please run smaller amounts of total protein, just to confirm that approximately the same amount of total protein was run in the western blot.

Reviewer #4 (Remarks to the Author):

I was asked by Nature Communications to provide feedback on whether concerns raised by Reviewer 2 have been appropriately addressed by the authors as Reviewer 2 is no longer available, and to provide additional comments on the manuscript.

Kojic et al. report on their production and testing of a recombinant human ELP 123 subcomplex and how individual ELP2 patient variants impact the stability, assembly and activity of the subcomplex. Neuroimaging of humanized ELP2-compromised mouse models also shows microcephaly and other structural changes in the cerebral cortex that are consistent with studies in ID/ASD patients. The authors go on to show that the reduced cerebral cortex results from a reduction of the progenitor pool and increased progenitor cell death that is not a result of UPR and ER stress. Connectome-wise comparisons reveal that mutant brains have increased limbic system connections. The authors also include structural information regarding the location of the Elp2 mutations in relation to the 123 complex as a whole, and bioinformatic data looking at the impact of compromised Elp2 function on the expression of codon-biased genes.

The breadth and depth of the data included in this study are impressive and achieved through novel and innovative approaches. They add considerable weight to the growing body of evidence indicating that Elongator's primary and perhaps exclusive function in the cell is the modification of tRNAs and the regulation of codon-biased genes. Their study also adds insight into how loss of this function translates to the maldevelopment of the nervous system and neurodegeneration. This work adds significant new insight into the specific function of ELP2 in the context of Elongator as well as lending significant support to previous studies. The authors have also made significant strides toward addressing pertinent reviewer concerns. Please see my specific comments regarding Reviewer 2 comments below.

A few concerns that although minor, need to be addressed/corrected. The authors fail to cite previous studies by Goffena et al., 2018 that first identified long, AA-biased genes as specific Elongator targets. Referencing this past work not only acknowledges foundational studies that contributed to the current work, but also lends more support and credence to their findings. Also, on page 17 in the last paragraph before the discussion, the authors state that "up-regulated proteins are biased towards the Elongator-independent AG-ending codons (Fig. 8h)". Classifying AG-ending codons as "Elongator-independent" is incorrect and fails to correctly portray how the Elongator complex regulates the translation of codon-biased genes. AG-ending codons are in fact translated by U34 modified tRNAs as wobble codons and multiple studies suggest that Elongator added modifications decrease the translational efficiency of AG-ending codons (see Goffena et al. 2018 for specific references), keeping the levels of proteins encoded by AG-biased transcripts in check. Indeed, the authors' own findings that upregulated proteins show a biased usage of GAG codons demonstrates their dependence on Elongator. Additionally, in the legend for Figure 8h, the authors state that codons decoded by mcm5s2U-bearing tRNAs are capitalized, but cag, tag, and gag are shown in lower case. This is incorrect. cag, tag, and gag, as mentioned above, are decoded by mcm5s2U34-bearing tRNAs through wobble pairing. It is for this reason that Elongator dependence is specific to genes that are AA or AG-biased, not just AA- or AG-rich. This distinction is important as it explains why genes that use a higher than average # of AA-ending codons, but also a higher # of AG-ending codons, do not show altered protein levels in the absence of Elongator.

Reviewer 2 Major Concerns:

12. The functional demonstration that Elp2 variants affects the ability of the Elp123 sub-complex to promote tRNA-induce acetyl-CoA hydrolysis in vitro is suggestive but not demonstrative of the negative impact of Elp2 variants on the activity of Elongator as a whole.

Reviewer 2 makes the case that the data provided by the authors do not fully demonstrate the

impact of Elp2 mutations on the activity of Elongator as a whole. I agree with the reviewer to the extent that the author's in vitro system for studying the impact of Elp2 mutations on the function of the Elp123 subcomplex cannot be assumed to represent exactly what happens in vivo and in the context of the complete Elongator complex. However, I do not believe this significantly detracts from the value and knowledge gained regarding the specific function of Elp2 as a component of Elongator and how its loss likely contributes to the described molecular defects, cellular abnormalities, and the pathophysiology in patients possessing Elp2 mutations. In my opinion the authors do not overstate the implications of their results. That said, the authors' claims would be more accurate if they included the term "in vitro" in their concluding statements for many of their findings. In particular, I would recommend that the last sentence of the results section 3 be changed to read as follows,

Our results clearly demonstrate that the identified patient-derived mutations influence the stability of both tested mammalian Elp2 proteins and have detrimental effect on the ability of hElp123 to induce the initial step of the cm5 modification reaction in vitro and likely impose a similar effect in vivo.

13. This concern has been adequately addressed by the authors

14. I agree with the authors that the reviewer's suggestion that the connectome studies be performed in adult mice is not a more appropriate approach and would not be more informative.

15-19c. These concerns have been adequately addressed by the authors

20. It is also surprising that the authors focused on chronic UPR-induced neurodegeneration in the cerebellum without checking ER stress and UPR in the cortex as this pathway was previously shown and experimentally tested by other to explain the microcephaly phenotype. This needs to be addressed for Elp2 humanized mice in their cortex.

Author Response: The study to which the reviewer found that a conditional Elp3 KO in cortical neurons led to the microcephaly phenotype as a result of elevated ER stress and UPR, by performing transcriptome analyses of 14.5 dpc forebrain tissue¹⁷. We performed the same analysis in this study and did not find upregulated UPR and ER stress-related transcripts. Accordingly, the microcephaly phenotype in the Elp2 mutant mice does not seem to be a consequence of ER stress and UPR.

This is an important discrepancy and the authors' data showing that UPR and ER stress-related transcripts were not upregulated in the forebrain of their knockouts at 14.5 dpc should be included in the manuscript.

21-25. These concerns have been adequately addressed by the authors

Lynn George

We thank the reviewers for their time and insight. In response to these comments we have performed additional experiments, added data to one figure, modified two others, created a new supplementary figure and edited the text to provide greater accuracy and clarity.

Responses to reviewers' comments

Reviewer #1 (Remarks to the Author):

I appreciate the detailed response the authors have made to the reviewers' concerns and for the changes they have made to the manuscript. I wont reiterate the strengths of the paper that I stated in my initial review, they still stand. Some of the concerns raised in my original review also still stand.

Response to the author's responses:

1.a Their study is novel and an important contribution in that it is the first investigation of Elp2 in the human and mouse nervous system in health and disease. However, it is not the first description of "Elongator" in the human and mouse nervous system and this point is essential. Several mouse studies have analyzed the nervous system of Elp1 and Elp3 mutant mice and identified many of the same major deficits in these mice, including behavioral deficits, so to state that "no previous studies.... have included any functional data" is simply not an accurate description of the literature.

Response: The reviewer is referring to a statement in our previous response not the manuscript *per se*. Nonetheless, we would like to clearly state that we have had no intention to question the scientific value of previously published work by our colleagues in the field. Furthermore, we would like to reiterate that our study differs from previous *in vivo* studies addressing the role of Elongator in the CNS. Due to the embryonic lethality of Elongator null mice, different groups have used conditional knock-out systems or electroporated inhibitory RNAs in neuronal tissues for their studies. Conditional loss of function studies results in highly targeted phenotypes whereas the modelling of missense patient mutations where all cells in the body carry the mutation has allowed us to not only confirm some previous observations but also define the role of Elongator complex function in systems such as the neural connectome.

1.b Moreover, these are not the first data to analyze the CNS of patients with a genetic mutation in an Elongator subunit. The study by Axelrod et al, 2010 titled "Neuroimaging supports central pathology in familial dysautonomia" uses diffusion tensor imaging and shows reduced white matter tracts, frontal lobe atrophy etc in patients with the FD mutation in ELP1. There are several other studies showing intellectual impairment in some FD patients.

Response: Of course, we acknowledge that CNS-related defects have been identified in some FD patients. However, we believe that we have made no claims in either the manuscript or the previous response that this is the first paper to CNS phenotype a patient carrying an Elongator mutation. We have now rephrased the sentence that relates to the

CNS defects and Elongator mutations in the introduction section as follows - “A potential role for Elongator in the central nervous system (CNS) has been indicated through the finding that a mutation in the *Elp6* gene leads to Purkinje neuron (PN) degeneration in mice¹⁴, identified central pathology in some FD patients¹⁵ and the population sequencing studies that identified rare *Elp2* variants to be potentially associated with ID¹⁶⁻¹⁸. The sentence includes the reference to which the reviewer refer (reference #15).

We do believe that this work does presents a comprehensive clinical analysis of multiple *ELP2* patients, accurate patient-specific modelling of multiple missense mutations which faithfully recapitulate complex patient features in mice, and biochemical data defining the effects of these mutations on the Elongator complex and the consequent effects on translation.

2. As for the structural data – absolutely the data from Dr. Glatt’s team is superb and yet one more important contribution from his group to our understanding of the structure and function of the *Elp123* complex.

Response: We appreciate the recognition of our *in vitro* work and the important contributions to the field.

3. In response to reviewer 2, this following statement was made by the authors “thus the *Elp2* mutations do not only affect neurogenesis in the murine brain but also development of interneurons”. This statement is non-sensical.

Response: We perhaps could have rephrased this statement to “The *Elp2* mutations perturb the development of cortical projection neurons and interneurons.” Since is this a comment regarding our response and is not in the manuscript we have taken no further action.

Comments on the revised manuscript.

1. The word “Elongator” in the title should be modified to “*Elp2*” to most accurately define the novelty of the study. Other people have already examined the Elongator complex subunits in the CNS of both humans and mice. Patients with *ELP3* and *ELP1* mutations/copy number variants do not all report ID nor ASD nor do the mouse models, and yet presumably their “Elongator” complex is also disrupted. This paper is on *Elp2* and that should be reflected in the title.

Response: We have altered the title of the manuscript to “Mutations in the *Elp2* subunit of Elongator perturb the epitranscriptome and lead to a complex neurodevelopmental phenotype in patients including intellectual disability and autism”

2. Abstract: The authors state that these data show a link between “translation kinetics and brain development”. No translation kinetics are interrogated in this study. tRNA modification is done as is a proteomic analysis – from those two investigations nothing can be concluded about “translation kinetics”

Response: It is well established in the field that the lack (or reduction) of U₃₄ modifications would lead to changes in the translation elongation speed of the ribosome¹, ribosome occupancy and the co-translational folding dynamics of the emerging proteins²⁻⁴. We agree with the reviewer that our data lack direct evidence for kinetic changes but considering direct evidence for reduced tRNA modifications levels and specific changes in the proteome it seemed reasonable to deduct a link between “translation kinetics and brain development”.

Nonetheless, we have rephrased the last sentence of the abstract, which now reads as follows - “Collectively, our data demonstrate an unexpected role for tRNA modification in the pathogenesis of monogenic ID and ASD and defines Elp2 as a key regulator of brain development.”.

3. Pg. 4. Since the authors are not thoroughly citing the Elongator literature, they should restrict their language to an analysis of Elp2. Following up on the comments made at the beginning of this review, this is not the first study to demonstrate CNS pathology and behavioural impairments in patients or mice with mutations in Elongator subunits.

Response: As previously explained, we included all relevant citations in the manuscript where applicable (in the updated version, we have added 3 more references related to the Elongator and FD – references #12, #13 and #15). The growing field of the Elongator research includes many relevant studies, but it is not possible to reference them all in this manuscript given the breath of our study and the limited number of references allowed by the journal. If the reviewer feels as though we have failed to reference a particularly key study to make a point not already made, then we will do our best to accommodate.

To address the reviewer’s concern, we amended the sentence on page 5 and used “*Elp2* mutations” instead of “Elongator mutations”.

4. Fig. 4. In the initial review, a question was raised about whether the increased grooming is a stress response to a new environment and suggested that 30 minute observation in the home cage be conducted. The authors state that data is now in Fig. 4 and in supplemental Fig. 4 and “30 minutes in the home cage” is described in the Methods, but its not clear in the figures or legends themselves that the time period depicted reflects that 30 minutes period in the home cage. Please clarify.

Response: We have now amended the figures and figure legends to clarify the experimental setup as follows - on the graphs it is now stated “Self-grooming time (s/30 min)” and in the figure legends “(recorded for 30 min in a home cage)”.

5. Fig. 7d Pg. 14. The authors conclude that progenitors are dying yet to definitely make that statement they would need to quantify the number of cleaved Caspase 3+/Ki67+ cells and show they are increased in the mutants. Also during a 24 hr BrdU pulse, many BrdU+ neurons will be included when calculating the number of progenitors that have exited the cell cycle.

Response: Following the recommendation by the reviewer, we quantified the number of dying neural progenitors (Sox2⁺CC3⁺ cells) and demonstrated that there is an increase in apoptosis of these cells in the mutant brain (**Fig. 7d**). We used Sox2 as a marker of neural progenitors rather than Ki67 given that some of the cells are likely to exit the cell cycle (thus, they would be Ki67⁻) and then undergo apoptosis.

We agree that during a 24 h BrdU pulse, not only neural progenitors but also some fully differentiated neurons would be included in the quantification. Regardless, these are BrdU⁺ cells that exited the cell cycle in that time window and the data demonstrate that the number of these cells is significantly higher in the *Elp2* mutant mice than in controls. To avoid any confusion, we replaced the word “progenitors” with “cells” on page 14.

6. The conclusions about the role of protein degradation and ER stress in cell death are still preliminary based on the data included here. To definitively invoke a role for ER stress in cell death, see Laguesse et al 2015 as an example (amongst many) of a more systematic and rigorous analysis of UPR in cell death. Based on Chop and Ub staining alone, one can “suggest” that ER stress is potentially contributing the death of the adult purkinje neurons.

Response: We have based our work and conclusions concerning UPR induction on the mentioned previous in-depth analyses showing the specific activation of the PERK pathway after Elongator depletion⁵. Throughout the whole manuscript, we are very careful not to overstate any mechanistic conclusions concerning different routes of UPR in specific neuronal subtypes.

Page 16 – “Indeed, the expression of ER stress-induced transcription factor CHOP and elevated levels of ubiquitination and activated caspase-3 in the degenerating neurons (**Supplementary Fig. 11d,e**), suggest UPR-mediated apoptosis as the most likely route of neuronal death.”

Page 20 - “Despite up-regulation of the ubiquitin-proteasome pathway in proliferating neural progenitors, the neural progenitors seem to avoid the UPR and ER stress-triggered death, in contrast to PNs and oligodendroglia. This raises the intriguing question whether UPR-induced cell death selectively affects PNs and oligodendrocytes in the brain.”

We have rephrased the section on page 17, which now reads as follows –

“Thus, perturbation of the function of Elongator not only deregulates the neurogenic developmental program but seems to also induce UPR in mature neurons and oligodendrocytes, which further promotes neurodegeneration and myelin loss, respectively.”.

Minor:

1. Supplementary table 2 – says statistically significant changed structures are in red; ventricles are listed which were supposedly enlarged in the mutant, but the p value is 0.95 – so there is a mistake somewhere.

Response: The mistake has been corrected in the revised version.

2. Pg. 16. Myelin “sheets” should be myelin “sheath”. But its unclear what you quantified with the MBP stain – the commissures which are myelinated, or the scattered MBP stain in the brain - better to just say you saw a reduction in MBP.

Response: We have now changed this to “myelin basic protein (MBP)”.

3. Again, the last sentence in the Discussion needs to be modified: This study is not the first to show “the central role of the elongator” in the CNS. As just one more example, see the study by Karlsborn et al., 2014 entitled “Familial dysautonomia patients have reduced levels of the modified wobble nucleoside mcm(5)s(2)U in tRNA”.

Response: We have added the respective reference and rephrased the concluding statement at the end of the discussion session, which now reads as follows – “Using patient-derived germline mutations, we have directly shown the central role of the Elongator tRNA modifying complex in the developing and homeostatic brain in both humans and mice.”.

Reviewer #3 (Remarks to the Author):

To our previous comments #26, #28, #30, #31, and #32, the authors sufficiently responded in data and words. In addition, in response to #27, the authors added an important data in Supplementary Figure 12, showing decreased Elp3 mutant protein levels compared to WT Elp3. Related to the previous comments #27 and #29, but I still have two concerns listed below.

Major point:

- In response to the comment #29, the authors responded by showing their criteria to have chosen 35 up/down-regulated proteins in Elp3 mutant/WT, and by performing statistical analysis. The response itself is OK, but now that it is clear that only 36 up/down-regulated proteins were chosen, I have to point out another thing. In the abstract, the authors wrote “specifically affecting long transcripts enriched in AA-ending codons”. This is a very general conclusion and a very strong conclusion. On the other hand, this observation is made on the analyses of only 36 proteins that were up/down-regulated. Amino acids encoded by AA-ending codons are present in almost all proteins of the proteome, and thousands of other proteins translated from long transcripts having many AA-ending codons were not up/down-regulated enough to be used for the analyses in Figures 8g and 8h. So, it is not appropriate to make a strong, general conclusion from Figures 8g and 8h. Nevertheless, these data may be somewhat suggestive and may be used to make discussions. And, the authors at least have enough data to say that Elp3 mutant brains show aberrant proteostasis and the unfolded protein response, which serve as a good explanation for mutant brain dysfunction. Thus, I think that Figures 8g and 8h should be removed from the main figure, and relevant words need to be removed from the abstract, and instead can be discussed in the Discussion section.

Response: To address the concerns of this reviewer and the supportive comments by reviewer #4 (see separate response), we have now moved **Fig. 8g** and **8h** to the **Supplementary Fig. 14**. We have also removed the sentence the reviewer refers to from the abstract and used it in discussion as follows – “We have demonstrated that key stages of brain development are coordinated by the tRNA modification activity of the Elongator complex, which is essential for maintaining protein homeostasis in the developing cortex and post-mitotic brain cells. As previously proposed⁵¹, this activity seems to be specifically important for the translation of long transcripts, which are biased towards the use of AA-ending codons.”.

Minor point:

- In response to the comment #27, the authors added a nice data showing the decreased Elp3 mutant protein compared to WT Elp3 protein. This is good. One small concern is that the loading control Actin bands are saturated in the western blot, thus not serving as a sufficient loading control. Please run smaller amounts of total protein, just to confirm that approximately the same amount of total protein was run in the western blot.

Response: In the revised version we have replaced the previous over-exposed Actin blot with a less exposed version of the loading control (**Supplementary Fig. 13**). The conclusions are unaltered.

Reviewer #4 (Remarks to the Author):

I was asked by Nature Communications to provide feedback on whether concerns raised by Reviewer 2 have been appropriately addressed by the authors as Reviewer 2 is no longer available, and to provide additional comments on the manuscript.

Kojic et al. report on their production and testing of a recombinant human ELP 123 subcomplex and how individual ELP2 patient variants impact the stability, assembly and activity of the subcomplex. Neuroimaging of humanized ELP2-compromised mouse models also shows microcephaly and other structural changes in the cerebral cortex that are consistent with studies in ID/ASD patients. The authors go on to show that the reduced cerebral cortex results from a reduction of the progenitor pool and increased progenitor cell death that is not a result of UPR and ER stress. Connectome-wise comparisons reveal that mutant brains have increased limbic system connections. The authors also include structural information regarding the location of the Elp2 mutations in relation to the 123 complex as a whole, and bioinformatic data looking at the impact of compromised Elp2 function on the expression of codon-biased genes.

The breadth and depth of the data included in this study are impressive and achieved through novel and innovative approaches. They add considerable weight to the growing body of evidence indicating that Elongator's primary and perhaps exclusive function in the cell is the modification of tRNAs and the regulation of codon-biased genes. Their study also adds insight into how loss of this function translates to the maldevelopment of the nervous system and neurodegeneration. This work adds significant new insight into the specific function of ELP2 in the context of Elongator as well as lending significant support to previous studies. The authors have also made significant strides toward addressing

pertinent reviewer concerns. Please see my specific comments regarding Reviewer 2 comments below.

A few concerns that although minor, need to be addressed/corrected. The authors fail to cite previous studies by Goffena et al., 2018 that first identified long, AA-biased genes as specific Elongator targets. Referencing this past work not only acknowledges foundational studies that contributed to the current work, but also lends more support and credence to their findings. Also, on page 17 in the last paragraph before the discussion, the authors state that “up-regulated proteins are biased towards the Elongator-independent AG-ending codons (Fig. 8h)”. Classifying AG-ending codons as “Elongator-independent” is incorrect and fails to correctly portray how the Elongator complex regulates the translation of codon-biased genes. AG-ending codons are in fact translated by U34 modified tRNAs as wobble codons and multiple studies suggest that Elongator added modifications decrease the translational efficiency of AG-ending codons (see Goffena et al. 2018 for specific references), keeping the levels of proteins encoded by AG-biased transcripts in check. Indeed, the authors’ own findings that upregulated proteins show a biased usage of GAG codons demonstrates their dependence on Elongator. Additionally, in the legend for Figure 8h, the authors state that codons decoded by mcm5s2U-bearing tRNAs are capitalized, but cag, tag, and gag are shown in lower case. This is incorrect. cag, tag, and gag, as mentioned above, are decoded by mcm5s2U34-bearing tRNAs through wobble pairing. It is for this reason that Elongator dependence is specific to genes that are AA or AG-biased, not just AA- or AG-rich. This distinction is important as it explains why genes that use a higher than average # of AA-ending codons, but also a higher # of AG-ending codons, do not show altered protein levels in the absence of Elongator.

Response: Following the request by reviewer #3, we have moved the proteome and codon bias analyses to the supplementary information. We do agree with reviewer #3 that the number of identified down- and up-regulated genes is simply too small to make a general statement about the codon-bias and the underlying regulatory link. We do believe that in the context of available literature our data are still valuable, and we included the analyses (and the data) in the newly created **Supplementary Fig. 14**. Nonetheless, we do also agree with the comments of reviewer #4 and we have rephrased the confusing sentence about AG-ending codons and removed “Elongator-independent” on page 17.

Of course, we acknowledge the foundational work of Goffena *et al.* (2018) and note that it was already referenced in the manuscript (page 17, reference #47 in the old version and #51 in the updated version of the manuscript). To further highlight the importance of this study for our work and similar efforts in the field, we now rephrased the sentence including this reference as follows – “In accordance with the previous study identifying long, AA-biased genes as specific Elongator targets in peripheral neurons⁵¹, we found that the proteome of the developing cortex in the *Elp2* mutants was biased towards shorter proteins (**Supplementary Fig. 14a**).”. In addition, we included a comment about the codon bias of our proteomic analysis (and the previous study) in the final paragraph of the discussion on page 21 – “As previously proposed⁵¹, this activity seems to be specifically important for the translation of long transcripts, which are biased towards the use of AA-ending codons.”.

Reviewer #4 (Remarks to the Author): Reviewer 2 Major Concerns:

“12. The functional demonstration that Elp2 variants affects the ability of the Elp123 sub-complex to promote tRNA-induce acetyl-CoA hydrolysis in vitro is suggestive but not demonstrative of the negative impact of Elp2 variants on the activity of Elongator as a whole.”

Reviewer 2 makes the case that the data provided by the authors do not fully demonstrate the impact of Elp2 mutations on the activity of Elongator as a whole. I agree with the reviewer to the extent that the author’s in vitro system for studying the impact of Elp2 mutations on the function of the Elp123 subcomplex cannot be assumed to represent exactly what happens in vivo and in the context of the complete Elongator complex. However, I do not believe this significantly detracts from the value and knowledge gained regarding the specific function of Elp2 as a component of Elongator and how its loss likely contributes to the described molecular defects, cellular abnormalities, and the pathophysiology in patients possessing Elp2 mutations. In my opinion the authors do not overstate the implications of their results. That said, the authors’ claims would be more accurate if they included the term “in vitro” in their concluding statements for many of their findings. In particular, I would recommend that the last sentence of the results section 3 be changed to read as follows,

Our results clearly demonstrate that the identified patient-derived mutations influence the stability of both tested mammalian Elp2 proteins and have detrimental effect on the ability of hElp123 to induce the initial step of the cm5 modification reaction in vitro and likely impose a similar effect in vivo.

Response: We fully agree with the reviewer and we appreciate the acknowledgment of our biochemical work on the mouse Elp123 complex. We have rephrased and completed the sentence in section 3 to highlight that our results have been obtained *in vitro* and are like to reflect the *in vivo* situation. The sentence on page 9 now reads as follows – “Our results clearly demonstrate that the identified patient-derived mutations influence the stability of both tested mammalian Elp2 proteins and have detrimental effect on the ability of hElp123 to induce the initial step of the cm5 modification reaction *in vitro* and likely impose a similar effect *in vivo*.”.

13. This concern has been adequately addressed by the authors

14. I agree with the authors that the reviewer’s suggestion that the connectome studies be performed in adult mice is not a more appropriate approach and would not be more informative.

15-19c. These concerns have been adequately addressed by the authors

20. It is also surprising that the authors focused on chronic UPR-induced neurodegeneration in the cerebellum without checking ER stress and UPR in the cortex as this pathway was previously shown and experimentally tested by other to explain the microcephaly phenotype. This needs to be addressed for Elp2 humanized mice in their cortex.

Author Response: The study to which the reviewer found that a conditional *Elp3* KO in cortical neurons led to the microcephaly phenotype as a result of elevated ER stress and UPR, by performing transcriptome analyses of 14.5 dpc forebrain tissue¹⁷. We performed the same analysis in this study and did not find upregulated UPR and ER stress-related transcripts. Accordingly, the microcephaly phenotype in the *Elp2* mutant mice does not seem to be a consequence of ER stress and UPR.

This is an important discrepancy and the authors' data showing that UPR and ER stress-related transcripts were not upregulated in the forebrain of their knockouts at 14.5 dpc should be included in the manuscript.

Response: We would like to emphasize again that the conditional depletion of *Elp3* would likely lead to a more severe phenotype in cortical progenitors than the patient-derived mutations in our study. The data from our transcriptional analyses are available in **Fig. 7f** and **Supplementary Data 1**. In addition, we have now included the data showing that UPR transcripts are not upregulated in the developing forebrains of the *Elp2* mutant mice in **Supplementary Fig. 10**. We added a sentence on page 15 of our manuscript, which reads as follows – “In contrast to previous findings⁵⁰, we did not observe the induction of UPR in cortical progenitors using patient derived germline mutations (**Supplementary Fig. 10**).” This is also discussed on page 20 “Despite up-regulation of the ubiquitin-proteasome pathway in proliferating neural progenitors, the neural progenitors seem to avoid the UPR and ER stress-triggered death, in contrast to PNs and oligodendroglia.”.

21-25. These concerns have been adequately addressed by the authors

Lynn George

References

- 1 Ranjan, N. & Rodnina, M. V. Thio-Modification of tRNA at the Wobble Position as Regulator of the Kinetics of Decoding and Translocation on the Ribosome. *Journal of the American Chemical Society* **139**, 5857-5864, doi:10.1021/jacs.7b00727 (2017).
- 2 Zinshteyn, B. & Gilbert, W. V. Loss of a conserved tRNA anticodon modification perturbs cellular signaling. *PLoS Genet* **9**, e1003675 (2013).
- 3 Nedialkova, D. D. & Leidel, S. A. Optimization of codon translation rates via tRNA modifications maintains proteome integrity. *Cell* **161**, 1606-1618 (2015).
- 4 Chou, H. J., Donnard, E., Gustafsson, H. T., Garber, M. & Rando, O. J. Transcriptome-wide Analysis of Roles for tRNA Modifications in Translational Regulation. *Mol Cell* **68**, 978-992.e974, doi:10.1016/j.molcel.2017.11.002 (2017).
- 5 Laguesse, S. *et al.* A dynamic unfolded protein response contributes to the control of cortical neurogenesis. *Developmental cell* **35**, 553-567 (2015).

Reviewer #1 (Remarks to the Author):

I think this manuscript is considerably strengthened with the additions and modifications made by the authors in response to all 4 reviewer comments. My remaining concern is still the contextualization of their study in the much broader scope of the existing literature on Elongator function in the nervous system. For example, while the following paragraph is an improvement:

"Pg. 4 "A potential role for Elongator in the central nervous system (CNS) has been indicated through the finding that a mutation in the Elp6 gene leads to Purkinje neuron (PN) degeneration in mice¹⁴, identified central pathology in some FD patients¹⁵ and the population sequencing studies that identified rare Elp2 variants to be potentially associated with ID¹⁶⁻¹⁸."

it still reflects a very narrow grasp of what has already been published in the field about Elongator's critical role in the CNS. For example, from human studies alone, we know that variants in ELP3 are associated with ALS (Simpson et al., 2009; Kwee et al., 2012), mutations in ELP4 with ID, Autism and Epilepsy (Addis et al., 2015; Toral-Lopez et al., 2020; + 3 papers on ELP4 and Epilepsy) in addition to the several studies showing that ELP1 mutations in FD cause intellectual impairment. In mice, reductions in Elp1 interfere with CNS development and cause the death of adult CNS neurons in addition to causing striking deficits in learning and memory (Chaverra et al., 2017), while mutations in Elp3 exacerbate ALS degeneration via its role in modifying tRNA (Bento-Abreu et al., 2018). Furthermore there is a vast literature on the critical role for ELP1 in the retina, a CNS structure, in both FD patients and mice: patients go blind due to the progressive loss of retinal ganglion cells. I understand you can only cite a limited number of references, but one would hope the authors would contextualize their study within the larger collective perspective of already published studies on the Elongator complex in the nervous system.

Please understand, I am not trying to be obdurate, but rather hoping this interesting study and important contribution on ELP2/Elp2 function can enrich our broader understanding of how the individual elongator subunits each contribute to nervous system development and function.

Reviewer #3 (Remarks to the Author):

Authors addressed all comments by Reviewer. The reviewer appreciate the effort of the authors.

Reviewer #4 (Remarks to the Author):

The authors have adequately addressed all concerns raised by Reviewer #2 as well as separate concerns that I raised as an additional 4th reviewer. I recommend publishing of the manuscript in Nature Communications.

Lynn George

We are very grateful to the reviewers for their time and insight. Please find below the response to the remaining concerns of Reviewer #1. We have edited the text to reflect the proposed changes.

Responses to reviewers' comments

Reviewer #1 (Remarks to the Author):

I think this manuscript is considerably strengthened with the additions and modifications made by the authors in response to all 4 reviewer comments. My remaining concern is still the contextualization of their study in the much broader scope of the existing literature on Elongator function in the nervous system. For example, while the following paragraph is an improvement:

"Pg. 4 "A potential role for Elongator in the central nervous system (CNS) has been indicated through the finding that a mutation in the Elp6 gene leads to Purkinje neuron (PN) degeneration in mice¹⁴, identified central pathology in some FD patients¹⁵ and the population sequencing studies that identified rare Elp2 variants to be potentially associated with ID¹⁶⁻¹⁸."

it still reflects a very narrow grasp of what has already been published in the field about Elongator's critical role in the CNS. For example, from human studies alone, we know that variants in ELP3 are associated with ALS (Simpson et al., 2009; Kwee et al., 2012), mutations in ELP4 with ID, Autism and Epilepsy (Addis et al., 2015; Toral-Lopez et al., 2020; + 3 papers on ELP4 and Epilepsy) in addition to the several studies showing that ELP1 mutations in FD cause intellectual impairment. In mice, reductions in Elp1 interfere with CNS development and cause the death of adult CNS neurons in addition to causing striking deficits in learning and memory (Chaverra et al., 2017), while mutations in Elp3 exacerbate ALS degeneration via its role in modifying tRNA (Bento-Abreu et al., 2018). Furthermore there is a vast literature on the critical role for ELP1 in the retina, a CNS structure, in both FD patients and mice: patients go blind due to the progressive loss of retinal ganglion cells. I understand you can only cite a limited number of references, but one would hope the authors would contextualize their study within the larger collective perspective of already published studies on the Elongator complex in the nervous system.

Please understand, I am not trying to be obdurate, but rather hoping this interesting study and important contribution on ELP2/Elp2 function can enrich our broader understanding of how the individual elongator subunits each contribute to nervous system development and function.

Response: We appreciate the reviewer's suggestions and we edited the text accordingly. The text on page 4 now reads as follows – "Role for Elongator in the central nervous system (CNS) has been indicated through identified central pathology in some FD patients, including myelination¹⁴ and retinal defects^{15,16} and the population genomic studies that linked *ELP3* variants with amyotrophic lateral sclerosis (ALS)^{17,18}, variants in the *ELP2* gene with ID¹⁹⁻²¹

and *ELP4* mutations with ID, autism and epilepsy²²⁻²⁴. Furthermore, studies in mice showed that a germline mutation in the *Elp6* gene leads to Purkinje neuron (PN) degeneration²⁵, retina-specific loss of Elp1 leads to retinal ganglion cell degeneration²⁶ and a conditional Elp1 deletion in the CNS perturbs the development of cortical neurons ultimately leading to their death and consequential learning and memory impairment²⁷.”

Furthermore, on page 5 we added the following sentence – “Reduced levels of the Elongator-dependent tRNA modifications have been confirmed in FD, ALS and cerebellar ataxia as a consequence of *ELP1*¹³, *ELP3*³⁵ and *Elp6*²⁵ mutations, respectively.”

Reviewer #3 (Remarks to the Author):

Authors addressed all comments by Reviewer. The reviewer appreciate the effort of the authors.

Reviewer #4 (Remarks to the Author):

The authors have adequately addressed all concerns raised by Reviewer #2 as well as separate concerns that I raised as an additional 4th reviewer. I recommend publishing of the manuscript in Nature Communications.

Lynn George

Reviewer #1 (Remarks to the Author):

Very interesting and important contribution to the field.